# The mutational signatures of formalin fixation on the human genome

Qingli Guo [1,2], Eszter Lakatos[2,3], Ibrahim Al Bakir [2], Kit Curtius[2,4], Trevor A. Graham [2,3] ✉ & Ville Mustonen [1,5] ✉

Clinical archives of patient material near-exclusively consist of formalin-fixed and paraffin-embedded (FFPE) blocks. The ability to precisely characterise mutational signatures from FFPE-derived DNA has tremendous translational potential. However, sequencing of DNA derived from FFPE material is known to be riddled with artefacts. Here we derive genome-wide mutational signatures caused by formalin fixation. We show that the FFPE-signature is highly similar to signature 30 (the signature of Base Excision Repair deficiency due to *NTHL1* mutations), and chemical repair of DNA lesions leads to a signature highly similar to signature 1 (clock-like signature due to spontaneous deamination of methylcytosine). We demonstrate that using uncorrected mutational catalogues of FFPE samples leads to major mis-assignment of signature activities. To correct for this, we introduce FFPEsig, a computational algorithm to rectify the formalin-induced artefacts in the mutational catalogue. We demonstrate that FFPEsig enables accurate mutational signature analysis both in simulated and whole-genome sequenced FFPE cancer samples. FFPEsig thus provides an opportunity to unlock additional clinical potential of archival patient tissues.

Patient samples are routinely processed with formalin fixation and paraffin embedding (FFPE) by pathology laboratories around the world. FFPE preserves tissue morphology and enables immunohistochemical analysis for clinical diagnosis[1,2]. However, genomic analysis of DNA extracted from FFPE blocks is problematic, as formalin fixation negatively impacts DNA quality and quantity compared to fresh frozen (FF) material[3,4]. The pathology archive of any large hospital is likely to contain tens of thousands of FFPE blocks. Enabling accurate genomic analysis of FFPE material would unlock tremendous translational research potential from these vast collections of archival material[5-7].

During the fixation step of FFPE preservation, buffered formalin (4% formaldehyde) penetrates the biospecimen and generates cross-links between intracellular macromolecules (DNA–DNA, DNA–RNA and DNA–protein). These cross-links stall DNA polymerases during library amplification[7-9]. As a consequence, the diversity and the number of templates that can be amplified by PCR from FFPE DNA is significantly depleted[4,10]. Furthermore, formalin causes hydrolytic deamination of cytosine bases to uracil[1,7], resulting in U:G mismatches where DNA polymerase incorporates adenine opposite to uracil in amplicon-based protocols, generating artefactual C:G>T:A substitutions in sequencing data[5-7].

To mitigate deamination artefacts, some FFPE sequencing library preparations provide repair treatment whereby uracil DNA glycosylase (UDG) is added to remove uracil bases prior to amplification[5,6,11]. However, formalin-induced deamination of 5-methylcytosine (5mC; exclusively present in CG dinucleotides) would be converted directly to thymine instead of uracil[3,10]. This second class of formalin artefacts is not corrected by the UDG

[1]Organismal and Evolutionary Biology Research Programme, Department of Computer Science, University of Helsinki, Helsinki, Finland. [2]Evolution and Cancer Laboratory, Centre for Genomics and Computational Biology, Barts Cancer Institute, Barts and the London School of Medicine and Dentistry, Queen Mary University of London, London, UK. [3]Genomics and Evolutionary Dynamics Laboratory, Centre for Evolution and Cancer, Institute of Cancer Research, London, UK. [4]Division of Biomedical Informatics, Department of Medicine, University of California San Diego, La Jolla, CA, USA. [5]Institute of Biotechnology, Helsinki Institute for Information Technology, University of Helsinki, Helsinki, Finland. ✉e-mail: trevor.graham@icr.ac.uk; v.mustonen@helsinki.fi

treatment; therefore, downstream bioinformatics approaches are necessary to attempt their removal[7].

Mutational signatures derived from whole-genome sequencing (WGS) data characterise the mutational processes that have acted upon the cancer genome[12,13]. Single base substitution (SBS) signatures are derived by considering the type of specific base pair change (e.g. C>T) together with the flanking base pair context (e.g. ACA>ATA)[12,13]. The recently updated mutational signature catalogue provides a comprehensive source of mutational processes active in human cancers, which is derived from an unprecedentedly large number of samples[14]. Activities of signatures have immediate translational relevance[15-19], for example homologous recombination (HR) deficiency signature (SBS3), which is one of the response indicators to poly (ADP-ribose) polymerase (PARP) inhibitors for targeted therapy[17,20,21].

Mutational signature analysis on FFPE material is problematic because of the artefactual mutations induced by formalin fixation[5-7]. Here, we use the statistical machinery of mutational signature analysis to derive a mutational footprint caused by formalin exposure during FFPE biospecimen processing. First, we identify formalin artefact mutational signatures in both unrepaired (without UDG) and repaired-FFPE (with UDG) samples, using paired FFPE and FF sample sequencing data from the same tissue. We next design and validate a decomposition algorithm, FFPEsig, to subtract FFPE artefacts and thereby infer biological mutation profile in a given FFPE sample. We demonstrate the efficiency of our method on synthetic and sequenced FFPE samples and show that FFPEsig can correctly recover the true activities of mutational signatures otherwise masked by FFPE-induced artefacts. Our method enables robust mutational signature analysis on FFPE samples, thus paving the way towards clinical implementation using FFPE WGS data.

## Results

### Formalin fixation artefacts are predominantly C>T mutations

To identify FFPE artefacts signatures, we used publicly available targeted panel sequencing data from two previous studies[10,11], in which triplicate samples (repaired FFPE, unrepaired FFPE and FF) were available. The study by Prentice et al. (hereafter study 1) comprised $n = 3$ colorectal cancers (CRC), and each cancer included nine samples: one FF sample, four unrepaired and four repaired-FFPE samples that were sequenced after a fixation time of 2, 15, 24 and 48 hours respectively[10,11]. In addition, study 1 included $n = 29$ patients for whom repaired and unrepaired-FFPE samples were available[10,11]. In the study by Bhagwate et al. (hereafter study 2), triplicate samples from $n = 4$ benign breast tissue were available[10,11]. In total, we obtained $n = 110$ FFPE samples, of which 32 (29%) had matched FF (see Methods).

We first focused on samples with matched FF available and examined the set of mutations detected in FFPE samples but not detected in all FF samples and removed mutations listed in germline SNPs databases (termed FFPE-only or discordant mutations; see Methods). Within the study 1 sample set, we discovered that C>T discordant mutations were common (937 out of 1300, ~72% in unrepaired FFPEs; 265 out of 679, ~39% in repaired FFPEs), and T>C mutations were also common (347/1300, ~27% in unrepaired FFPEs; 393/679, ~58% in repaired FFPEs) (Supplementary Fig. 1). In comparison, discordant mutations from study 2 were primarily C>T mutations (10,519/ 10,575, ~99.5% in unrepaired and 724/896, ~81% in repaired FFPEs), with very few T>C mutations detected (30/10,575, ~0.28% in unrepaired and 85/896, ~9% in repaired FFPEs) (Supplementary Fig. 2). Overall, both studies observed large numbers of C>T artefacts and this class of mutation was reduced dramatically in samples repaired with UDG treatment. These observations suggest that C>T artefacts are directly caused by the formalin treatment. This is in agreement with previous studies reporting that deaminated cytosine and 5mC result in C>T artefacts[5-7] (Supplementary Fig. 3).

To examine whether T>C mutations are also true artefacts of FFPE, we counted the proportion of C>T and T>C mutations present in two or more samples both within and between tissues (termed concordant mutations) in the complete mutation list without any filters applied (Supplementary Fig. 4). The complete list may consist of unfiltered SNPs and recurrent PCR/sequencing artefacts, as well as recurrent FFPE artefacts. We assume most of FFPE-induced mutations to be randomly located across the genome and so expect the genomic coordinate of mutations to differ between samples. We observed that an average of ~30% C>T mutations were shared by at least two samples, in contrast to ~88% for T>C mutations (Supplementary Fig. 4a). We next compared the pair-wise concordant mutation ratio across three patients: ~10% of C>T mutations and ~59% of T>C mutations were shared by the sample pairs on average (Supplementary Fig. 4b). We noted that the concordant T>C mutations were similarly prevalent among FF sample pairs in study 1 (~57.8%, 1275/2207), compared to the FFPE sample pairs in this study (~58.4%, 1260/2156). This high prevalence of T>C in FF samples is particularly observed in study 1 but not in general FF CRC samples, so seems to be the consequence of an undefined defined batch effect in this study. Overall, we found that T>C mutations are not randomly distributed, suggesting that they are batch-related artefacts likely due to library preparation or downstream pipeline.

In contrast, C>T mutations showed a substantially lower concordant ratio (Supplementary Fig. 4). We noted that a relatively higher proportion of concordant C>T mutations were observed among repaired-FFPE pairs (Supplementary Fig. 4b). We found that the majority of those concordant mutations were found in NCpG contexts, and so were the concordant mutations from unrepaired-FFPE pairs. This explains the higher concordant ratio in repaired pairs since they have a similar chance of observing concordant mutations at CpG sites (the numerator) but have a much smaller total artefact count (the denominator) compared to the unrepaired pairs (Supplementary Fig. 4b).

We next studied the relationship between FFPE-only mutation count and formalin fixation time. Our assumption was that the number of mutations arising from formalin damage would increase when the exposure (formalin fixation) time was longer. Indeed, we observed a positive correlation for C>T mutations (slope = 7.28, intercept = 150.28 for unrepaired-FFPE; slope = 0.69, intercept = 73 for repaired FFPEs) (Fig. 1a). However, the T>C mutations showed an opposite trend in unrepaired FFPEs (slope = −0.05, intercept = 116.52) (Supplementary Fig. 1a). These results further confirmed that C>T mutations were true formalin-induced artefacts, but T>C mutations were likely associated with the library preparation or other downstream steps.

To validate these observations, we conducted a thorough literature review and identified additional 20 studies looking at formalin-induced artefacts (Supplementary Table 1). All of them (100%) reported C>T mutations due to formalin exposure. Only 3 of 20 (15%) reported additional T>C and 2 of 20 (10%) reported C>A mutations. A similar survey performed by Do and Dobrovic also observed a primarily C>T landscape with non-conclusive evidence regarding other mutation types[7]. Therefore, we conclude that C>T mutations are dominant among FFPE-induced artefacts, whereas occasional T>C or C>A artefacts are batch-related artefacts likely due to PCR related procedures on formalin-treated DNA[22-24].

### Unrepaired-FFPE signature mirrors SBS30; repaired-FFPE signature mirrors SBS1

We next used FFPE-only mutations to learn the mutational signature induced by formalin. We used all FFPE samples from study 1 and 2 ($n = 110$) and excluded all T>C mutations as we had confirmed that these mutations were not formalin associated artefacts. We therefore assigned zeros to T>C channels. The exclusion of this class was equivalent to treating them as missing data. The samples in the respective studies were sequenced using different cancer-gene panels,

thus the mutational opportunities, determined by the frequency of each trinucleotide context in the panel, differed between studies (Supplementary Fig. 5). Therefore, we applied study specific normalisation on the mutation counts to project the values onto human genome context, which enabled direct comparison between the studies.

The cluster of normalised 80-channel mutational profiles (without T>C) from the entire combined set of $n = 110$ FFPE samples was represented using t-distributed stochastic neighbour embedding (t-SNE) method[25] (Fig. 1b). Samples from the two studies showed no batch effect and clearly separated into two clusters of unrepaired and

repaired samples. Two repaired samples from study 1 clustered with unrepaired FFPEs, which we suspect is due to poor response to UDG treatment[26]. By contrast, we also clustered normalised mutational profiles of T>C mutations, which showed large variability among samples regardless of repair treatment status and the study of origin (Fig. 1c).

To derive FFPE signatures, we used t-SNE clustering to select representative samples to exclude possible outliers. We performed an iterative process where each iteration was defined by the random seed inputted to the t-SNE algorithm. For each t-SNE embedding, we calculated the spatial density of the clustered data measured by a

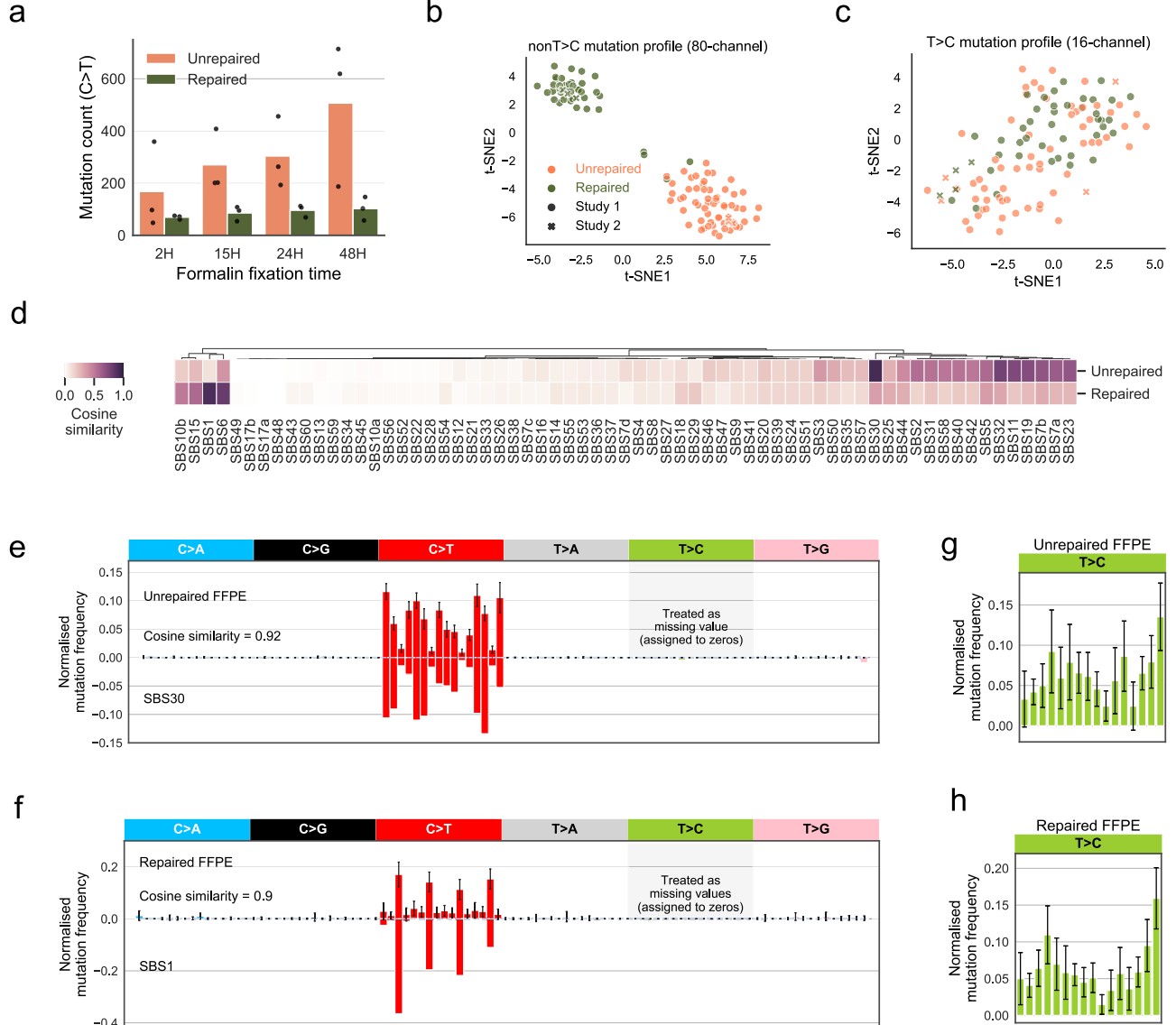

**Fig. 1 | Mutational signatures of formalin exposure. a** C>T FFPE-only mutation count increases with formalin fixation time. We observed this increase in both unrepaired and repaired-FFPE samples from study 1 (the fixation group). FFPE-only mutations refer to mutations that are only discovered in FFPE but not in FF samples or known germline databases. The bar height represents the average C>T count in $n = 3$ patients, and the individual counts are marked as black dots. **b** Consistent and separable mutational patterns observed for unrepaired-FFPE and for repaired-FFPE samples using 80-channel spectrum (non-T>C). We clustered the normalised 80-channel mutation profiles ($n = 110$) from study 1 and 2 using t-SNE (see Methods). **c** No consistent and separable mutational patterns observed for T>C mutations. We clustered the normalised T>C mutation profiles ($n = 110$) from study 1 and 2 using t-SNE. **d** Comparison of our derived FFPE signatures to known COSMIC SBS

signatures. **e, f** Unrepaired signature is highly similar to SBS30 (**e**) and repaired signature is highly similar to SBS1 (**f**). We treated T>C features as missing data due to the strong batch-effect found in study 1, which is also observed in a few other studies shown in Supplementary Table 1 and therefore they were assigned to zeros. We noted that zero values are approximately close to the true T>C mutation probabilities in FFPE datasets without this batch-effect (Supplementary Fig. 6f). Error bars indicate the standard deviation in $n = 55$ independent samples with top 50% density in t-SNE cluster (see Methods). **g, h** Large variability in T>C mutation channels. We derived the T>C patterns using the same methods applied in (**e, f**). The error bar showed the standard deviations in $n = 55$ independent samples with top 50% density within the t-SNE (see Methods).

gaussian kernel, and selected samples in regions of high spatial density (top 50%) as our representative sample subset (Supplementary Fig. 6a). The averaged values of all mutation channels from this representative subset generated one set of candidate formalin signatures. Our final formalin signatures were derived from the mean of all candidates collected from 100 random t-SNE embeddings (Supplementary Fig. 6b, c, Supplementary Data 1).

We compared the derived FFPE-artefact signatures to COSMIC SBS signatures (V3, May 2019 version)[14] (Fig. 1d). We found that unrepaired-FFPE signature is highly similar to SBS30 (cosine similarity = 0.92; Fig. 1e), and the repaired-FFPE signature shares a similar pattern with the ageing signature−SBS1 (cosine similarity = 0.90; Fig. 1f). We noted that the features of both FFPE signatures agreed with the expected effects of formalin-induced artefacts with or without UDG treatment (Supplementary Fig. 3). SBS30 has been validated as a mutational footprint of *NTHL1* mutations that disrupt base excision repair (BER)[27,28]. SBS1 is well-known as a clock-like signature that positively correlates with age[29]. We also note that the repaired-FFPE signature shares even greater similarity (0.95) with clock-like signature 1 released in COSMIC V2 (March 2015 version), which was derived from a relatively smaller cohort.

Despite the high similarity, we examined the C>T mutation channels where the fold-change was >2 between the signature pairs in Fig. 1e, f. The unrepaired-FFPE signature only differs in NCT context from SBS30 (Fig. 1e). The repaired-FFPE signature mostly differs in non-CpG mutation contexts which are absent in SBS1 (Fig. 1f), and the few mutations in those channels likely remained unrepaired even after UDG treatment. However, we noted that the above dissimilarities might not suffice to make an easy distinction between the two sets of signature pairs by decomposition algorithms.

We attempted to apply the same method to derive patterns for T>C mutation channels (Fig. 1g, h) but did not observe distinguishable patterns between repaired or unrepaired-FFPE samples. We found the variability of the T>C mutation channel was much bigger than that of the C>T channel (Fig. 1e, f), confirming that T>C mutations are not driven by the same underlying process. Overall, the inconsistent error pattern for these batch-related T>C mutations found in study 1 ruled out the possibility of extracting them using a similar strategy as a noise signature, in contrast to formalin-induced C>T mutations. Separate experimental studies are required to investigate the molecular cause of these spurious T>C mutations.

To validate the robustness of our derived formalin signatures, we: (1) performed independent signature inference from study 1 and 2, which revealed almost identical patterns (cosine similarity ≥ 0.98; Supplementary Fig. 6d, e); and 2) derived a separate repair signature from a third study[30] (Supplementary Fig. 6f). This third study reported strictly filtered somatic mutations for $n = 11$ lung adenocarcinoma samples[30]. We found a total of 1041 FFPE-only mutations and the aggregate profile of these mutations is highly similar to the repaired-FFPE signature we derived from study 1 and 2 (cosine similarity = 0.93, Fig. 1f), implying the FFPE samples in study 3 were repaired using UDG. These results together confirmed our discovered FFPE signatures are highly robust across samples collected by different laboratories and/or processed via different pipelines.

We noted that study 3 contains ~7.5% (79 out of 1041) T>C mutations, similar to study 2 repaired-FFPE samples, which strongly supports our earlier conclusion that the T>C artefacts discovered in study 1 were batch-related and they should not be taken into account when investigating the direct effect of formalin on the genome.

## Accurate correction by FFPEsig on majority of simulated FFPE samples

We next designed and implemented an algorithm called FFPEsig to correct artefacts from FFPE mutation profiles (see Methods). The algorithm decomposes the observed aggregate mutational catalogue in a given FFPE sample into FFPE-artefact (the noise) and true biological mutations (the signal). To test the performance of FFPEsig, we added the same amount of FFPE artefacts ($10^4$) to all PCAWG fresh tumour profiles[14,31] in silico (Supplementary Fig. 7), and then attempted to remove them using FFPEsig. We omitted T>C mutations in our simulations to match real-life samples where it would be unknown whether a FFPE sample contains batch-related T>C artefacts.

The noise count of $10^4$ used in our simulation set-up yielded a signal-to-noise ratio (SNR) of 1/17 on average, which falls in the range of excess mutations observed in real FFPE samples after applying variant filters[7,32,33]. In Fig. 2a, we show an example of one simulated CRC FFPE sample together with its biological and FFPEsig corrected profiles. In this case, FFPEsig successfully inferred the biological mutation catalogue with ~0.98 accuracy (measured by cosine similarity between the true and corrected profiles on C>T channels). We noted the overall correction accuracy was slightly higher when we used the full channel profile (Supplementary Fig. 8). Going forward, we used the stricter evaluation focusing only on C>T mutation channels.

Next, we refitted signature activities using the three profiles shown in Fig. 2a. We assigned activities to the three signatures (SBS1, 5 and 18) that were identified by the PCAWG team using the true biological (fresh frozen) profile. The corrected FFPE profile produced highly similar contributions inferred from the real biological profile (Fig. 2b). However, when using the uncorrected FFPE profile, the contributions of SBS1 and SBS18 were greatly underestimated. Because relative signature activities were inferred, these errors propagated and caused incorrect inference of all signatures, most notably of SBS5.

Next, we applied FFPEsig to all PCAWG samples ($n = 2780$) with $10^4$ mutations of in silico FFPE noise added to their true mutation profiles. Overall, FFPEsig achieved ~0.90 and ~0.87 correction accuracy on average for unrepaired-FFPE and repaired-FFPE samples, respectively (Fig. 2c, d). We classified cases with correction accuracy >0.90 as well-corrected samples (Fig. 2d), which yielded $n = 1770$ (~64%) unrepaired and $n = 1462$ (~53%) repaired well-corrected samples.

To examine the possible factors which could influence the correction accuracy, we evaluated: (1) SNR, measured by biological C>T mutation count (the signal) divided by the introduced noise count; and (2) the signal-to-noise similarity (SNS), measured by cosine similarity of C>T channel between the true biological catalogue (the signal) and the FFPE-signature. We found that poorly corrected cases were due to low SNR and/or high SNS (Fig. 2c). In particular, samples with low SNR (equivalent to having low C>T biological mutation load) were difficult to correct regardless of the SNS (Fig. 2c). We further separated these two factors and confirmed that correction performance improved with increasing SNR (Fig. 2e), and with smaller SNS (Fig. 2f).

We continued our in silico evaluation by examining FFPEsig performance on our synthetic FFPE samples but separated into cancer types (Fig. 2d). The efficacy of correction varied significantly across 26 main cancer types (with $n > 20$ per cancer type). Overall, FFPEsig was most accurate in skin melanoma (accuracy = ~0.98) due to its higher mutation load (mean = ~96,361) and lower SNS (mean = ~0.55), followed by bladder transitional cell carcinoma (Bladder-TCC, accuracy = ~0.96) and lung squamous cell carcinoma (Lung-SCC, accuracy = ~0.96). In contrast, FFPEsig performed poorly in pilocytic astrocytoma (CNS-PiloAstro, accuracy = ~0.60), medulloblastoma (CNS-Medullo, accuracy = ~0.79) and myeloproliferative neoplasm (Myeloid-MPN, accuracy = ~0.80), because of the low SNR (≤0.06) in this cancer types.

We noted that FFPEsig had different performance between unrepaired and repaired FFPEs within certain cancer types (Fig. 2d). There were 17 out of 26 cancer types with detectable difference ($P < 0.05$) and 14 of 17 with significant difference ($P < 0.001$). For instance, the noise correction worked significantly better for unrepaired-FFPE samples in pancreatic adenocarcinoma (Panc-AdenoCA) with 94% of well-corrected samples, in contrast to only 43% for

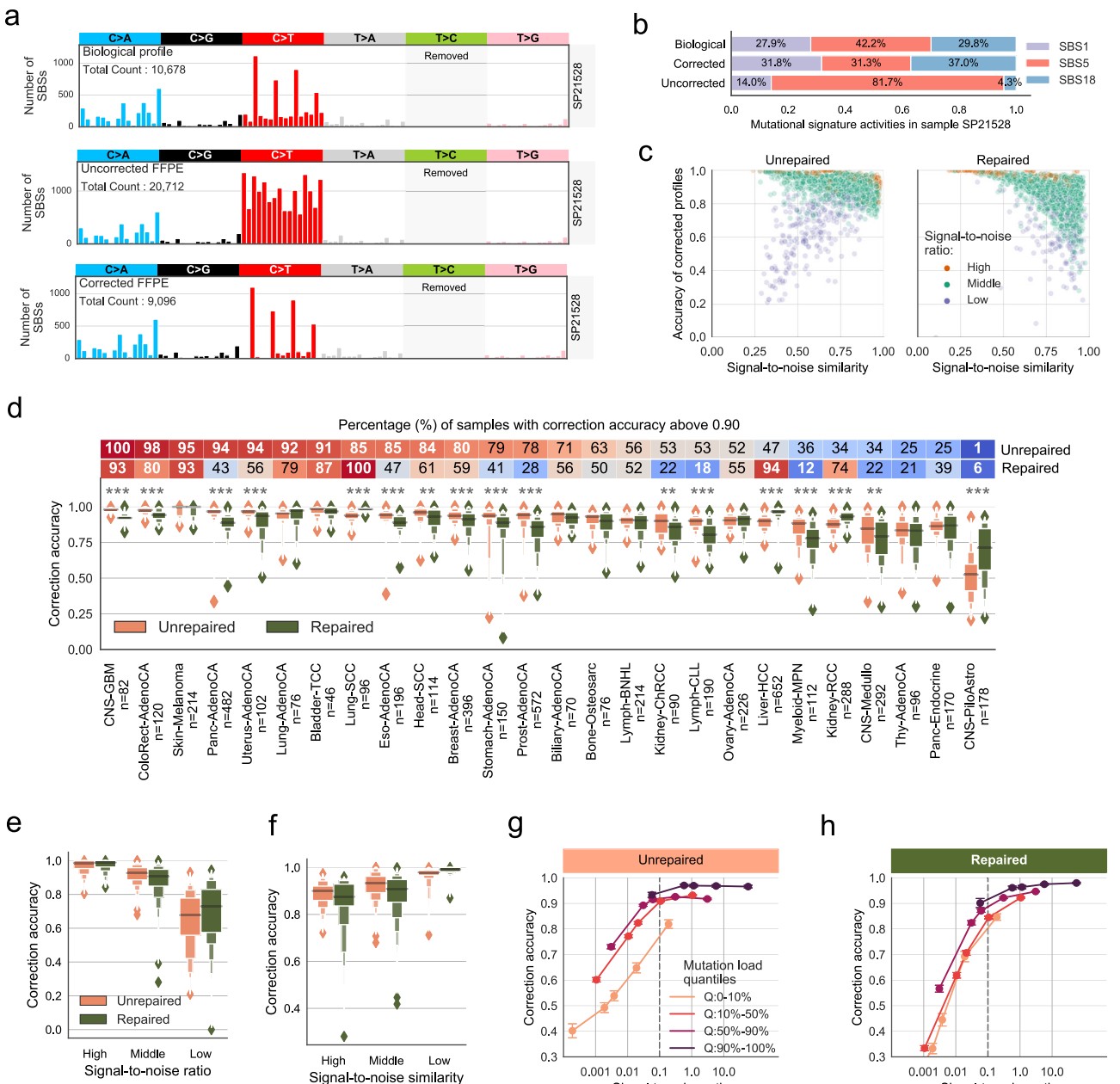

**Fig. 2 | Correction of FFPE artefacts in synthetic FFPE samples using FFPEsig.**
**a** The correction result for one CRC sample as an example. **b** Similar activities resulted from the corrected profile compared to the true biological profile for the CRC sample in **a**. **c** Correction accuracy for all synthetic FFPE samples. We grouped $n = 2780$ samples into three categories according to biological C>T mutation count: high (top 10%, $n = 278$), low (bottom 10%, $n = 278$) and middle (the remaining 80%, $n = 2224$). **d** Correction accuracy varied among cancer types. The percentage of samples with accuracy >0.90 is annotated in the heatmap bar. Data are presented using a Letter-Value plot and the black line corresponds to the median of the dataset and every further step splits the remaining data into two halves (the same for **e**, **f** below). The statistical difference between repaired versus unrepaired FFPEs is derived from the two-sided Mann–Whitney $U$ test. $P \le 0.001$ (***); $P \le 0.01$ (**); $P \le 0.05$ (*). **e** Positive correlation between signal-to-noise ratio and correction

accuracy. We classified all samples based on SNR into three groups: high (top 10%, $n = 278$), low ($n = 278$, bottom 10%) and middle (the remaining 80%, $n = 2224$). **f** Negative correlation between signal-to-noise similarity and correction accuracy. Unrepaired: $n = 278$ (high), $n = 2023$ (middle) and $n = 201$ (low) samples. Repaired: $n = 244$ (high), $n = 1984$ (middle) and $n = 274$ (low) samples. **g, h** FFPEsig works well in samples with SNR above 0.1 for both unrepaired (**g**) and repaired (**h**) FFPEs. We generated five sets ($n = 2780$ per set) synthetic samples by adding increasing noise ($10^3$, $10^4$, $5 \times 10^4$, $10^5$ to $10^6$) to PCAWG samples. We divided samples in each set into four categories depending on biological C>T mutation load (from the lowest to the highest): Q:0–10% ($n = 278$), Q:10–50% ($n = 1112$), Q:50–90% ($n = 1112$) and Q:90–100% ($n = 278$). Data are presented as mean values within each category ± 95% confidence interval. All statistics are derived from biological independent samples.

repaired FFPEs. This difference was caused by the low SNS (0.58) in unrepaired FFPEs, which is much higher (0.88) in repaired FFPEs. Furthermore, FFPEsig worked successfully in only repaired FFPEs for Lung-SCC and liver hepatocellular carcinoma (Liver-HCC), with 100% and 94% well-corrected samples, respectively, for the opposite reason.

We next evaluated how FFPEsig performance depends on SNR by generating more synthetic samples with five increasing levels of FFPE noise to PCAWG fresh tumour profiles (Fig. 2g, h). We then split all samples at each noise level into four categories based on an increasing order of true biological mutation counts, in order to separately analyse hyper- and hypo- mutated cancers. Overall, FFPEsig performed well in

most samples when SNR > 0.1, achieving mean accuracy of ~0.93 for both unrepaired and repaired FFPEs. However, its performance dropped quickly when the SNR was smaller than 0.1, and this limitation is understandable since the stochastic variability of the noise substantially outweighed the signal. We noted that the correction algorithm had high accuracy on hyper-mutated samples (Q:90–100%) regardless of noise levels but could not successfully correct hypo-mutated samples (Q:0–10%) in most cases (due to the low SNR). Therefore, FFPEsig correction should be applied with caution in samples where prior knowledge suggests very low true mutation load.

## Uncorrected FFPE profiles lead to mis-assignment of signature activities that can be rectified by FFPEsig

Next, we systematically evaluated the impact of FFPE artefacts on signature decomposition (Fig. 3; Supplementary Figs. 9, 10). We refitted signature activities in uncorrected (with simulated FFPE artefacts) and corrected (by FFPEsig) profiles in samples with accuracy >0.90 shown in Fig. 2c. We compared the derived signature activity weights against the values inferred from real biological mutation profiles from the same tumour (Fig. 3). We limited refitting to the signatures identified in each sample by the PCAWG team under the assumption that the PCAWG-identified signatures are the true mutational processes active in those tumour samples. All active signatures were adjusted to 80-channel spectrum. It yielded 7359 signatures in $n = 1770$ well-corrected unrepaired FFPEs and 6057 signatures in $n = 1462$ well-corrected repaired FFPEs (the 1st panel in Fig. 3a, b).

We quantified the false positive (FP) and false negative (FN) rates on the inferred activities (Supplementary Fig. 9). We assigned a binary label to each activity inference, in which 0 means absent (relative contribution ≤0.1) and 1 means present (relative contribution >0.1). The binary classifications based on corrected profiles are highly reliable with 8-10% false discovery rate (Supplementary Fig. 9a, b), and the majority of the mis-assignment happens to a flat signature—SBS5, which is problematic to recover in signature refitting analysis in general[14].

In contrast, using uncorrected profiles has a profound impact while determining the present/absent status of the signatures, with a total error rate of ~41% and ~37% for unrepaired and repaired-FFPE samples, respectively (Supplementary Fig. 9c, d). The recall rate is particularly low in unrepaired-FFPE samples (49%), which shows about half of true signatures would not be detected (predicted as absent) if the uncorrected profiles are used. We observed a relatively lower recall rate of ~23% using uncorrected profiles for repaired FFPEs. We assume the generally active SBS1 in cancer genomes buffered the impact in repaired FFPEs.

To quantify the relative contribution changes, we next measured the error of inferred signature activity using the absolute difference (between the inferred and the true signature contributions, termed $\varepsilon$). Overall, the refitted contributions from corrected FFPE profiles recovered true activities with negligible error ($\varepsilon = -0.06$ for all FFPE samples) (the 1st panel of Fig. 3a, b). On the other hand, signature proportions derived from uncorrected FFPE profiles were grossly misestimated compared to true values ($\varepsilon = -0.20$ for unrepaired; $\varepsilon = -0.18$ for repaired FFPEs), highlighting the necessity for artefact correction prior to mutation signature fitting.

We next studied the impact of FFPE artefacts on the individual signatures which were being detected in at least 20 tumour samples in the PCAWG cohort (Fig. 3c, d, Supplementary Fig. 10). Again, using uncorrected FFPE profiles in signature analysis led to significant mis-assignment for many signatures (Fig. 3). For example, SBS3 (signature of HR deficiency) was largely underestimated in both unrepaired FFPEs ($\varepsilon = -0.34$) and repaired FFPEs ($\varepsilon = -0.31$). The activity of BER signatures was grossly overestimated in unrepaired FFPEs ($\varepsilon = -0.36$). Similarly, the overall ageing signature (SBS1) was commonly overestimated in repaired samples where the refitting method likely assigned FFPE artefacts to SBS1 ($\varepsilon = -0.25$), whereas their activities were systematically

underestimated in unrepaired FFPEs (Supplementary Fig. 10a). In addition, signatures without known clear aetiology (e.g. SBS8, 9, 12, 16) were also greatly mis-assigned using uncorrected FFPE profiles in both repaired and/or unrepaired FFPEs (Supplementary Fig. 10).

In contrast, the application of FFPEsig enabled reliable and robust prediction of signature contributions for all signature groups using corrected profiles (Fig. 3c, d). Errors of activities inferred from corrected profiles were significantly lower than those from uncorrected profiles among 20/21 and 19/21 signature groups for unrepaired and repaired FFPEs, respectively. We did not detect such a significant decrease of error in two sets of signatures, namely POLE-signature (in unrepaired and repaired FFPEs) and MSI-signature (in repaired FFPEs). We propose this is because: 1) the POLE/MSI signatures were not characterised in C>T mutation channels; and 2) the FFPE artefacts we added in the simulation have minor impact on hyper-mutation samples such as POLE/MSI (Fig. 2g, h).

We also compared the reconstruction accuracy of corrected and uncorrected profiles against the real biological catalogues using cosine similarity (Fig. 3e, f). The reconstructed mutational profiles were calculated as the product of active signatures and their inferred activities. We calculated the reconstruction accuracy from the biological profiles (mean = ~0.97) as a baseline comparison. This reconstruction accuracy was followed very closely by that of corrected FFPEs (~0.97 for unrepaired, ~0.96 for repaired FFPEs). However, uncorrected FFPEs only achieved ~0.86 and ~0.77 averaged reconstruction accuracy for unrepaired and repaired-FFPE, respectively, which is significantly lower than that of corrected FFPE samples.

## A case study of correcting FFPE artefacts in real FFPE CRCs shows consistent results with synthetic samples

Next, we performed whole-genome sequencing on two tumour FFPE samples (unrepaired versus repaired), and on the matched normal tissue DNA from the same CRC patient (see Methods; paired FF was not available). The mean coverages of the sequencing data were 46× (unrepaired), 43× (repaired) and 43× (the matched normal), with >98.8% reads mapped to the genome. Following our filtering, we detected 13,208 and 6107 somatic SBS in unrepaired and repaired-FFPE, respectively (Supplementary Fig. 11a).

In particular, the two types of dominant mutations in our FFPE samples were C>T and T>C, and together they contributed 64.7%-66.6% to the total mutations (Supplementary Fig. 11b, Supplementary Data 2). For C>T mutations, we expected them to be a mixture of FFPE artefacts and real biological mutations, because of the relative abundance (~35%) of C>T mutations in PCAWG CRCs. T>C mutations accounted for 41.2% (5469/13,280) and 39.8% (2431/6107) in our unrepaired and repaired FFPEs, but only ~16% on average in PCAWG fresh CRCs (Supplementary Fig. 11c). Given that our earlier analysis pointed to batch-related T>C mutations possibly caused by FFPE DNA sequencing protocols or downstream steps (Fig. 1c, Supplementary Table 1), we excluded the T>C mutations from our analysis. We also found the overall mutation load in the repaired WGS-CRC was lower than in the unrepaired sample (Supplementary Fig. 11b). It is partially due to the UDG treatment (leading to a decreased C>T load) and our mutation calling and filtering methods that discard low allele-frequency (low coverage) mutations from all channels.

Since paired FF was not available, to provide the ground truth mutational signature, we were inspired by results found in study 2[10], where both repaired and unrepaired-FFPE samples contained the majority of the variants found in the paired FF sample. Thus, we used concordant mutations between the repaired and unrepaired samples with more strict filtering (supporting reads ≥5 in both FFPEs) as an approximation for the true biological mutation profile of the tumour: this yielded a total of 1040 concordant true somatic mutations (Supplementary Fig. 11a, b), and 656 of them remained after excluding T>C mutations (top panel of Fig. 4a).

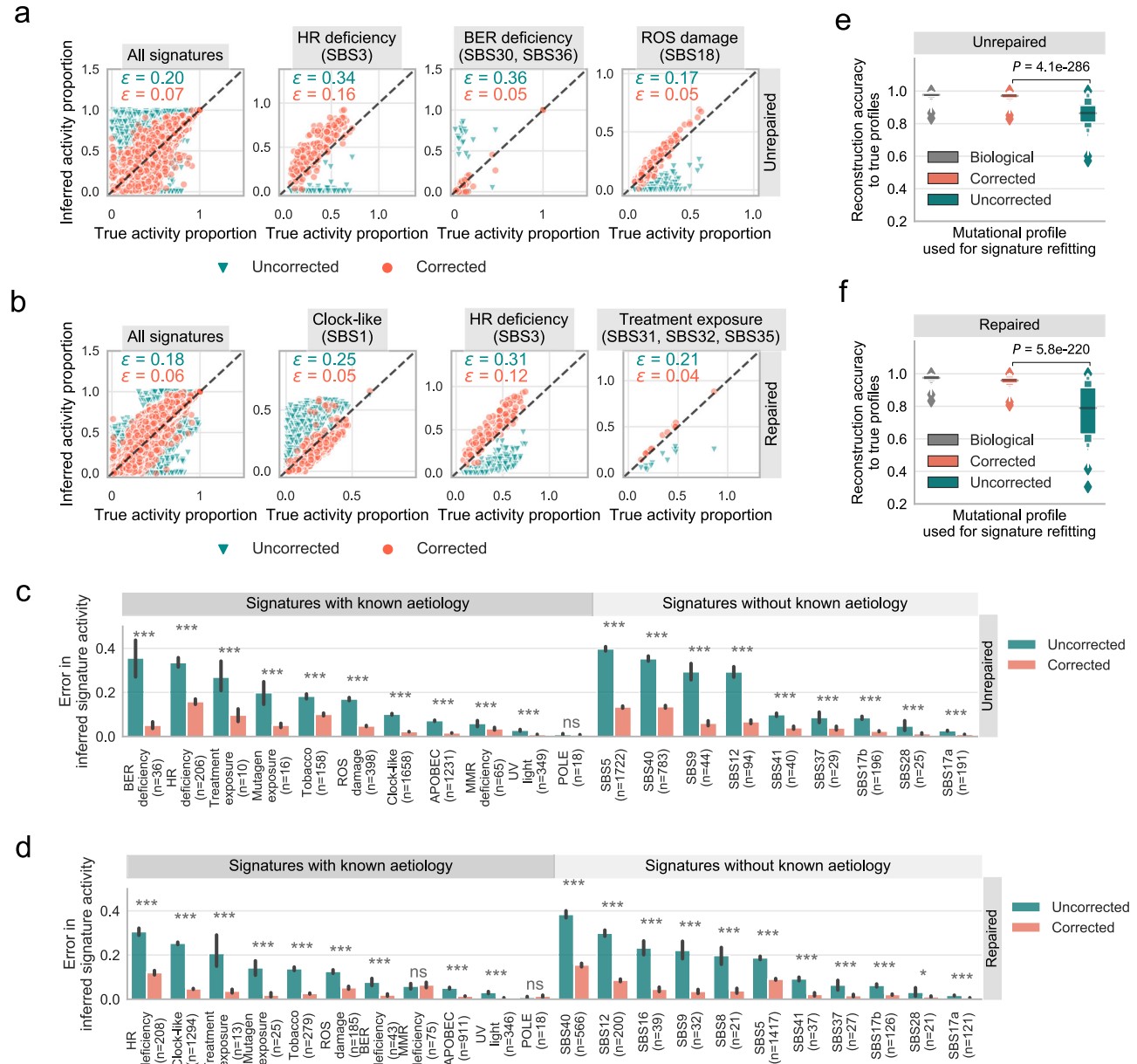

**Fig. 3 | Mis-assigned signature activities from uncorrected FFPE profiles can be ratified by FFPEsig. a, b** Similar signature activities inferred from corrected profiles and the true values. We compared the signature activities inferred from corrected and uncorrected profiles against true activity proportions from biological profiles. We used $n = 1770$ well-corrected unrepaired FFPEs (**a**) and $n = 1462$ well-corrected repaired FFPEs (**b**). We used adjusted 80-channel signatures (non-T>C; see Method). Activity error ($\varepsilon$) is calculated as the absolute difference between the true and inferred relative activities. Mean of the error is annotated in the upper left within each plot. We also included a few signatures with large errors while using the uncorrected profiles, including HR (homologous recombination) deficiency, BER (base excision repair) deficiency, ROS (reactive oxygen species) damage, clock-like, and treatment exposure. **c, d** Error of inferred signature activities is significantly higher using uncorrected profiles in unrepaired (**c**) and repaired (**d**) FFPE samples.

Data are presented as mean values ± 95% confidence interval (error bar). We grouped the signatures based on their aetiology information. One-sided Wilcoxon signed-rank test was used to calculate the $P$ values: $P \leq 0.001$ (***); $P \leq 0.01$ (**); $P \leq 0.05$ (*); $P > 0.05$ (ns: not significant). **e, f** Reconstruction accuracy of corrected profiles is significantly improved compared to uncorrected profiles for $n = 1770$ unrepaired (**e**) and $n = 1462$ repaired (**f**) FFPEs. All samples are simulated from true biologically independent tumours (Supplementary Fig. 7). The reconstruction accuracy is calculated according to the true biological profiles. Data are presented using a Letter-Value plot and the black line in the middle box corresponds to the median of the dataset. Every further step splits the remaining data further into two halves. Significant difference of reconstruction accuracy between corrected and uncorrected profiles is tested using the two-sided Wilcoxon signed-rank test.

To explore shared biological mutation patterns in a larger CRC cohort, we performed hierarchical clustering on the $n = 60$ PCAWG fresh CRCs. Indeed, we discovered the subtype CRC samples share highly homologous mutational patterns with average cosine similarity of 0.90 for MSS-CRCs sample pairs, 0.92 for MSI-CRCs and 0.96 for POLE-CRCs (Supplementary Figs. 12a, 13a). To further explore the most conserved mutation patterns within each subtype, we performed a similar similarity analysis on six mutation types separately, which

showed C>A and C>T mutations have the strongest power in classifying CRC subtypes (Supplementary Figs. 12b, 13b). We therefore compared the concordant C>A somatic mutation pattern observed in our two FFPE CRCs to the PCAWG CRCs and identified that our sample was an MSS-CRC (Fig. 4b).

We then applied FFPEsig on the observed mutation profiles from the two FFPE-CRC samples (Fig. 4a, Supplementary Fig. 14). We evaluated the corrected profiles by comparing them to concordant

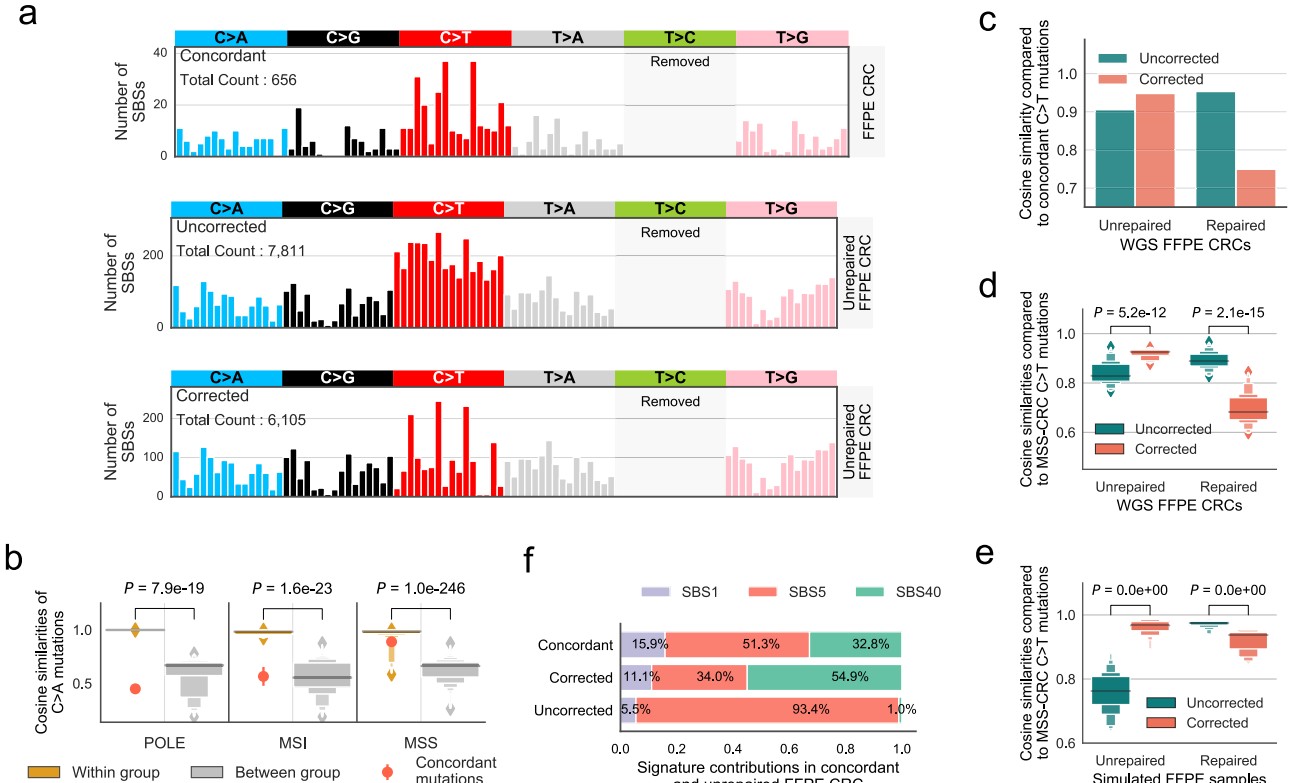

**Fig. 4 | Applying FFPEsig on two real CRC FFPE samples. a** Corrected mutational profile of unrepaired-FFPE CRC tumour is highly similar to concordant (true) somatic mutation pattern. CRC: colorectal cancer. We removed T>C mutations for clear visualisation of other channels (otherwise will be masked), and their full 96-channel profiles are shown in Supplementary Fig. 11d–f. **b** The tumour is an MSS CRC based on the concordant mutation profile. We calculated cosine similarities of unique sample pairs within and between three subgroups (POLE, MSI and MSS) on the most conserved C>A mutations (Supplementary Fig. 13b). There are $n = 28$ (POLE-POLE), $n = 36$ (MSI-MSI), and $n = 903$ (MSS-MSS) independent sample pairs within each subgroup, and $n = 416$ (POLE-MSS/MSI), $n = 459$ (MSI-POLE/MSS) and $n = 731$ (MSS-MSI/POLE) sample pairs between the subgroups. POLE: polymerase epsilon mutated, MSS: microsatellite stable, MSI: microsatellite instability. Data are presented using a Letter-Value plot and the black line corresponds to the median of the dataset, and every further step splits the remaining data into two halves (the

same for **d**, **e** below). The *P* values were derived from the two-sided Mann–Whitney *U* test. **c**, **d** FFPEsig correction works well for unrepaired but not for repaired FFPE according to concordant mutations (**c**) and to PCAWG MSS-CRCs (**d**). We calculated the cosine similarity of corrected and uncorrected FFPE profiles to $n = 43$ independent MSS-CRC samples in **d**. The *P* values were derived from the two-sided Mann–Whitney *U* test. **e** The correction also works well for unrepaired but not for repaired synthetic MSS-CRC FFPE samples. We repeated our analysis in (**d**) on simulated FFPE samples. Therefore, we compared each simulated sample to all MSS-CRC profiles but their real biological profile to match with the scenario that the two real CRC tumours have no paired FF sample. In total, we obtained $n = 1806$ independent data points for repaired and unrepaired samples each. The *P* values were derived from the two-sided Mann–Whitney *U* test. **f** Similar activity fits observed between concordant and corrected (but not uncorrected) unrepaired-FFPE profiles. Source data are provided as in a Source Data file.

mutation catalogue as well as to all PCAWG MSS-CRC samples under the assumption that corrected profiles of our samples should show higher similarity to both positive controls compared to uncorrected profiles. For unrepaired-FFPE CRC, the accuracy improved from ~0.91 before correction to ~0.95 after correction to concordant mutations (Fig. 4c). Furthermore, when compared to PCAWG MSS-CRCs, FFPEsig correction led to a significant increase of cosine similarity from ~0.84 to ~0.92 (Fig. 4d). However, the correction on repaired-FFPE CRC showed the opposite results. We further validated our observations using simulated FFPE MSS-CRCs and confirmed that the correction was only beneficial for unrepaired but not repaired FFPEs (Fig. 4e). This is because the biological MSS-CRC profiles are highly similar to the repaired-FFPE signature (~0.98 on C>T channels) and therefore the algorithm could not distinguish true mutations from artefacts. In this case, the FFPE artefacts would be misread as the ageing signature.

We evaluated the impact of correction by FFPEsig on signature decomposition in our unrepaired FFPE CRC (Fig. 4f). We refitted activities to the three signatures (SBS1, 5 and 40), which are reported as being active in one PCAWG MSS-CRC sample which has the highest cosine similarity to the concordant mutation profile (using C>A channel). We again found that using the uncorrected FFPE-CRC profile led to misinterpretation of the underlying mutational processes (Fig. 4f).

We further investigated how our corrected profile from unrepaired-FFPE could contribute to CRC subtyping. Application of MSIsensor[34] detected 8.3% of microsatellite sites with somatic changes in the unrepaired CRC, but only 0.23% from the repaired CRC. 8.3% exceeds the 3.5% threshold to call MSI[34], and so application of MSI-sensor to an unrepaired CRC sample could lead to miscalling of MSI status for the tumour. We therefore attempted to classify the sample using the conserved mutation patterns (C>A and/or C>T) into CRC subtypes. We were unable to obtain the correct subtype using the uncorrected FFPE profile (Supplementary Fig. 15a, b). However, following corrections using FFPEsig, we could clearly distinguish that the sample was MSS. In addition, we found that the C>A mutation pattern itself could also classify our sample, and this pattern was only marginally affected by artefact correction as FFPE artefacts are found mostly in C>T channels (Supplementary Fig. 15c).

**The potential of using 80-channel signatures for refitting analysis in FFPE samples**

In our previous analyses we omitted T>C mutations due to their unclear origin. We next performed a detailed analysis of how signature decomposition is affected by such removal of T>C mutations. We compared the attributed mutation count (or activity) of each signature

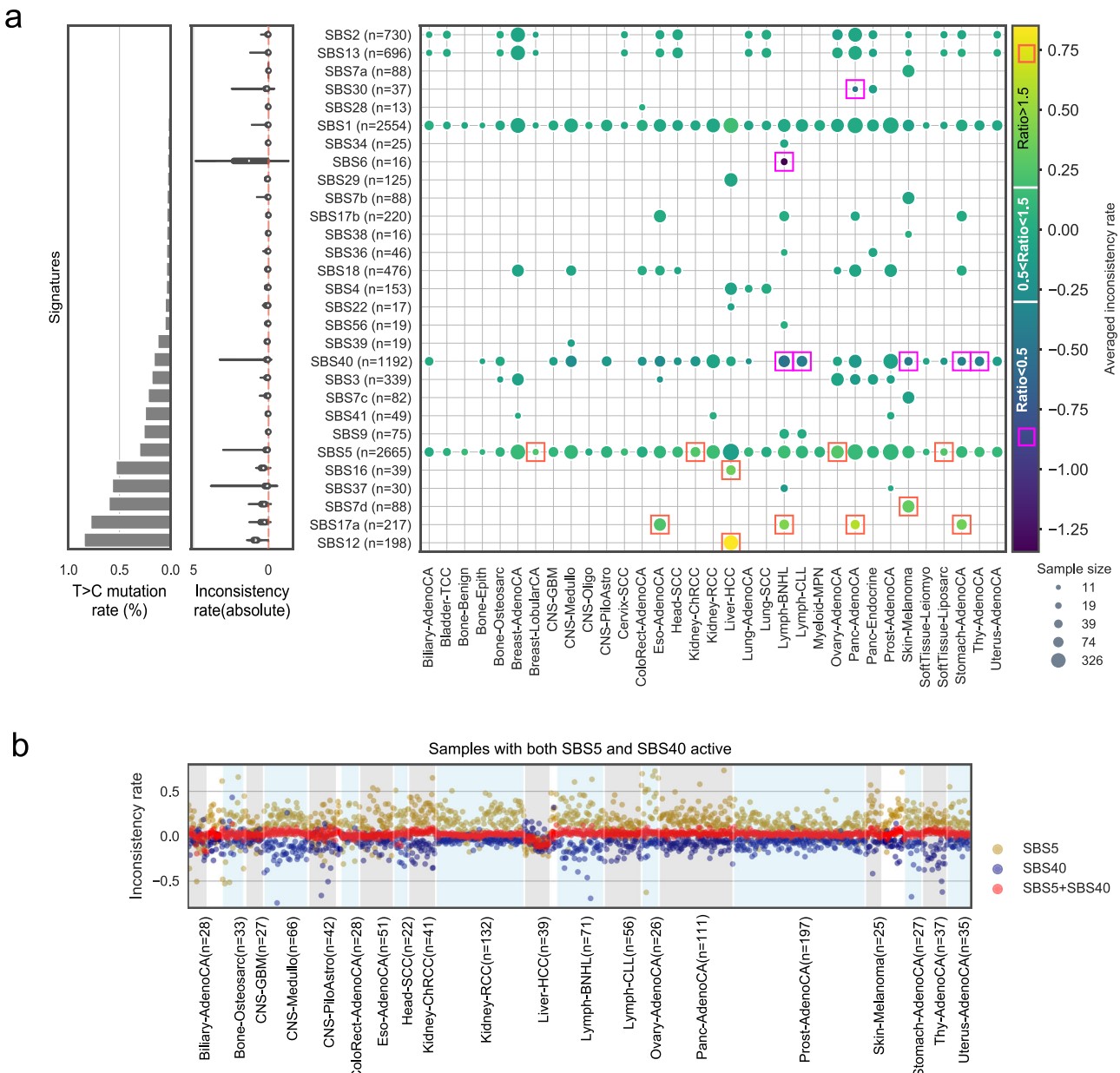

**Fig. 5 | Using adjusted 80-channel signature has minimal effect on activity inference. a** Similar activities inferred by 96-channel (96c) and 80-channel (80c) signatures in most PCAWG samples. We show the signatures that are active in at least 20 cancer samples. We use an inconsistency rate to measure the dissimilarity of the inferred activities, which is calculated using log₁₀(activity_80c/activity_96c). The three panels in **a** share the same y-axis and the labels are shown in the right panel. Left panel: sum of mutational probabilities of T>C channels of the signatures. Middle panel: violin plot of absolute inconsistency rate for each signature. The white dot represents the median value. The thick grey bar in the centre represents the interquartile range and the thin grey line represents the rest of the distribution.

Right panel: heatmap of mean inconsistency rate for all signatures in different cancer types. Orange rectangle marks signatures with the average activity ratio (activity_80c/activity_96c) above 1.5, which indicates that 80c activity is larger than 1.5 times of 96c activity. The purple rectangle marks the averaged activity ratio below 0.5, which indicates 80c activity is smaller than 50% of 96c activity. The radius of each circle represents the sample size (in log scale). **b** The challenge of assigning activities between two flat and similar signatures (SBS5 and SBS40). The light-blue and light-grey shading areas are used for annotating main cancer types (n > 20 sample per type). The cancer types with n < 20 samples are shown in unshaded areas.

by supplying our refitting model with 80-channel (80c; T>C removed) and with 96-channel (96c) signatures on PCAWG mutational catalogues (see Methods; Supplementary Fig. 16). The log₁₀ signature activity ratio of 80c to 96c was used to estimate how dissimilar their activity values were (termed as inconsistency rate). The bigger the absolute inconsistency rate is, the more different the activities are.

We refitted 10,312 mutational signature activities for 29 active signatures from 2726 PCAWG genomes (Fig. 5a), and an additional 54 genomes were excluded due to either low reconstruction accuracy (<0.85; n = 35) by 96c signatures or too small of a sample size (<10

cases per signature per cancer type; n = 19). The mean inconsistency rate among 10,312 refits was 0.013 (Fig. 5a). We considered signatures with inconsistency rate between −0.30 and 0.18, equivalent to actual activity ratio from 0.5 to 1.5, as having well-refitted results. Of the originally inferred 10,312 signature activities, 8938 (86.7%) were well-refitted when only 80c spectrum was used. Overall, 24 of 29 signatures were considered well-refitted.

For the five signatures that were poorly refitted using 80c, four of them had high T>C mutation rates, namely SBS7d, 12, 16 and 17a (Fig. 5a). The inconsistency rate was significantly correlated with T>C

mutation rate of signatures ($r = 0.54$, $P < 1e−16$, Spearman correlation). We grouped the refitted data based on cancer types and found that the majority of the above five signatures with inconsistent refits were each only reported in one cancer type, except for SBS17a which was present in four cancer types. SBS6 also had a high inconsistency rate and was mostly detected in non-Hodgkin lymphoma (lymph-BNHL), likely due to relatively higher similarity shared with SBS1 (0.77). Taken together, removing T>C mutations had a very minor impact on refitting analysis for the majority of the cases (-86.7%).

In addition, SBS5 and SBS40 showed noticeable differences between 96c and 80c fits in several cancer types. Since these two flat signatures are highly similar (cosine similarity is 0.83 using 96c, and is 0.86 using 80c), we expect the general decomposition algorithm to have problems distinguishing them even using 96c signatures. Thus, we hypothesised that while the inferred signature activity of SBS5 or SBS40 might be varied within a sample using 96c or 80c signatures, the sum of the activities of the two signatures would be fairly constant.

We tested our hypothesis on samples with both signatures active (Fig. 5b). As expected, the sum of activities converged well with the mean inconsistency rate of 0.02, but individual attribution for SBS5 was higher by 80c (mean inconsistent rate of 0.15) and lower for SBS40 (mean inconsistency rate of −0.19), and the two individual attributions were negatively correlated ($r = −0.69$, $P = 6.22e−164$, Spearman correlation). We note that these two signatures are assigned less accurately also when using 96-channels, as more data is needed to separate them into more distinguishable patterns.

Finally, we examined signatures where removal of the T>C mutations was most likely to be detrimental for signature identification. We compared all possible signature pairs among 65 COSMIC V3 SBS signatures (Supplementary Fig. 17). As expected, the overall similarities between any two signatures tended to increase (Supplementary Fig. 17a, b). Five signature pairs became highly similar (>0.80) using 80c. Three out of them are reported to be biological/non-artificial mutation processes, namely SBS3-SBS5, SBS40-SBS12 and SBS40-SBS16 (Supplementary Fig. 17c). However, two signature pairs became even more distinguishable using 80c. Therefore, we concluded that reducing to 80-channel signatures by removal of T>C channels tended to have a minor effect on signature refitting.

### Recommended workflow for applying FFPEsig

Finally, we designed a flowchart that summarised how our above findings could be applied to optimise signature analysis for real FFPE samples (Supplementary Fig. S18). The aim of the flowchart is to guide the complete experimental workflow, both in the wet-lab and in the downstream bioinformatics analysis. Firstly, we suggest that users leverage prior knowledge from publicly available sequencing data from fresh tumours to decide whether or not chemical repair should be used in the laboratory analysis. Next, we advise users to compute the expected SNR given the observed mutations in the sample as suggested in the flowchart to maximise the signal for FFPEsig.

For already existing data without direct/obvious chemical treatment status available, we suggest the users contact the original data generator or consult the manufacturing company using the DNA extraction kit information. Nonetheless, the users should always apply the matched FFPE signature corresponding to how the sequencing data was generated.

For FFPE samples with any suspicion of excess amounts of T>C mutations of unclear origin (as observed in study 1 or our FFPE samples), we recommend the users to omit all T>C mutations in their signature analysis. We demonstrated above that the remaining 80 features still convey enough information for the algorithm to correctly complete the activity refitting task in most of the cases (Fig. 5a, Supplementary Fig. 16). Our solution of omitting T>C features is equivalent to performing feature selection on the 96 mutational channels to drop features with large unexplained variance, which is task-oriented and

proven to work. Nevertheless, the omission of T>C channel can cause difficulties to recover signatures featured in these channels, e.g. SBS12 and SBS16 (exclusively found in Liver-HCC) (Fig. 5a), and therefore we advise manual curation in samples where these signatures are suspected to be prevalent.

For FFPE samples without evidence of excess artefacts on T>C channels (i.e. their proportion and pattern are highly similar to the FF samples of the same cancer type, such as in study 2 and 3), we suggest our established FFPE signatures appropriately represent the artefacts in the data (Supplementary Figs. 2, 6f). The users therefore could apply FFPEsig using the 96-channel spectrum to obtain the true underlying biological processes.

We used our main set of simulated data as a benchmark to examine the performance of FFPEsig when the flowchart guidance for use was followed (Supplementary Fig. 18). We first assigned the ideal DNA repair treatment status to $n = 26$ main cancer types (with $n > 20$) by choosing the protocol with lower SNS (Supplementary Fig. 19). Then, we sorted the $n = 26$ cancer types into unsuitable for correction ($n = 8$), repaired-protocol ($n = 10$) and unrepair-protocol ($n = 8$) if we apply 0.1 as a cut-off of SNR (Supplementary Data 3). Overall, we obtained 1564 (-79%) well-corrected cases among 1979 suitable samples from $n = 18$ cancer types, and this figure increased to -86% if we apply SNR cut-off of 0.2. Note that here we applied an automated measurement of SNR which very likely under-estimates those of real samples; the overall achievement of in a real scenario variant filtering can be adjusted to increase SNR and rescue samples despite low biological mutation burden.

## Discussion

In this study, we derived genome-wide mutational signatures that result from formalin exposure in FFPE biospecimens and designed an algorithm, FFPEsig, to detect and remove artefactual-FFPE mutations from measured mutational profiles. We found repaired and unrepaired formalin mutational artefacts were predominantly distributed in C>T mutation channels and their mutational signatures were consistent across independent experimental studies (Fig. 1 and Supplementary Table 1). FFPE artefacts observed following chemical repair, a widely used protocol, mirror ageing signature SBS1 (Fig. 1f). In particular, they both are caused by deamination of 5-methylcytosine (5mC): SBS1 is due to spontaneous deamination in vivo; whereas the artefact signature is caused by chemically-modified deamination in vitro[7,29] (Supplementary Fig. 3). When the chemical-repair step is absent, the artefact signature in FFPE samples is highly similar to SBS30 (Fig. 1e). Biological SBS30 occurs more rarely: it is caused by loss-of-function in glycosylases in BER due to biallelic inactivation mutations in *NTHL1*, and patients carrying this variant are with an increased lifetime risk for CRC, breast cancer, and colorectal polyposis[27,28,35]. More generally, our results show that there is not necessarily a direct 1-to-1 mapping relationship from mutational process to a unique signature profile (as also questioned in[36]). Nevertheless, our findings speak to the utility of constructing a common carcinogen signature database[36,37].

The correction accuracy of FFPEsig was demonstrated on synthetic FFPE samples (Fig. 2). The accuracy in the majority of the samples was very high. Poorer performance occurred when (a) SNR is <0.1, and/or (b) for samples where the true mutational profile closely resembled the FFPE-artefact signature. To obtain a reasonable SNR, users can apply upstream filters to exclude the easy-to-remove FFPE artefacts which are known to have, for example, low allele-frequency, orientation bias or low-quality metric[7,32,33]. We advise future users to adjust their filtering if they suspect their data might fall closer to the SNR = 0.1 limit (Supplementary Fig. 18). We also noted that C>T artefacts accumulate rapidly with the duration of formalin exposure (Fig. 1a), and so, knowledge of the fixation time is a useful pre-analytical factor of determining the SNR, which could impact the downstream signature analysis. Over-fixation (e.g. over 48 hours)

should be avoided as the resulting high burden of formalin-induced artefact can be problematic for FFPEsig correction.

We demonstrated the need for correcting observed FFPE mutation profiles before inferring signature activity using both in silico and patient data. Uncorrected FFPE artefacts lead to misinterpretation of signature contributions in the majority of cases (Fig. 3). Of particular note, activities of HR deficiency tend to be largely underestimated in uncorrected FFPE samples, which could cause implications for patients who are suitable for PARP inhibition therapy in breast, ovarian and other cancers[17,20,21]. SBS1 and SBS30, on the other hand, are both likely to be overestimated in repaired and unrepaired FFPEs, respectively (Fig. 3). Since SBS1 is active in almost all tumours as well as in healthy tissue types[29], the impact of this mis-assignment can potentially have wide-reaching consequences for signature analysis in almost all tissue types. Our FFPEsig software provides a robust correction and enables accurate downstream mutational signature analysis. We also note that the statistical machinery within FFPEsig is generalisable and could be repurposed to correct for mutational noise from any source.

We observed that an excess number of putatively artefactual T>C mutations in FFPE samples from study 1 and biological interpretation of these mutations must be performed with extra care. We suspect the choice of DNA polymerase used in PCR or downstream steps, in combination with the DNA being FFPE derived, may be associated with those batch-related artefacts but have not investigated this suspicion. It is well-known that DNA polymerases exhibit varied levels of fidelity and by-pass efficiency, which is also termed as translesion synthesis (TLS)[38,39]. TLS represents the ability of reading through and incorporating a wrong base opposite a damaged site in DNA templates[7]. Indeed, T:G pairs are found as the most frequently synthesised and most easily extended base substitution errors for *Taq* DNA polymerase, which led to A:T>G:C artefacts[23,24]. Further, Y-family by-pass DNA polymerase would also cause a great number of A:T>G:C artefacts as the mis-incorporation of dGTP opposite of T is even more efficient than inserting dATP in this family[39]. Nonetheless, these batch-related artefacts are exclusively distributed in T>C channels without a consistent noise pattern (Fig. 1c). For FFPE samples with an excessive amount of T>C mutations present, our analysis suggests it is safe to remove this mutation class and perform analysis using the remaining 80-channel (non-T>C) profiles for the signature analysis as they contain enough information for the adequate signature decomposition (Fig. 5 and Supplementary Fig. 16).

In conclusion, here we identified two mutational signatures, linked to repaired and unrepaired-FFPE, which are highly similar to COSMIC signatures SBS1 and SBS30, respectively. We further developed FFPEsig software to accurately remove FFPE-induced mutational artefacts and demonstrated efficacy in silico and in new samples. It is necessary to correct FFPE artefacts prior to downstream mutational signatures analysis. Careful application of our approach will enable the robust study of mutational signatures in the enormous FFPE archives that exist around the world.

## Methods

Our research complies with all relevant ethical regulations. The archival FFPE samples were analysed in accordance with ethical approval from the UK Research Ethics Committee (REC: 18/LO/2051 IRAS:249008) whereby anonymised archival FFPE blocks were provided to the researchers without the requirement for patient consent.

### Targeted sequencing data

We used targeted sequencing data from two previous publications to learn FFPE signatures[10,11]. Prentice et al. has collected three sets of samples from CRC patients, stratified by fixation time (*n* = 3 patients), DNA extraction kits (also termed baseline; *n* = 20) and storage time (block-age, *n* = 9), to examine the impact of these factors on somatic mutation detection in FFPE samples[11]. Samples collected in the fixation

group were fixed in formalin for 2, 15, 24 and 48 hours for both repaired and unrepaired FFPEs, and paired FF samples were also available.

To validate if true somatic mutations are detectable in FFPE samples, Prentice et al. applied several filters on the mutation calling results, which could have filtered FFPE artefacts out. However, for our purpose of learning FFPE noise signatures, we pre-processed the whole mutations list to exclude non-FFPE artefacts as much as possible (also termed as FFPE-only mutations). Mutations were excluded if they met any of the following criteria, (1) being detected in all FF samples; (2) being detected in the matched normal samples; (3) with >0.90 posterior probability of being somatic mutations; and (4) being detected as common SNPs in germline databases using ANNOVAR[40].

We also included targeted panel sequencing data from study 2 in our analysis[10]. There were four normal breast tissues collected in the study. Triplicate samples were collected for each of them, including FF, repaired and unrepaired-FFPE. Further details about sample collection and preparation can be found in the original studies.

### Mutational opportunities of the targeted regions

To obtain mutational opportunities, we calculated all possible mutation frequencies of the targeted sequences in study 1 and 2. The FASTA sequences for targeted regions were downloaded for study 1 and for study 2. The whole-genome mutation opportunity was taken from[41].

### Deriving FFPE signatures

To derive FFPE signatures, we first applied t-distributed Stochastic Neighbour Embedding (t-SNE) for dimensionality reduction using cosine distance matrix of the merged 80-channel mutational probabilities. Based on the two principal components provided by t-SNE, we selected representative samples for both repaired and unrepaired-FFPE clusters using data point density estimated by gaussian kernel (from scipy.stats) (Supplementary Fig. 6a). The high-density samples (top 50%) were used to generate one set of FFPE signature candidates. With repeating the above procedure for 100 times using different initial values, we took the averaged values of each channel as the final FFPE signatures (Supplementary Fig. 6b, c).

### FFPE-only mutation pattern in the third study

We also validated our derived FFPE signature using a third independent study[30], in which filtered mutations are available for *n* = 11 lung adenocarcinoma FFPE samples. We selected the mutations with zero alternate reads in the validation sample from the FFPE-data list as FFPE-only mutations. The mutational profile is obtained from these FFPE-only mutations via SigProfilerMatrixGenerator[42] (Supplementary Fig. 6f).

### Simulation of FFPE samples

To simulate FFPE samples, we added different amounts of noise mutations with Poisson noise to biological mutation catalogues of 2780 canner genomes provided in Pan-Cancer Analysis of Whole Genomes (PCAWG) project by International Cancer Genome Consortium (ICGC)[14,31] (Supplementary Fig. 7). We then omitted all T>C mutations prior noise correction/signature decomposition, to simulate the real-life scenario where it is unknown whether a FFPE sample contains batch-related T>C artefacts.

### Development of FFPEsig for artefacts correction

We denote the observed mutation counts from the FFPE sample by $\mathbf{V}$, which was considered as a linear combination of FFPE-artefact signature $\mathbf{W_1}$ and biological mutation spectrum $\mathbf{W_2}$ with their corresponding attributions/activities $H_1$ and $H_2$. Thus, we have:

$$\mathbf{V} \approx \sum_{\mathbf{i} \in (1,2)} \mathbf{W_i} * H_i \qquad (1)$$

In this model (1), $\mathbf{V}$ and $\mathbf{W_1}$ were known and the task was to infer $\mathbf{H} = [H_1, H_2]^T$ and $\mathbf{W_2}$. Here, we utilised generalised Kullback-Leibler

(KL) divergence between reconstructed $\hat{\mathbf{V}} = \sum_{i \in (1,2)} \mathbf{W_i} * H_i$ and the observed profile **V** as the cost function and applied Lee and Seung's multiplicative update rules to minimise the cost function[43].

This update process iterated over at least 200 and up to 3000 steps by default until it met our termination criteria defined here. We calculated the convergence ratio using the average KL divergence from the last batch of 20 iterations divided by the second last batch of 20 iterations. The algorithm would terminate if the convergence ratio reached 0.95 (or defined by the users), or if the iteration process reached the maximum 3000 steps. The above one whole process provided inferred $\mathbf{W_2}$ and **H** as one candidate solution. We collected 100 candidate solutions using different random status and averaged them as our final solution for all samples analysed for FFPE noise correction in this study.

### Comparison of decomposition results between corrected and uncorrected mutation profiles

To compare the impact of FFPE artefacts on signature activity inference, we refitted active signatures in well-corrected FFPE samples identified in Fig. 2c. The active signatures for each sample were used for refitting if the original activities were greater than 0. Three 80-channel profiles of each sample (true biological, corrected, and uncorrected) were used for the refitting analysis using our locally implemented refitting algorithm to exclude possible bias introduced by different tools. We also studied the impact of FFPE artefacts on refitting for each individual signature, with or without known aetiology and such information was retrieved from COSMIC website (visited in April 2021).

### DNA extraction and genome sequence of FFPE CRC samples

The male patient with ulcerative colitis was diagnosed with cancer in the transverse colon at age 48 in St. Mark's Hospital, London, United Kingdom. The samples were collected and analysed in accordance with ethical approval from the UK Research Ethics Committee (REC: 18/LO/2051 IRAS:249008−Fulham committee). All samples were anonymised to the researchers.

Formalin-fixed paraffin-embedded (FFPE) sections of 10μm thickness were deparaffinized, rehydrated and lightly stained with methyl green. The annotated H&E was used as a guide for epithelial enrichment through targeted needle scraping of slides (for estimated epithelial cellularity >50%). To collect matched normal tissue, targeted scraping of serosal tissue from FFPE blocks was taken from a small intestinal segment distal to the cancer.

DNA was extracted using a modified protocol of the High Pure FFPE DNA Isolation Kit (Roche Life Science, Penzburg, Germany). The normal tissue DNA sample and one tumour DNA sample were repaired using the NEBNext FFPE DNA Repair Mix (New England Biolabs, Inc) following the manufacturer's recommendations. The other tumour DNA was left unrepaired. DNA libraries were prepared using the NEBNext Ultra II DNA Library Prep Kit for Illumina (New England BioLabs, Ipswich, Massachusetts, USA), followed by equimolar pooling strategy. Finally, all DNA libraries were sequenced on NovaSeq S2 for 50 bp paired-end reads.

### Somatic variants calling in WGS FFPE CRCs

The paired-end reads underwent initial quality control with FastQC[44] followed by default adaptor trimming with Skewer[45] and were subsequently aligned to the GRCh38 reference genome with BWA-MEM[46]. Aligned reads were sorted by genome coordinate (SortSam, Picard) and duplicate reads were flagged with GATK's MarkDuplicates[47]. The two FFPE tumour samples were called against the matched normal separately using the Mutect2 somatic variant caller from GATK[47]. Variants were marked with filters by FilterMutectCalls. Variants were kept if they were PASS by Mutect2, aligned to a canonical chromosome, had a total allelic depth of greater or equal to 10 in both the

tumour and normal sample and had 3 or more reads supporting the alternative allele in the tumour sample. The filtered variants from two FFPE tumour samples were merged into a single VCF file using VCFtools[48].

We used Platypus on the merged VCF file as the candidate somatic variant list and integrated local alignment with multi-sample variant calling to assess the evidence for these variants across all samples[49]. The resulting VCF file was further filtered to only contain variants: (1) if the FILTER flag was PASS or other acceptable filters (alleleBias, Q20, QD, SC, HapScore); (2) the variant was not a known germline variant; (3) a genotype was called for all samples; the genotype phred score was 10 or more in all samples; (4) the normal sample had no reads containing the variant and at least 3 or more reads supported the variant in a tumour sample. Variants present in two FFPE samples with 5 or more supporting reads were classified as concordant mutations.

### Comparison of signature refitting results of 80-channel and 96-channel spectrum

To obtain the set of 80-channel signatures, we dropped T>C mutation channels of COSMIC SBS signatures and renormalised them to sum up to 1. We next refitted 80c and 96c active signatures to the mutational catalogues with and without T>C mutations accordingly. The inferred activities for 80c signatures were then rescaled by dividing them by the total mutation frequencies of non-T>C mutation channels of corresponding 96c spectra. These rescaled 80c attributions were then used to compare to those inferred from 96c signatures.

### Statistics and reproducibility

We used the original sample sizes from cancer types without excluding any individuals. No statistical method was used to predetermine sample size. To derive statistics, we focused on cancer types or signature groups with sample size over 20. To demonstrate the impact of signature decomposition results using corrected and uncorrected FFPE profiles, we focused on well-corrected samples (accuracy > 0.90). The experiments were not randomised. The Investigators were not blinded to allocation during experiments and outcome assessment.

### Reporting summary

Further information on research design is available in the Nature Research Reporting Summary linked to this article.

## Data availability

The BAM and VCF of WGS FFPE-CRC data generated in this study have been deposited in the EGA database under accession code EGAS00001005331. The raw BAM data are protected and are not available due to data privacy laws, and the access can be obtained with the agreement with our Data Sharing Policy.

Source data are provided with this paper, which are available to download from our GitHub repository. All raw and processed data used in our study are available to download from this website.

Mutation list is available to download for study 1 and study 3. Mutations from study 2 are available upon request to the authors of the original study. Human genome assembly GRCh38 is downloaded from here. PCAWG signatures, mutational profiles and signature activity data are available from this website.

## Code availability

FFPEsig is implemented in python[50]. All data analysis codes (Jupyter Notebook and HTML format) can be found in our Github repository.

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

## Acknowledgements

We thank Virinder Singh Reen, Dr. Ignacio Vázquez-García and Nicole Rusk for helpful comments on the manuscript. We thank the team at St Mark's Hospital London, UK, Andrea Sottoriva, Inma Spiteri, Ann-Marie Baker, Salpie Nowinski, Jacob Househam and Chris Kimberley for support with FFPE sample provision and analysis. We also thank Prof. Amy C. Degnim, Dr. Chen Wang and the Mayo Clinic for help with data sharing. T.A.G acknowledges funding from Cancer Research UK (A19771 and A16581) and the Barts Charity (472–2300). E.L. is also supported by funding from Cancer Research UK (A19771). V.M. acknowledges funding from the Academy of Finland (345829). Open access funded by Helsinki University Library. This research utilised Queen Mary's Apocrita HPC facility, supported by QMUL Research-IT.

## Author contributions

Q.G., T.A.G. and V.M. conceived the study. Q.G. designed, carried out the data analysis, implemented the algorithm and interpreted the initial results. Q.G. and V.M. designed the algorithm. Q.G. and T.A.G. wrote the paper. V.M., T.A.G. and E.L. supervised data analysis. Q.G. and E.L. carried out the WGS FFPE case study. I.AB provided FFPE samples and performed genome sequencing. K.C performed mutation calling on the FFPE case. All authors read, edited, and approved the final paper.

## Competing interests

The authors declare no competing interests.

## Additional information

**Peer review information** *Nature Communications* thanks other anonymous reviewer(s) to the peer review of this work. Peer review reports are available.

