## [Peer Review File · Nature Communications]

REVIEWER COMMENTS

Reviewer #1 (Remarks to the Author): Expert in computational cancer genomics

In this manuscript, the authors address the problem of mutational artifacts in somatic DNA sequencing datasets obtained from FFPE specimens. These artifacts contribute a large number of observed mutations, rendering FFPE samples largely unusable for cancer genomics studies. Therefore, reliable post-hoc mitigation of the mutational artifacts would be an important advancement and could unlock the information contained in the potentially numerous FFPE samples stored in hospitals. Here, the authors present a computational algorithm based on signature deconstruction to improve the same in FFPE samples. While not aiming to determine for any given variant whether it was caused by the true somatic mutation process or by the FFPE storage, I think such an approach is still timely and relevant, as it allows for a more accurate prediction of the mutational processes that were at play during a given tumor's evolution. However, I also see some limitations, which might make its broad applicability uncertain.

1. I understand that the simulation study to assess the performance of FFPEsig practically only adds C>T mutations to the PCAWG samples. This is done according to the signature that was learned before from the FFPE-only mutations ignoring the T>C class. Then mutations are removed based on the same signature, i.e. only from the C>T class, and accuracy is measured from cosine similarity in the C>T class. Is this correct? It would be good to make this clearer, especially the caption of Fig. 1 should clearly state that T>C mutations were excluded in d and e.

2. Study 2 found up to 76 times as many artifacts as true somatic variants in unrepaired samples (across all mutation classes). In line with this, here the authors find for the C>T class a ratio of $0.989/0.011 = 89.9$ times as many discordant C>T artifacts as concordant C>T variants in unrepaired samples of study 2 (l. 87). These numbers are far from the assumptions made in the simulated data (Fig. 2 and Fig. 3), where the number of simulated artifacts (10^4) is on average about the same as the number of true variants in the PCAWG data (i.e. corresponding to a ratio of 1 instead of ~ 90). Similarly, in the two WGS CRC samples, a ratio of 12.8 was found (using a proxy for concordant mutations). This suggests that typical correction accuracies for real data would correspond to the regime 10^5 - 10^6 FFPE artifact count in Fig. 2g, yielding relatively low correction accuracies (0.85-0.90) and thus limiting the use cases for FFPEsig.

3. Continuing on point 2, the scope of the analysis of the dependence on the number of mutational artifacts (Fig. 2g) is limited, as it was restricted to 219 a priori very well-predicted tumors, which are enriched with having high tumor mutation burden (i.e. for which even an artifact admixture of 10^6 could represent only a fraction of the total mutation count). It would therefore be much more informative to express the noise count as a fraction of the total mutation count.

4. The analysis of the signature weight inference (Fig. 3) is limited, as it was restricted to the ~60% of samples that could successfully be reconstructed ("well-corrected samples"). In reality, it will be hard to assess if any given sample is well-corrected by FFPEsig and one would have to base that assessment on the suspected signature similarity between the FFPE artifact signature and the unknown true mutational process signatures active in the sample (cf. Fig. 2f) as well as the unknown ratio of FFPE artifacts to true somatic mutations (cf. Fig. 2e).

5. The information that is gleaned from analyzing the effects of FFPE artifacts on mutational signature inference (Fig. 3) is how the relative signature weights change (which is the main appeal of the presented algorithm). This analysis is somewhat confounded by the fact that even if a signature activity (weight*total number of mutations) in an uncorrected sample were perfectly inferred, the weight (shown in Fig. 3) would show a nonzero error epsilon, as the total number of mutations in the uncorrected sample is larger by 10^4 due to the admixture with simulated artifacts. For most applications, the signature activity is, however, the most important quantity.

6. Along the same lines, for many/most clinical applications, the relative contribution of a given signature is not relevant, as long as the signature is identified when it is present. For example, the panel for SBS3 in Fig. 3a shows that indeed in some uncorrected samples the contribution of SBS3 would not be identified. However, I am wondering to which extent this depends on the algorithm that is being used (described in the Methods). Some commonly used algorithms for signature refitting employ a lower threshold for the signature weights they assign (e.g. deconstructSigs by default does not report signatures that have a relative weight of 0.05 or less). Is this done here as well? If not, how robust is the assignment of zero weight to a given signature?

7. More broadly, in order to assess the usefulness of FFPEsig in the clinical setting, it would be good to have a quantitative assessment of the number of cases in which a binary signature signal (present/absent) is changed from present to absent when using uncorrected samples. This is particularly relevant as a majority of uncorrected samples shown in Fig. 3a,b and Fig. S7 has non-zero weights for a majority of signatures, implying no real need to correct the signature weights. Similarly, it would be good to quantify the inverse situation, in which a signature is wrongly inferred as present in an uncorrected sample (for which all known mutational signatures would have to be considered).

8. In Fig. S3, which describes the data from study 1, panel b shows the pairwise genetic similarity between samples. For C>T mutations, one observes a high similarity (~25%) between the repaired samples of a given patient, but also between repaired samples of different patients. As the underlying patient genomes are distinct, the probability to create an overlap of mutations of 25% between two samples from different patients is, however, negligibly small (the expected fraction of overlap would be on the order of the per-site mutation probability of <1%; calculated based on the mutation numbers

shown in Fig. S1, which are ~50% of all mutations, and the fact that study 1 used ~30 genes in their targeted sequencing). In addition, not even the high overlap between samples from the same patient is entirely clear, since if it is due to true somatic mutations that remain in the repaired samples, these samples should have an equally high or higher overlap with the fresh frozen tumor sample from the same patient. Yet the overlaps between FFPE samples and their associated frozen tumor samples are among the lowest measured overlaps in the figure. Similarly, it is not understandable why the fraction of shared T>C mutations is >50% for all sample pairs. I think this opens up questions about the reliability of the dataset from study 1. The authors mention that they use more mutations from study 1 than that study's authors, as they impose less strict filtering rules. Could this be at the root of these very surprising findings? Is there any admixture with germline polymorphisms playing a role (which, however, could not explain the lack of overlap with fresh frozen samples)?

9. In their own experiment, the authors find, similarly to study 1, a high proportion of T>C variants. However, referencing study 1 and the unexplained observations regarding T>C mutations from Fig. S3, the authors decide to again discard this important class of mutations. It is also curious that the repair procedure here, unlike in study 1 and 2, did not have an exclusive effect on C>T variants, as the fractions of mutations in the six different mutation classes remained the same after repair of more than half of all mutations (Fig. S8b), implying that similar fractions of C>T and T>C mutations were repaired. Could the unusually high amount of T>C variants have something to do with the mutation calling and filtering (e.g. too stringent filtering of germline SNPs)?

10. The signature extraction on FFPE-only mutations excludes T>C mutations. Given the relatively large amount of this mutation class among FFPE-only mutations (>50% in the study 1 repaired and unrepaired data, ~10% in repaired data in study 2, and an estimated ~41% in the WGS data from two FFPE CRC samples), it appears to be a severe drawback of the method that it cannot handle it, as addressed in Figs. 5, S13 and S14.

11. Another way of distinguishing between FFPE artifacts and true somatic variants is to leverage that the vast majority of artifacts should be present as singletons in the sample of sequenced cells, which implies that they will have extremely low variant allele frequencies after sequencing. Can the authors incorporate this piece of information into their algorithm to increase its accuracy?

Reviewer #2 (Remarks to the Author): Expert in mutational signatures and mutagenesis

In this manuscript, Guo et al. propose that a computational correction to mutational signature analyses to filter out artefactual mutations caused by FFPE treatment of tissue specimens. They reanalyze published data, do a little bit of de novo experimental work, and run through extensive computational

exercises using the FFPEsig algorithm they proposed. I come away from reading this manuscript feeling less enlightened and more perplexed about this topic because the data presentation was not so straightforward to understand, the information presented don't fit together in ways that make sense, the foundation this manuscript is built on is questionable, and some of the methodology is very dodgy.

This entire manuscript is based on results from two other studies comparing FFPE genomes that were treated with a repair cocktail or not. A major problem is that the two cited studies don't agree with each other very well (Supp Figures 1 and 2). In study 1, FFPE induced a few hundred C>T and T>C mutations. Repair cocktail treatment corrects most of the C>T mutations, but not the T>C. In study 2, FFPE induced a huge number of C>T, but relatively few T>C (it's hard to tell exactly how many because of the vertical scale). When given repair treatment, most of the C>T are removed, but that reveals some T>C. Indeed, the two cited studies seem more inconsistent than consistent. The authors go to great lengths to argue that the T>C don't matter. If the T>C part can be dropped out, then the two studies are more consistent. I find that unpersuasive. We don't know if study 1 is more or less typical than study 2. Because of this uncertainty, I don't think it makes sense to sweep all the T>C mutations under the rug. I agree that C>T is more discordant between samples than T>C. But the concordance ratios for T>C aren't that high, they are around the 0.5 range (Supp Figure 3). The claim that "T>C mutations are highly repeated" is an overreach.

The authors should be aware of a paper from Ketan Patel in Molecular Cell from December 2020, which showed that mice deficient in aldehyde detoxification produce a mutational signature that is more complex, with increases in many substitution types. This is consistent with the chemistry of aldehydes, which can react with multiple DNA moieties. Patel rightly notes that formaldehyde reacts with guanine, and this might explain why there's a subset of C>T substitutions that are not rectified by the repair cocktail. Patel's results lend further credence to thinking that the total FFPE-induced signature is more complex than simply causing more C>T substitutions. Saying that the SBS30-like pattern is the FFPE signature is another overextended claim. It would be more accurate to call it an FFPE induced cytosine deamination pattern, which is a subset of the total FFPE signature.

The claim that SBS5 accounts for 74.1% of uncorrected FFPE sample SP21528 looks very questionable (Figure 2). SBS5 has a prominent T>C component, which is not apparent in this sample. If the uncorrected sample is modeled as having that much of SBS5, surely the reconstruction error would be large?

Data presentation is a bit confusing. If I understand it right, Figure 2 deals entirely with "synthetic" FFPE samples, except for part d. In the synthetic samples, FFPE mutations are deliberately spiked in, the data set is computationally decomposed into COSMIC signatures, and not surprisingly, some of the C>T substitutions "stick" to the fake FFPE mutations that were spiked in. So the net result is a slight correction at the margins. 2d deals with real, unaltered cancer samples but the accuracy results are pretty mixed. I wonder if the correction is introducing more confusion and obfuscation for some

samples instead of better insight? There are samples where the correction accuracy is especially low. This suggests a one size (or one correction) fits all approach doesn't really make sense. I can think of at least two possible reasons for so much variability. Maybe some samples fit the study 1 profile, where FFPE induces many T>C, which aren't corrected, while other samples fit the study 2 pattern more, where the correction does better. Another possibility is that some of the variability is related to different treatment of samples. Were some samples sequenced fresh, without going through FFPE? How much variability was there in FFPE treatments? Supposedly, there were some samples with 10^6 FFPE artefact mutations. I don't think this can be a major reason for the spotty performance of the correction because there aren't many cancer genomes with 10^6 mutations, even in total.

The censoring of T>C substitutions (Figure 4 and thereafter) is very dubious and objectionable. The importance of T>C mutations is uncertain and surely up for debate. Summarily removing T>C mutations is very much contrary to scientific transparency, especially considering that T>C substitutions are the most common type of mutation in the Figure 4 data set! As a reviewer, I am highly skeptical of this ploy, which throws away a lot of data. The authors admit that T>C substitutions can be frequent (~40% in their WGS samples) and do not show consistent patterns. In other words, they don't know what to do with all the T>C substitutions. I also noticed that the proportions of the six substitution types are pretty much the same between unrepaired and repaired samples. That seems a surprising result because it implies that the repair cocktail reduced all mutation types on a roughly equal basis, which appears to contradict the two cited studies showing that C>T mutations are rectified. There's more going on here than the authors are wanting to tell in their story and there are parts that don't fit neatly together. All the more reason for them to redouble their investigations with an open mind so they can write a paper that enlightens more than it baffles.

We would first like to thank both reviewers for providing helpful feedback on our manuscript. We have carefully assessed and replied to each point. This has required substantial new analyses, which now is included in our revised manuscript. The revisions strengthen the conclusions of our initial submission: that we identified the mutational signature of formalin on the cancer genome and serve to strengthen the applicability of our method.

The order of our reply is based on the original order of the comments received from both reviewers. For example, “Reply 1.x” is our response to “Comment 1.x” from Reviewer #1, and “Reply 2.x” is for “Comment 2.x” is from Reviewer #2. Please note the line number used in this reply is based on our revised manuscript, and the figure index is according to the revised figures.

Table of Content

Comments from Reviewer #1

Reply-to-Comment 1.0

Reply-to-Comment 1.1

Reply-to-Comment 1.2

Reply-to-Comment 1.3

Reply-to-Comment 1.4

Reply-to-Comment 1.5

Reply-to-Comment 1.6

Reply-to-Comment 1.7

Reply-to-Comment 1.8

Reply-to-Comment 1.9

Reply-to-Comment 1.10

Reply-to-Comment 1.11

Comments from Reviewer #2

Reply-to-Comment 2.0

Reply-to-Comment 2.1

Reply-to-Comment 2.2

Reply-to-Comment 2.3

Reply-to-Comment 2.4

Reply-to-Comment 2.5

Reply-to-Comment 2.6

References

Comments from Reviewer #1

Reply-to-Comment 1.0

Comment 1.0. *In this manuscript, the authors address the problem of mutational artifacts in somatic DNA sequencing datasets obtained from FFPE specimens. These artifacts contribute a large number of observed mutations, rendering FFPE samples largely unusable for cancer genomics studies. Therefore, reliable post-hoc mitigation of the mutational artifacts would be an important advancement and could unlock the information contained in the potentially numerous FFPE samples stored in hospitals. Here, the authors present a computational algorithm based on signature deconstruction to improve the same in FFPE samples. While not aiming to determine for any given variant whether it was caused by the true somatic mutation process or by the FFPE storage, I think such an approach is still timely and relevant, as it allows for a more accurate prediction of the mutational processes that were at play during a given tumor's evolution. However, I also see some limitations, which might make its broad applicability uncertain.*

Reply 1.0. We are delighted that the reviewer found both our findings and method timely for unlocking clinical applications in archival biospecimens. We thank the reviewer for their positive and accurate summary of our paper. Indeed, our method does not focus on distinguishing whether an individual mutation is true (biological) or not, but rather focuses on delineating biological signal from aggregated mutation patterns. We treat formalin fixation as an additional mutational process, which can be identified and removed using the statistical framework underlying the COSMIC mutational signatures. We hope our additional work and following replies can address the limitations highlighted by the reviewer.

Reply-to-Comment 1.1

Comments 1.1 *I understand that the simulation study to assess the performance of FFPEsig practically only adds C>T mutations to the PCAWG samples. This is done according to the signature that was learned before from the FFPE-only mutations ignoring the T>C class. Then mutations are removed based on the same signature, i.e. only from the C>T class, and accuracy is measured from cosine similarity in the C>T class. Is this correct? It would be good to make this clearer, especially the caption of Fig. 1 should clearly state that T>C mutations were excluded in d and e.*

Reply 1.1. Thank you. Yes, the correction accuracy is measured using cosine similarity in C>T channels as it reflects the performance of FFPEsig most accurately (more conservative than using all of the channels). We did not include T>C class while deriving the FFPE-signatures (they were treated as missing data and assigned to zeros), the rationale for which is explained in detail below in Reply 1.10 & 2.1, and we have now added this information in the caption of Fig 1.

Reply-to-Comment 1.2

Comment 1.2. *Study 2 found up to 76 times as many artifacts as true somatic variants in unrepaired samples (across all mutation classes). In line with this, here the authors find for the C>T class a ratio of $0.989/0.011 = 89.9$ times as many discordant C>T artifacts as concordant C>T variants in unrepaired samples of study 2 (l. 87). These numbers are far from the assumptions made in the simulated data (Fig. 2 and Fig. 3), where the number of simulated artifacts (10^4) is on average about the same as the number of true variants in the PCAWG data (i.e. corresponding to a ratio of 1 instead of ~ 90). Similarly, in the two WGS CRC samples, a ratio of 12.8 was found (using a proxy for concordant mutations). This suggests that typical correction accuracies for real data would correspond to the regime 10^5 - 10^6 FFPE artifact count in Fig. 2g, yielding relatively low correction accuracies (0.85-0.90) and thus limiting the use cases for FFPEsig.*

Reply 1.2. We thank the reviewer for raising this concern. We think the noise count (10^4) used in our main simulation reflects the realistic noise ratio in real FFPE samples. Because our aim is to correct a reasonable amount of FFPE noise in a setting where typical artefact-filtering steps (e.g., as recommended by GATK best practices) have been applied to the data prior to the correction. But we should have been clearer about this and the noise limits that FFPEsig is able to handle. We think the upstream artefact-filtering steps are necessary to get rid of the easy-to-remove FFPE artefacts, even when using FFPEsig, as it allows to concentrate the “signal” (true mutations) to enable a confident correction. Otherwise, the total mutation counts in most cancer genomes would be neglectable when mixed with >1 million artefacts, as the reviewer pointed out. We now clarified this information in our revised manuscript in lines 211~213 & 545~547.

In our main simulation set-up, we added $\sim 10^4$ noise mutations on top of somatic mutation profiles derived from fresh tumours in the PCAWG project, which yielded an averaged noise ratio of 17 (rather than 1), which is calculated using mean of biological-mutation-load/noise-load of C>T mutations. This corresponds to the assumption that the majority of artefacts have been previously filtered, for example low frequency variants

(which are expected to be enriched for FFPE artefacts) were already discarded. The approximate noise ratio of our WGS samples is ~10.6 in unrepaired-FFPE (similar to the reviewer's estimation) and ~4.7 in repaired-FFPE (calculation also made on C>T mutations). We note that both values are conservative estimates given there are some true somatic mutations that are not included in the concordant mutations due to our stricter filtering for this class. Nonetheless, both ratios, following our extensive filtering pipeline described in the methods, are matchable to the simulation noise set-up.

In the processing of study 1 and 2 data we, on the other hand, included the artefacts from an unfiltered mutation list in order to best map and learn the FFPE noise pattern. The high noise ratio highlighted by the reviewer above is due to the "raw" nature of these datasets. We believe the expected noise ratio in post-filtered data sets is substantially lower. Herein, we re-calculated the noise ratio on the filtered mutation list in study 1, where only mutations with probability score of ≥ 0.9 reported by the mutation caller MutationSeq¹ were retained. The filtered data of study 1 yielded a noise ratio range from 1 to 7 for n=21 repaired and unpaired samples. But we noted that n=3 unrepaired FFPE samples fixed in formalin for 48 hours still contained significantly higher noise ratios (26, 45 and 77) than the other n=21 samples. Therefore, our set up of noise ratio is suitable for most of the FFPE samples, except those with exceptionally high fixation time. We have now included this information in our discussion (lines 550~552). We did not calculate the noise ratio in study 2 as the authors used normal breast tissue in their study, so it lacks the reference to compute such a value.

Reply-to-Comment 1.3

Comment 1.3. *Continuing on point 2, the scope of the analysis of the dependence on the number of mutational artifacts (Fig. 2g) is limited, as it was restricted to 219 a priori very well-predicted tumors, which are enriched with having high tumor mutation burden (i.e. for which even an artifact admixture of 10^6 could represent only a fraction of the total mutation count). It would therefore be much more informative to express the noise count as a fraction of the total mutation count.*

Reply 1.3. We thank the reviewer for this suggestion. We followed the suggestion and used SNR as the measurement in our analysis (see below for new Fig 2g & 2h). We added five increasing levels of noise, from 10^3 , 10^4 , 5×10^4 , 10^5 to 10^6 artefactual mutations to all tumours (n=2,780) instead of 219 well-predicted cancers. These five levels of noise count are equivalent to a noise ratio of 1.7x, 17.4x, 87.1x, 174.1x, 1741.7x compared to the signal count on average. We then split n=2,780 samples into four categories based on an increasing order of true-mutation counts, in order to separately analyse hyper- and hypo-mutated cancers. The

four categories include, 1) n=278 samples with the lowest mutation count (Q:0-10%); 2) n=1,112 samples with the lower medium mutation count (Q:10%-50%); 3) n=1,112 with higher medium count (Q:50-90%) and 4) n=278 samples with the highest count (Q:90%-100%).

Overall, we observed that FFPEsig performs well in most samples when SNR is ~ 0.1 or higher, achieving mean accuracy of 0.93 and 0.94 for unrepaired and repaired FFPEs, respectively. However, its performance drops quickly when the signal/true-mutation-count is smaller than 10% of the noise count. Similar to any other computational tools, FFPEsig also has its limits but it is understandable it would reach the limit if the stochastic variability of noise itself is even larger than the signal. In real FFPE samples, this cut-off is also achievable based on our WGS samples as well study 1 (Reply 1.1).

We also noted that the correction was accurate for samples from Q:90-100% regardless of added noise counts (0.91-0.97 for unrepaired FFPE; 0.92-0.98 for repaired FFPE), even the most extreme added FFPE noise (10^6 artefactual-mutations) accounts to an SNR of 0.05 or above for these samples. In contrast, samples from Q:0-10% were typically not well corrected by FFPEsig across almost all noise levels (0.52-0.81 for unrepaired-FFPE; 0.34-0.89 for repaired-FFPE), as in this case even a low FFPE noise count led to SNR below 0.01 (100x artefacts compared to true biological mutations) due to the low true biological signal. Therefore, FFPE noise is expected to have limited effect and be easily rectified in hyper-mutated samples, but FFPEsig correction should be applied with caution in samples with very low mutation load. We further discussed this topic in Reply 1.4.

Fig 2g-h. The correction of FFPEsig works well in samples with SNR > 0.1 . For both unrepaired (g) and repaired (h) FFPE types, we generated five sets of simulated samples (n=2,780 per set) by adding five increasing noise levels from 10^3 , 10^4 , 5×10^4 , 10^5 to 10^6 noise mutation count to all n=2,780 samples. For each set of samples, we divided them into four quantile-categories depending on their true biological mutation load, 1) Q:0-10%, samples with the lowest

mutation count ($n=278$); 2) Q:10%-50%, with the lower medium mutation count ($n=1,112$); 3) Q:50-90%, with higher medium count ($n=1,112$) and 4) Q:90%-100%, with the highest count ($n=278$). We split the samples as their biological mutation loads varied significantly which consequently lead to different SNRs under the same noise level. For example, the second node of each line is corresponding to samples mixed with 10^4 artefacts. We also highlighted the cut-off of SNR (0.1) which is used in our recommended workflow. The changes of averaged correction accuracy with 95% confidence intervals (shown in error bar) were plotted as a function of the averaged signal-to-noise ratio for each quantile-category separately.

We also changed mutation load measurement to SNR in figure 2c & e, but the results are the same given all the samples were mixed with the same amount of noise (10^4). Therefore, the original mutation load group ("high", "medium", and "low") maps perfectly to the new groups identified using SNR. All these changes are now included in the revised manuscript (mainly in lines 237~244 & 269~282).

Reply-to-Comment 1.4

Comment 1.4. *The analysis of the signature weight inference (Fig. 3) is limited, as it was restricted to the ~60% of samples that could successfully be reconstructed ("well-corrected samples"). In reality, it will be hard to assess if any given sample is well-corrected by FFPEsig and one would have to base that assessment on the suspected signature similarity between the FFPE artifact signature and the unknown true mutational process signatures active in the sample (cf. Fig. 2f) as well as the unknown ratio of FFPE artifacts to true somatic mutations (cf. Fig. 2e).*

Reply 1.4. We thank the reviewer for raising this practical problem from the potential users' perspective. It is correct that FFPEsig obtained overall ~60% of well-corrected samples for all PCAWG samples ($n=2,780$). However, this figure varied dramatically among different cancer types from 2% to 100% (Fig 2d). Therefore, we argue that for certain cancer types the probability of good correction is significantly higher; and a given cancer type itself is an important *prior* information that can help in knowing ahead of time whether a high quality correction is to be expected. Indeed, in our manuscript we have extensively mapped out this context at cancer type specific level (Fig 2d) - the goodness of correction depends on the true mutation burden and common mutational signatures in the cancer type, both of which can be reasonably expected to be approximately known before proceeding with mutational signature analysis.

More specifically, the two main factors influencing FFPEsig correction success are the SNR and signal-to-noise similarity (SNS). We can get an approximate estimation of both by leveraging information from the increasingly large sets of existing data versus the observed values in the given FFPE sample we wish to analyse. Herein, we propose an approach for approximating SNR and SNS from known cancer type-specific characteristics:

- 1) Estimation of SNR, which can be obtained using the total number of observed FFPE mutations (O) and averaged mutation load (A) in the known cancer type as $A/(O-A)$. For example, it could be challenging to apply FFPEsig on an FFPE sample with 100,000 mutations but only ~500 mutations on average are reported in known fresh frozen samples. We think >10% of SNR would be appropriate for applying FFPEsig (see Reply 1.3) (Fig 2g & 2h). The upstream artefact-filtering steps can help to concentrate this ratio (see Reply 1.1 & 1.11).
- 2) Estimation of SNS. We could compare the overall similarities between known cancer mutation profiles against FFPE signatures to select a more suitable protocol for a given cancer type (Fig S9). We found significant SNS differences were observed in 24 out of 26 cancer types, and only 2 of them showed no differences. In all cancer types, selecting the protocol with the lower averaged similarity profile is recommended.

Fig S9. Similarities between PCAWG samples with FFPE signatures. We compared the overall similarities (on C>T channel) between known fresh frozen cancer profiles against two

FFPE signatures. We recommended applying this prior knowledge while deciding whether enzymatic treatment would impact the FFPEsig correction.

To provide guidance for potential users, we designed a decision-making flowchart summarising our recommended protocol in deciding whether FFPEsig is applicable for a sample from a given cancer type (Fig S10). We applied this protocol on our main set of simulation data (noise= $\sim 10^4$) and re-evaluated FFPEsig performance. In total, following the guidelines of Fig S10, we retained 2,086 samples from $n=19$ cancer types, of which our methods obtained overall 1,672 ($\sim 80\%$) well-corrected cases. With more conservative SNR cut-offs of 0.2 and 0.5, this increases to 86% and 92%, respectively. And a general higher SNR result could be achieved with more efficient filtering methods becoming available. We now added this part of analysis in lines 284~301.

Fig S10 Recommended decision-making flow chart for mapping out the suitable application context for FFPEsig correction. We applied this flowchart on samples from $n=26$ PCAWG cancer types (each type with sample size >20). Estimation of SNR was obtained via $A/(O-A)$, where O is the total number of observed FFPE mutations in the simulated FFPE samples and A is the averaged mutation load in the given cancer type. An averaged SNS was calculated using cosine similarities between FFPE signatures with known biological profiles from the same cancer type (Supplementary Fig. 9).

Table R1. Cancer types (n=7) that may not be suitable for correction with FFPEsig

cancer_type	SNR_median	Recommendation	well_corrected_samples	total_sample	SNR_cutoff_0.1_passed
CNS-Medullo	0.042358	Unrepaired	51	146	Not Passed
CNS-PiloAstro	0.007433	Unrepaired	2	89	Not Passed
Kidney-ChRCC	0.070471	Unrepaired	24	45	Not Passed
Lymph-CLL	0.073857	Unrepaired	38	95	Not Passed
Myeloid-MPN	0.058774	Unrepaired	18	56	Not Passed
Panc-Endocrine	0.068668	Repaired	38	85	Not Passed
Thy-AdenoCA	0.043909	Unrepaired	12	48	Not Passed

We found n=7 cancer types are not suitable to apply FFPEsig due to an general low mutation load (manifesting as a low SNR < 0.1) (Table R1). We note that low mutation load poses a challenge to signature decomposition in general. Indeed, we found that in these 7 cancer types, the reconstruction of signature activity on fresh-frozen material works less well than for the other types (Fig R1).

Fig. R1 Fresh frozen samples with smaller SNR have lower reconstruction accuracy. We grouped all PCAWG samples (no FFPE noise added) into two categories based on their SNR values in our main simulation set (with 10^4 FFPE noise added).

Reply-to-Comment 1.5

Comment 1.5. The information that is gleaned from analyzing the effects of FFPE artifacts on mutational signature inference (Fig. 3) is how the relative signature weights change (which is the main appeal of the presented algorithm). This analysis is somewhat confounded by the fact that even if a signature activity (weight*total number of mutations) in an uncorrected sample were perfectly inferred, the weight (shown in Fig. 3) would show a nonzero error

epsilon, as the total number of mutations in the uncorrected sample is larger by 10^4 due to the admixture with simulated artifacts. For most applications, the signature activity is, however, the most important quantity.

Reply 1.5. We agree with the reviewer that absolute signature activity is an important quantity, but we think a correct proportional decomposition comes first in terms of interpreting the data in a biologically meaningful way. In the signature framework, a higher proportional weight means larger contribution to the cancer genome, and the dynamic change of the contributions implies shifts of the leading mutagenesis pathway. For example, the signature contribution introduced by chemotherapy drug(s) could provide longitudinal evaluation of drug-response and further guide the treatment plan according to the attributable measures².

The absolute signature activity, as the reviewer mentioned, is derived from $\text{weight} \times \text{total-mutation-count}$. With correctly assigned weights determined, this absolute activity could be used for comparison purposes only if other data generation conditions are the same for all samples. As those conditions, from sample pre-processing, to sequencing depth and the choice of mutation callers, could impact the total number of mutations, making absolute signature activities a biased measure. Unfortunately, it is often not feasible to eliminate all these factors, especially for FFPE samples (uncertain degree of DNA damage, varied storage time etc.). Therefore, we believe relative signature contributions (weights) have more general applicability. Consequently we focused on finding correct signature contributions from corrected FFPE mutation profiles using FFPEsig, in which we believe it has achieved a high level of success compared to uncorrected samples (Fig. 3).

Reply-to-Comment 1.6

Comment 1.6. *Along the same lines, for many/most clinical applications, the relative contribution of a given signature is not relevant, as long as the signature is identified when it is present. For example, the panel for SBS3 in Fig. 3a shows that indeed in some uncorrected samples the contribution of SBS3 would not be identified. However, I am wondering to which extent this depends on the algorithm that is being used (described in the Methods). Some commonly used algorithms for signature refitting employ a lower threshold for the signature weights they assign (e.g. deconstructSigs by default does not report signatures that have a relative weight of 0.05 or less). Is this done here as well? If not, how robust is the assignment of zero weight to a given signature?*

Reply 1.6. We thank the reviewer for raising this question. Our refitting algorithm reports the optimal weights solution for the given signatures in order to minimise the generalised

Kullback-Leibler divergence between the input and reconstructed mutation profile. The tool simply reports the optimal solution, including those with relative weights <0.05 , which is unlike deconstructSigs. Therefore, the close-to-zero weights for SBS3 in Fig 3a are what we got directly from the tool without any modification. An additional filtering (such as the reviewer mentions) could be readily applied on these raw results. However, we think it would be better for the users to decide how they want to further process the results depending on their needs, as the relative importance of false negative or false positive rates will be application specific.

To test how robustly our method could establish absence/presence of a given signature, we simulated $n=1000$ samples with average total mutation load 10^4 , and each of them having ~ 4 active signatures on average randomly selected from 13 signatures (SBS 1, 2, 3, 5, 8, 9, 13, 17a, 17b, 18, 37, 40 and 41) reported in PCAWG breast cancers. The weights of active signatures were simulated from Dirichlet distribution ($\alpha=0.5$) and the Poisson noise was added to each channel. We next assigned activity-weights to the breast cancer signatures given the 96-channel mutation counts of each simulated sample using our refitting algorithm and deconstructSigs³. We derived a binary confusion matrix for the prediction results for these two methods separately (Fig. R2). We applied the same threshold of 0.05 used by deconstructSigs to classify “0” (activity <0.05 , inactive) or “1” (activity >0.05 , active) labels on the true and predicted values.

Overall, our refitting algorithm performs similarly to deconstructSigs, yielding the same precision rate of 0.99, and a somewhat higher recall rate (0.98) compared to deconstructSigs (0.94) (Fig. R2). We found FFPEsig is more likely to assign an “active” label to an “inactive” signature (false positives). This type I error rate is slightly higher in FFPEsig (0.23%, $n=30$) than that in deconstructSigs (0.12%, $n=16$). However, deconstructSigs made much more type II errors (1.18%, $n=154$) than FFPEsig (0.52%, $n=68$), which we believe would cause a more severe impact while making clinical decisions. Taken together, we believe the refitting algorithm applied in our work is highly robust while assigning zero to a given signature.

Fig R2. Confusion matrix of binary classification of assigned activities using FFPEsig (a) and for deconstructSigs (b). We assigned label “1” to activity weights of 0.05 or more, and “0” to weights smaller than 0.05. True labels are determined using true activities generated during simulation. And predicted labels are determined using assigned weights by the two refitting algorithms. TN: true negatives; TP: true positives; FP: false positives; FN: false negatives. Precision = $TP/(TP+FP)$; recall = $TP/(FN+TP)$; accuracy = $(TN+TP)/(TN+TP+FN+FP)$; F1 Score = $2 * precision * recall / (precision+recall)$.

We disagree with the reviewer’s statement that the relative signature contribution is not relevant. We think it is a rather important quantitative measure (see Reply 1.5). For example, in a binary decision-making case, it would be difficult to call HR-deficiency for a cancer sample with only 0.06 contribution of SBS3, but the situation will become easier if the values are (say) >20%. Furthermore, whether the relative contribution is required or not in clinical application is still up for debate. With ever increasing numbers of clinical-relevant signatures being discovered/validated, various application scenarios would occur accordingly, therefore we think it is too early to conclude that the relative contribution of a given signature is not relevant.

We also note that while we offer a ready-made tool for subtracting FFPE artefacts from the observed mutation profiles, the FFPE signature itself we describe in this work and the corrected profiles can be directly integrated with other decomposition methods for activity assignment (which in turn might offer binary classifications or relative signature contributions). Indeed, how to best infer signature activities while utilising biomedical prior information as effectively as possible is a very active research area and any gains there can be integrated with our method in the future.

Reply-to-Comment 1.7

Comment 1.7. *More broadly, in order to assess the usefulness of FFPEsig in the clinical setting, it would be good to have a quantitative assessment of the number of cases in which a binary signature signal (present/absent) is changed from present to absent when using uncorrected samples. This is particularly relevant as a majority of uncorrected samples shown in Fig. 3a,b and Fig. S7 has non-zero weights for a majority of signatures, implying no real need to correct the signature weights. Similarly, it would be good to quantify the inverse situation, in which a signature is wrongly inferred as present in an uncorrected sample (for which all known mutational signatures would have to be considered).*

Reply 1.7. We followed the reviewer’s suggestions and quantified the false positive/negative rates on the inferred signature activities (Fig. S11). In our study, we provided the active

signatures identified by PCAWG team to the refitting algorithm under the assumption that, the PCAWG-identified signatures are the true mutational processes active in the cancer genomes, which allows us to 1) compare the decomposition results against the ground-truth deviations; 2) study the direct impact of decomposition using different profiles of a given sample without being confounded by overfitting problems during model selection for active signatures.

We assigned the binary labels to each activity inference, in which '0' means absent with contribution 0.1 or less and '1' means present with relative contribution >0.1. The overall assignment made using the corrected profiles by FFPEsig is highly reliable, with <5% of type I and type II errors, which are mostly contributed by misassignment between the two highly similar featureless signatures (SBS5 and 40).

In contrast, using uncorrected profiles has a profound impact while determining the present/absent status for a signature, with a total error rate of 40% and 25% for unrepaired and repaired FFPE samples, respectively. We noted the recall rate (type II error) is particularly high in unrepaired FFPE samples (52%), which means nearly half of true signatures would be predicted as absent using uncorrected FFPE profile. We observed a less severe, but still highly noticeable, influence using uncorrected profiles in repaired FFPEs. We assume the generally active SBS1 in cancer genomes buffered the impact in repaired FFPEs.

Taken together, in the revised manuscript (lines 308~310 & 314~328) we now demonstrate that uncorrected profiles severely influence signature contribution results, regardless of applying a binary and/or relative contribution in a specific case. We also demonstrate that these errors are rectifiable by FFPEsig.

Fig. S11 Confusion matrix of binary classification of assigned activities using corrected profiles by FFPEsig (a-b) and for uncorrected profiles mixed with FFPE noise (c-d). We assigned label “1” to activity weights of 0.1 or more, as being present, and “0” to signatures with contribution smaller than 0.1 as being absent. The true labels (y-axis) are determined using activities inferred from true biological profiles. And the predicted labels are determined using weights inferred from simulated unrepaired (a, c) and repaired (b, d) FFPE samples with or without correction using FFPEsig. TN: true negatives; TP: true positives; FP: false positives; FN: false negatives. Precision = $TP/(TP+FP)$; recall = $TP/(FN+TP)$; accuracy = $(TN+TP)/(TN+TP+FN+FP)$; F1 Score = $2 * precision * recall / (precision+recall)$.

Reply-to-Comment 1.8

Comment 1.8. In Fig. S3, which describes the data from study 1, panel b shows the pairwise genetic similarity between samples. For C>T mutations, one observes a high similarity (~25%) between the repaired samples of a given patient, but also between repaired samples of different patients. As the underlying patient genomes are distinct, the probability to create an overlap of mutations of 25% between two samples from different patients is, however, negligibly small (the expected fraction of overlap would be on the order of the per-site mutation probability of <1%; calculated based on the mutation numbers shown in Fig. S1, which are ~50% of all mutations, and the fact that study 1 used ~30 genes in their targeted sequencing). In addition, not even the high overlap between samples from the same patient is entirely clear, since if it is due to true somatic mutations that remain in the repaired samples, these samples

should have an equally high or higher overlap with the fresh frozen tumor sample from the same patient. Yet the overlaps between FFPE samples and their associated frozen tumor samples are among the lowest measured overlaps in the figure. Similarly, it is not understandable why the fraction of shared T>C mutations is >50% for all sample pairs. I think this opens up questions about the reliability of the dataset from study 1. The authors mention that they use more mutations from study 1 than that study's authors, as they impose less strict filtering rules. Could this be at the root of these very surprising findings? Is there any admixture with germline polymorphisms playing a role (which, however, could not explain the lack of overlap with fresh frozen samples)?

Reply 1.8. We thank the reviewer for their observation. We compared the chance of observing the same mutation in two samples or more using the complete dataset in study 1 without any filtering (Fig. S4a & 4b). It may contain unfiltered germline SNPs and very few somatic mutations as the reviewer pointed out; the initial aim of this analysis was to check the randomness of C>T and T>C under the assumption that true FFPE artefacts should be more random than real mutations or germline variants. T>C mutations apparently are not random, and they are also observed in fresh frozen samples (Fig. S4b). Thus, we excluded them when deriving FFPE signatures (equivalent to assign them to zeros). We further summarised the possible mutagenesis pathways for these batch-related T>C mutations (see Reply 2.1) and discussed the impact of them on our findings in Reply 1.10.

We also noticed that the concordance ratio of C>T is higher in repaired samples (~25%, exactly like the reviewer calculated), followed by unrepaired (~9%) and fresh frozen samples (~7%). We hypothesise that this was caused by the shared CpG sites between individuals, as of which 70%-80% are methylated in humans⁴. In Fig. S3 of the revised manuscript, we now summarise the mutagenesis pathways of C>T artefacts generation via deaminated cytosine (non-CpG sites) and 5-methylcytosine (almost exclusively in CpG sites) to highlight where observed FFPE artefacts originate from.

Fig. S3 The hydrolytic deamination of cytosine & 5-methylcytosine lead to C>T/G>A sequencing artefacts in unrepaired and repaired FFPE samples via different pathways. Formalin fixation could cause hydrolytic deamination of cytosine bases to uracil (left grey-panel), and the deamination of 5-methylcytosine (5mC) in CpG dinucleotides converts directly to thymine (right grey-panel). The deamination results in U:G mismatches where DNA polymerase incorporates adenine opposite to uracil in amplicon-based protocols, generating artefactual C:G>T:A substitutions in sequencing data. To mitigate deamination artefacts, some FFPE sequencing library preparations include “repair treatment” whereby uracil DNA glycosylase (UDG) is added to remove uracil bases prior to amplification. This repair method removes uracil but leaves the abasic sites (AP sites) on the DNA templates, which are typically not replicable or at very slow synthesis efficiency. Therefore, only a small number of artefacts would appear in the seq-data. However, UDG does not repair artefacts pathways from 5mC.

To test our hypothesis, we extracted concordant mutations shared between sample pairs and studied their patterns. We classified the data into A (n=36), B (n=132), C (n=144) and D (n=132) four groups depending on the sample pair information as demonstrated in Fig. R3. Concordant mutations shared by fresh frozen and FFPE samples (group A) are expected to be the mixture of unfiltered SNP and a few somatic mutations. Therefore, group A can also be

considered as a control group. In contrast, concordant mutations from the other three groups (B, C & D) are the ‘non-random’/concordant artefacts shared by two FFPE samples.

Fig. R3 Schematic Figure shows an example of four groups of concordant mutations based on the sample-pair information. We grouped the concordant mutation data shown in Fig. S4b (C>T panel) into four categories, with the diagonal sample-pair excluded as they are from the same sample. Group A refers to concordant mutations observed in a fresh frozen sample with a repaired FFPE sample; B refers to concordant mutations observed between two repaired samples; C refers to concordant mutations from a repaired-unrepaired FFPE sample pair; D refers to concordant mutations derived from two unrepaired FFPE samples. We showed the concordant mutation pattern from each group on the right panel as an example.

We then compared the concordant mutation profiles to repaired and unrepaired signatures (Fig. 1d & 1e). We found all non-random artefacts profiles share a highly similar pattern to the repaired FFPE signature (Fig. R4), with averaged cosine similarity 0.92, 0.91 and 0.86 for group B, C and D, respectively. In contrast, the non-random artefacts in these three groups are dissimilar to unrepaired FFPE signatures (cosine similarity B=0.17, C=0.21, D=0.54). We also noticed that the non-random mutations from two unrepaired FFPE samples (group D) are dominantly distributed in CpG sites. So, we think FFPE noise is more prevalent at CpG sites (5mC), but more random at non-CpG sites. We believe this imbalanced distribution can explain the higher concordant ratio in repaired samples, because they have a similar chance of observing conserved artefacts at CpG sites (the ‘numerator’, not being affected by UDG), but a much smaller total noise count (the ‘denominator’). This part of analysis is mentioned in lines 116~126.

Fig. R4 Cosine similarities between concordant mutation pattern and the two FFPE signatures. The group information of A ($n=36$), B ($n=132$), C($n=144$) and D($n=132$) is demonstrated in Fig. R3.

Following from the above analysis, we believe the concordant mutations do not influence our ability to discover/evaluate FFPE-related artefacts and study 1 provides an adequate dataset for this analysis. In particular, the following observations are available to confirm the validity of this dataset:

- We observed the noise load increased along with the longer fixation time in their experiment results (Fig. 1a).
- We also found the FFPE noise pattern of study 1 perfectly clustered with study 2 ($n=8$) (Fig. 1b), which is further cross-validated by the additional data (Fig. S6d, see Reply 2.1).
- We gained more understanding regarding how random the observed artefacts are (Fig. R4).

We agree that the data shows the batch effect for containing a large number of T>C mutations, which we discuss in detail in Reply 2.1. However, this does not affect the usability of these data for deriving FFPE signatures as FFPE noise dominantly appears in C>T channels (Table S1), and the batch-effect is exclusive to T>C channels. We believe that there are several additional sources of noise (both formalin-related and independent) in archival data compared to fresh frozen samples, which limits the potential of using this vast collection of archival samples. We believe our novel discovery of the FFPE noise patterns is a valuable input to the field and indeed alleviates these limitations.

Reply-to-Comment 1.9

Comment 1.9. *In their own experiment, the authors find, similarly to study 1, a high proportion of T>C variants. However, referencing study 1 and the unexplained observations regarding T>C mutations from Fig. S3, the authors decide to again discard this important class of mutations. It is also curious that the repair procedure here, unlike in study 1 and 2, did not*

have an exclusive effect on C>T variants, as the fractions of mutations in the six different mutation classes remained the same after repair of more than half of all mutations (Fig. S8b), implying that similar fractions of C>T and T>C mutations were repaired. Could the unusually high amount of T>C variants have something to do with the mutation calling and filtering (e.g. too stringent filtering of germline SNPs)?

Reply 1.9. We thank the reviewer for this observation. The UDG (uracil DNA glycosylase) treatment only targets the uracil DNA lesions generated from the deaminated cytosines in FFPE samples, so indeed is not expected to have an effect on T>C mutations. We think the following few factors may cause this ‘non-exclusive’ effect in our WGS samples (lines 403~409).

The first reason we suspect is indeed due to mutation calling as well as filtering as the reviewer mentions. Our mutations were jointly called from mutect2 and platypus, whereas the variants in targeted sequencing data were called using one mutation caller only (MutationSeq¹ for study 1; smCounter⁵ for study 2). It is well-known that different mutation callers are implemented with varied algorithms causing the biased mutation yield⁶.

In addition, filtering mutations from targeted panels benefits more due to the substantially higher depth and the enriched prior knowledge for those hot-spot genomic regions, which unfortunately is not the case for WGS data. Therefore, targeted panel-seq data typically is more accurate to determine variants, whereas mutation filtering in FFPE WGS is more sensitive to sequencing coverage. Indeed, we found the coverage of filtered mutations in repaired FFPE WGS (mean: 36 for all filtered; 42 for concordant mutations) is significantly lower than that of mutations detected in unrepaired FFPE (mean: 42 for all mutations; 55 for concordant mutations) (Fig. S13a, p -value $<1.0e^{-16}$, one sided Mann-Whitney U test). This ‘low’ coverage of mutation sites in repaired samples would reduce the amount of overall mutation that can pass the filters and appear as a reduction of mutational numbers (in other than C>T channels) upon repair.

In our study, samples are labelled as repaired-FFPE if UDG was added, otherwise they are unrepaired-FFPE. We used a different DNA library preparation kit, NEBNext FFPE DNA Repair Mix (New England Biolabs, Inc), whereas study 1 and 2 both are using QIAseq GeneRead FFPE kit. The commercial DNA quality-repair protocols (differs from the UDG repairment) applied on both WGS libraries could have an elimination effect on replicating DNA templates containing those T>C mutations. Furthermore, The different amplification used (e.g.

WGA, PCR *Taq* and PCR cycles) that are undocumented could potentially contribute to this effect as well.

Reply-to-Comment 1.10

Comment 1.10. *The signature extraction on FFPE-only mutations excludes T>C mutations. Given the relatively large amount of this mutation class among FFPE-only mutations (>50% in the study 1 repaired and unrepaired data, ~10% in repaired data in study 2, and an estimated ~41% in the WGS data from two FFPE CRC samples), it appears to be a severe drawback of the method that it cannot handle it, as addressed in Figs. 5, S13 and S14.*

Reply 1.10. We understand the reviewer's concern. We would like to note that exactly as the reviewer listed, the proportions of T>C mutations are largely varied across the dataset we used, as well as the noise patterns, suggesting that T>C mutations might be enriched for additional processing and batch-related artefacts, but importantly are not attributable to formalin exposure. T>C mutations may arise from a number of other processes in the laboratory, which we detail in our Reply 2.1.

As we demonstrate in Table S1 below, we are not the only team that discovered this discrepancy in FFPE samples. A similar survey is also reported in a highly cited review paper by Do *et al.*⁷. Taken all evidence together, we believe the chance of observing large T>C mutations in the FFPE samples is low, but this fact has been biased by the selection of dataset we used in our previous manuscript. To study the alternative sources of these batch-related mutations (we proposed a few in Reply 2.1), separate experimental studies are required, which however is beyond the scope of this study.

We would also like to emphasise that our heuristic approach of excluding T>C channels has negligible impact on our main aims of deriving an FFPE noise pattern and providing and characterising a method to rectify FFPE noise in signature analysis.

- Extensive evidence showed that C>T mutations are predominant in FFPE noise and share consistent patterns through studies (Table S1, Fig. 1 & Fig. S6d). For T>C mutations, we treated them as 'missing data' with assignment of zeros, which is further confirmed as a reasonable approximation by Fig. S6d. Similarly, the two aging signatures (SBS1 in COSMIC v3 and the older sig1 in v2) share the main feature in CpG sites (C>T) but slight changes in non-CpG sites. The mutation pattern indeed evolves when a larger or better dataset becomes available. However, subtle changes

of T>C channels would not affect the determined/cross-validated feature of FFPE signatures (in C>T channels).

- Based on our review of the existing literature of FFPE-related artefacts, we believe that high T>C numbers are rarely observed (Table S1). However, there could be a few laboratories that would observe data with large T>C present, which prompted us to explore if the 80 non-T>C channels contain enough information for most of the decomposition. We show in Fig. 5 & Fig. S18 that most signatures are reconstructed well using the limited channel information - and the two main signatures being affected using 80-channel are SBS40 and SBS5, which are considered “featureless” signatures and are highly similar to each other. Indeed, we show that these signatures are assigned less accurately even when using 96-channels (Fig. 5b), as more data is needed to separate them into more distinguishable patterns.

Reply-to-Comment 1.11

Comment 1.11. *Another way of distinguishing between FFPE artifacts and true somatic variants is to leverage that the vast majority of artifacts should be present as singletons in the sample of sequenced cells, which implies that they will have extremely low variant allele frequencies after sequencing. Can the authors incorporate this piece of information into their algorithm to increase its accuracy?*

Reply 1.11 We thank the reviewer for this suggestion. We agree that artefacts may be enriched for low-AF and/or orientation biased mutations in FFPE samples. However, there is no one-set of cut-off values that could fit for all FFPE samples due to the much more complex pre-analytical factors, which include, but not limited to, uncertain degree of DNA damage, storage time, tumour purity and sequence depths. Also, concentration of SNR in different cancer types also requires flexibility and empirical knowledge. For example, a less strict filter can be applied for skin cancer (Fig. 2g & 2h), which may not be suitable for other types. Therefore, we would like to offer a flexible off-the-shelf approach and choose to focus on the "pure" mutational signatures. We believe FFPEsig could aid in correcting the varied noise in FFPE samples. We nonetheless advise potential users to perform upstream filtering in lines 543~555.

Comments from Reviewer #2

Reply-to-Comment 2.0

Comment 2.0. *In this manuscript, Guo et al. propose that a computational correction to mutational signature analyses to filter out artefactual mutations caused by FFPE treatment of tissue specimens. They reanalyze published data, do a little bit of de novo experimental work, and run through extensive computational exercises using the FFPEsig algorithm they proposed. I come away from reading this manuscript feeling less enlightened and more perplexed about this topic because the data presentation was not so straightforward to understand, the information presented don't fit together in ways that make sense, the foundation this manuscript is built on is questionable, and some of the methodology is very dodgy.*

Reply 2.0. We thank the reviewer for their comments. It is unfortunate that the reviewer after reading our work felt less enlightened and more perplexed. After carefully reading the reviewer's comments, we realised that there are some misunderstandings that we hope are clarified in the revised manuscript as well as in our replies below.

To answer all the questions, we included additional analysis and new data, aiming to strengthen our findings and to adjust our method. We also included new illustrations to further clarify the mechanisms of FFPE artefacts generation and our *in silico* experiment design. We hope following these revisions the reviewer will be reassured about the validity of our study.

Reply-to-Comment 2.1

Comment 2.1. *This entire manuscript is based on results from two other studies comparing FFPE genomes that were treated with a repair cocktail or not. A major problem is that the two cited studies don't agree with each other very well (Supp Figures 1 and 2). In study 1, FFPE induced a few hundred C>T and T>C mutations. Repair cocktail treatment corrects most of the C>T mutations, but not the T>C. In study 2, FFPE induced a huge number of C>T, but relatively few T>C (it's hard to tell exactly how many because of the vertical scale). When given repair treatment, most of the C>T are removed, but that reveals some T>C. Indeed, the two cited studies seem more inconsistent than consistent. The authors go to great lengths to argue that the T>C don't matter. If the T>C part can be dropped out, then the two studies are more consistent. I find that unpersuasive.*

Reply 2.1 It is true that there is discrepancy in T>C mutations among two studies we analysed. In our study, we aimed to analyse whether the large T>C mutations are FFPE-related. We found that T>C mutations are also present in fresh frozen samples, are non-randomly distributed over the targeted gene panel and are not positively correlated with fixation time length (Fig. S4). We therefore conclude that they are not caused by formalin exposure. Our aim was to identify the pattern introduced by FFPE and consequently we excluded non-FFPE noise when deriving FFPE signatures.

To address the reviewer's concern, we have now explicitly and extensively addressed the discrepancy of T>C mutations found in two studies. Specifically, we now: 1) summarise results about FFPE artefacts from the available literature; 2) confirm our inferred-FFPE signature using a third independent dataset.

- 1) Among the 20 publications we reviewed, all of them (100%) reported that C>T mutations are predominantly present (Table S1), 3 of 20 (15%) also included T>C mutations and 2 (10%) of them also detected C>A mutations. A similar survey is also reported in a highly cited review paper by Do *et al.*⁷. All these results agreed with our previous conclusion that formalin fixation artefacts are predominantly C>T mutations. We further summarised the mutagenesis pathways of artefactual C>T mutations in our revised manuscript (Fig. S3).

Despite that C>T mutations are consistently being observed in FFPE data, a few studies also showed other mutation types, like T>C and C>A (Table S1). These studies either reported T>C or C>A, on top of C>T, suggesting that both T>C and C>A mutations are batch related artefacts rather than FFPE induced.

Table S1. FFPE artefacts in other publications

Studies	Sample information	DNA extraction kit	Types of NGS	Sequencing platform	UDG	Predominant artefacts	Authors' results/conclusions
Flores Bueso et al. ⁸	Escherichia coli cells	QIAGEN FFPE DNA kit	WGS	Illumina HiSeq	Yes & No	C>T	"Our data suggest that DNA damage found in bacterial FFPE DNA is primarily driven by oxidation and cytosine deamination"
Chen et al. ⁹	FFPE samples (n=7)	Ion AmpliSeq Library Kit 2.0	TES	Personal Genome Machine (PGM) sequencing platform	Yes & No	C>T	"The baseline noise in normal peripheral blood and formalin-fixed paraffin-embedded samples detected by next-generation sequencing (NGS) is dominated by C:G>T:A mutations, which are signature mutations of cytosine deamination"
Williams et al. ¹⁰	basal cell cancer (n=1)	NA	TES	solid-phase sequencing	Not found	C>T	"A total of 28 artificial mutations were recorded, of which 27 were C-T or G-A transitions. "
Do et al. ¹¹	Squamous cell lung-carcinomas (n=3)	DNeasy Tissue and Blood KIT(Qiagen)	TES	Illumina MiSeq	Yes & No	C>T	"When the prevalence of each SNC type was examined, C:G>T:A were by far the most frequent in all 3 samples"
Do & Dobrovic et al. ¹²	Squamous cell lung carcinomas(n=5)	DNeasy Tissue and Blood kit (Qiagen)	TES	Sanger Sequencing	Yes & No	C>T	"the sequence artefacts detected in the FFPE tumour DNAs were almost exclusively C:G>T:A base substitutions (16/17)"; "Sequencing of these samples showed multiple non-reproducible C:G>T:A artefacts"
Yost et al. ¹³	Breast cancers (n=2)	BiOstic FFPE Tissue DNA Isolation kit (MO BIO, Carlsbad, CA, USA)	WGS	SOLID	Not found	C>T	"The tumor samples show differing amounts of FFPE damaged DNA sequencing reads revealed as relatively high alignment mismatch rates enriched for C · G > T · A substitutions compared to germline samples"
Spencer et al. ¹⁴	lung adenocarcinoma (n=16)	QIAmp Micro DNA kit (Qiagen, Valencia, CA)	TES	Illumina HiSeq 2000	Not found	C>T	"C to T transitions were significantly increased in FFPE tissue compared with frozen tissue ($P = 3.98 \times 10^{-10}$, Student's t -test), with a corresponding increase in G to A transitions"
Oh et al. ¹⁵	Cancer sample (n=5)	unknown	WES	Illumina HiSeq 2000	Not found	C>T	"In this analysis, we re-confirmed that C>T and G>A base transitions occurred specifically in the FFPE samples as formalin fixation artifacts."

Serizawa et al. ¹⁶	esophageal cancer(n=135)	QIAamp DNA FFPE tissue kit (Qiagen, Venlo, Netherlands)	TES	MiSeq sequencer (Illumina)	Yes & No	C>T	“We also confirmed the efficacy of UDG pretreatment in reducing C:G > T:A SNVs, which are the predominant type of sequencing artifacts observed in FFPE DNA“
Lin et al. ¹⁷	Lymph node tissues (n=16), neoplastic tissues (n=118)	QIAamp DNA kit (Qiagen, Valencia, CA)	TES	Pyrosequencing and sanger sequencing	No	C>T	“Baseline noise is consistent with spontaneous and FFPE-induced C:G -> T:A deamination mutations ”
Ofner et al. ¹⁸	Melanoma (n=96)	Biostic FFPE Tissue DNA Isolation-kit	TES	Sanger Sequencing	Not found	C>T	“C>T: 77.1%” (Table 2); “In our case, 8 out of 11 non-reproducible artifacts were C:G>T:A transitions”
Gallegos Ruiz et al. ¹⁹	non-small cell lung cancer(n=47)	QIAamp DNA mini kit	TES	Sanger Sequencing	No	C>T	“As detailed in Table 2, all artifactual mutations resulted from C>T or G>A transitions”
Sah et al. ²⁰	Cancer samples (n=44)	RecoverAll™ Total Nucleic Acid Isolation Kit for FFPE (Life Technologies)	TES	MiSeq (Illumina, San Diego, CA, USA)	No	C>T	“we observed that 75% of the false positives reported in Table 1 were C>T or G>A transition mutations.”
Alborelli et al. ²¹	Lung carcinoma (n=12)	Maxwell® 16 FFPE Plus LEV DNA kit (Promega, Wisconsin, USA, Cat.No. AS1135)	TES	Ion S5XL™ instrument (Thermo Fisher Scientific).	Yes	C>T	“Discordant variants were mainly unique to FFPE samples (34/40 discordant variants) and mostly C:G > T:A transitions with low allelic frequency, likely indicating formalin fixation artifacts”
Parker et al. ²²	Colon cancer (n=10)	FFPE DNA-extraction kits (Qiagen, Toronto, ON, Canada)	WES	HiSeq 2500 instruments (Illumina)	Not found	C>T	“The excess variants were classified as being consistent with the result of a base transition event due to cytosine deamination, which we refer to here as an FFPE transition variant (ie, C>T or G>A transitions).”

Quach et al. ²³	Normal colon	DNeasy Tissue Kit, Qiagen, Valencia, CA	TES	ABI 377XL automated sequencer	No	C>T & T>C	“Mutation types were different after fixation, with a predominance (92%) of transition mutations”; “Point mutations at A:T base pairs were significantly (p= 0.034) more frequent than at G:C pairs in the fixed DNA (2.9 to 1 versus a ratio of 1.2 to 1 in this DNA sequence)”
Marchetti et al. ²⁴	Lung-tumor (n=70)	phenol–chloroform protocol	TES	Sanger sequencing	No	C>T & T>C	“22 (92 percent) of these mutations were C→T/G→A or A→G/T→C transitions.”; “All the uncommon mutations detected were found to be artifacts.”
Wong et al. ²⁵	Ovarian cancer (n=70)	Not found	TES	Sanger sequencing	No	C>T & T>C	“There were 42 transitions (76%) and 13 transversions (24%). Transitions included 23 GC>AT and 19 AT>GC. Transversions included 7 GC>TA and 6 AT>TA.”
Do & Dobrovic et al. ²⁶	non-small cell lung cancer(n=4)	QIAamp DNA blood kit (Qiagen, Hilden, Germany)	TES	Sanger Sequencing	No	C>T	“In this study, nearly all the base changes were G to A or C to T mutations”
	non-small cell lung cancer(n=1)	QIAamp DNA blood kit (Qiagen, Hilden, Germany)	TES	Sanger Sequencing	No	C>T & C>A	8/17 C>T; 9/17 C>A, according to Table 2 (HotStar HiFidelity, Qiagen)
Lamy et al. ²⁷	Colorectal cancer (n=1,130)	RecoverAll™ Total Nucleic Acid Isolation Kit	TES	ABI PRISM 3130x/ Genetic Analyzer	Yes	C>T & C>A	“As a whole, 283 KRAS artefactual mutations were recorded from 187 analyses: 148 (52.3%) corresponded to G>A transitions, 103 (36.4%) to G>T transversions, and 32 (11.3%) were G>C transversions.”

n: FFPE sample size; TES: targeted exon sequencing; WES: whole exome sequencing; WGS: whole genome sequence

We suspect the choice of polymerase used in PCR (and reaction conditions) might be associated with those batch-related artefacts. It is well-known that DNA polymerases exhibit different levels of fidelity and by-pass efficiency, also termed as translesion synthesis (TLS)^{23,28}. Fidelity refers to the chance of incorporation of a wrong base opposite a correct base; and TLS represents the ability of reading through damaged DNA templates and incorporating a wrong base opposite a damaged base⁷. These two properties are particularly essential while replicating FFPE DNAs in vitro as they typically contain a much higher level of lesions. For example, Do *et al.* found nearly all C>A artefacts (9/10) in PCR products using HotStar HiFidelity (Qiagen) as DNA polymerase, but only 1/10 in HotStarTaq (Qiagen) PCR products and none in FastStart High Fidelity (Roche) products²⁶.

Similarly, T:G mispair is found as one of the most frequently produced and most easily extended base substitution errors for *Taq* DNA polymerase, which will lead to T>C accounts for 67% of the artefacts (Table S2)²⁹. The study also found when dNTP concentration is lowered from 800 μ M to 6 μ M, only T:G mismatches or perfect base pairs are extended³⁰. Further, Y family bypass DNA polymerase would cause a great number of A:T>G:C artefacts as the misincorporation of dGTP opposite of T is even more efficient than inserting dATP in this family²⁸.

Table S2. Mutational spectra of artefacts during PCR using six PCR enzymes.

Error types	Taq	AccuPrime• Taq , HF	KOD	Pfu	Phusion	Pwo
Transitions						
A•T >G•C	67	12	9	3	6	6
G•C >A•T	28	5	5	5	5	4
Transversion						
A•T >T•A	1	1	1	1		
A•T >C•G	2					
G•C >T•A						3
Indels						
	1 (T del.)		1 (T del.)		2 (A ins., TCT del in (TCT) ₅ run)	

Insertion = ins.; deletion = del. This table is adopted from ²⁹.

Taken together, separate experiment studies are required to investigate the above possible mechanisms for those batch-related artefacts. However, as they are not caused by formalin fixation damage and most of the laboratories do not observe them, we believe they have no impact on the foundation of this study. We now included these new findings in 95~145 & 572~588.

- 2) We further validated our discovery using an independent dataset from a third study³¹. Unlike study 1 & 2 where all variants were available, only strictly filtered somatic mutations are released in study 3. Among these mutations, we found 1,041 FFPE-only mutations in n=11 lung adenocarcinoma FFPE exome-seq DNA data. The aggregate profile shares 0.92 cosine similarity with the repaired FFPE signature derived above (Fig. S6d), implying the original data was repaired using UDG. This result provides an independent validation of our discovered FFPE signatures (lines 195~202). Furthermore, it also supports our conclusion that T>C artefacts discovered in study 1 were batch-related rather than by formalin fixation, as these mutations are absent in samples of this third datasets.

Fig. S6d The aggregated FFPE-only mutation profile from lung adenocarcinoma (LUAD) samples (n=11). FFPE only mutations are found in the filtered mutation list of FFPE samples in study by Van Allen et al.³¹. It shares a highly similar pattern with our discovered repaired FFPE signature (cosine similarity = 0.92).

Reply-to-Comment 2.2

Comment 2.2 We don't know if study 1 is more or less typical than study 2. Because of this uncertainty, I don't think it makes sense to sweep all the T>C mutations under the rug. I agree that C>T is more discordant between samples than T>C. But the concordance ratios for T>C aren't that high, they are around the 0.5 range (Supp Figure 3). The claim that "T>C mutations are highly repeated" is an overreach.

Reply 2.2 The concordance ratio is showing randomness of observed mutations. A lower ratio means higher randomness, and being randomly distributed is one of the key characteristics of FFPE noise. As reviewer 1 pointed out in their Comment 1.8, even a concordance of 0.25 is unexpected, and while might be explained by biases in repair

treatment, are generally indicative of another source of noise, e.g. a batch effect. Therefore, we took 1) the high concordance observed in T>C mutations; and 2) the fact that T>C mutations were not enriched in study 2 or consistently identified in the literature; and concluded that they are not associated with FFPE noise, and they should be excluded while deriving FFPE noise signatures (see also Reply 2.1).

Reply-to-Comment 2.3

Comment 2.3. *The authors should be aware of a paper from Ketan Patel in Molecular Cell from December 2020, which showed that mice deficient in aldehyde detoxification produce a mutational signature that is more complex, with increases in many substitution types. This is consistent with the chemistry of aldehydes, which can react with multiple DNA moieties. Patel rightly notes that formaldehyde reacts with guanine, and this might explain why there's a subset of C>T substitutions that are not rectified by the repair cocktail. Patel's results lend further credence to thinking that the total FFPE-induced signature is more complex than simply causing more C>T substitutions. Saying that the SBS30-like pattern is the FFPE signature is another overextended claim. It would be more accurate to call it an FFPE induced cytosine deamination pattern, which is a subset of the total FFPE signature.*

Reply 2.3 We thank the reviewer for highlighting this paper. We would like to note that the study from Patel's laboratory derived an SBS5/40-like signature in HSPC colonies (n=6) from double-knockout (*Aldh2* and *Adh5*) mice (Fig. R5, top panel)³². ALDH2 and ADH5 metabolise endogenous formaldehyde and mice with dysfunction of both genes have been detected with ~44 μ M formaldehyde in the serum. The authors observed an increase of ~200 SBS (predominantly T>A mutations) per genome from bone marrow cell colonies cultured for ~3 weeks. However, a recent study led by Serena Nik-Zainal reported a very different mutation pattern observed in human iPSCs with direct exposure to formaldehyde (120 μ M) (Fig. R5, lower panel)³³. We suspect this difference is due to *Aldh2*^{-/-} *Adh5*^{-/-} models may not exclusively metabolise formaldehyde, other mutagenic compounds may also occur.

Fig. R5 Two signatures reported in cell culture experiments with low volume of endogenous (a) or environmental (b) formaldehyde exposure. We used the figures provided in the two original papers^{32,33}.

In our study, we aim to identify the noise pattern generated during formalin-fixation in clinical archival samples. However, the above two studies are studying how living cells respond to low-dose formaldehyde^{32,33}. We found there are two fundamental differences between their work and our research topic:

- 1) **The DNA repair system functions on living cells, but not in human tissues dissected from patients (e.g. formalin fixed tissues).** The exposure to formaldehyde happens to non-living cells in FFPE samples, but to living cells in iPSCs or HSPCs culture. Kucab *et al.* validated that repair pathways, like DDR, DSB, MMR and TC-NER, are fully operative in the living cell system³³. Therefore, in living cells, the final mutation profiles are jointly shaped by DNA damage (formaldehyde) and the repair pathways, whereas FFPE tissue lacks the latter. We believe this explains why we observed much more mutations in FFPE samples but not in the above iPSCs or HSPCs cell culture experiment.
- 2) **The dose of formaldehyde used in fixation and cell-culture is on a completely different magnitude.** The cultured cells are exposed to very low levels of formaldehyde from 40 to 120 μ M, but a much higher dose, 4% of formaldehyde in formalin agent, is used to fix the cell structure in FFPE samples. With such different levels of mutagen concentration, the DNA damage levels are much different as well.

In summary, we believe the SBS5/40-like pattern discovered by Dingler *et al.*³² is different from the FFPE signatures discovered in our study due to difference in materials studied (living vs dissected samples) and applied formaldehyde concentration.

Reply-to-Comment 2.4

Comment 2.4 *The claim that SBS5 accounts for 74.1% of uncorrected FFPE sample SP21528 looks very questionable (Figure 2). SBS5 has a prominent T>C component, which is not apparent in this sample. If the uncorrected sample is modeled as having that much of SBS5, surely the reconstruction error would be large?*

Reply 2.4. We thank the reviewer for this observation. There is a misunderstanding in this analysis setting, and we apologise for the previous unclear description. In this analysis, we provided the active signatures identified by the PCAWG team to the refitting algorithm under the assumption that the PCAWG-identified signatures are the true mutational processes

active in the cancer genomes. Therefore, it is correct that SBS5 accounts for 74.1% in simulated FFPE profile, if the three true active signatures (SBS 1, 5 and 18) identified by the PCAWG analysis are given to the model. We now clarified this in our revised manuscript in lines between 222~224.

As the reviewer pointed out, the reconstruction error would be large for the uncorrected FFPE samples (Fig. R6). This suggested that the true signatures are not able to explain the FFPE sample with mixed noise ('Uncorrected'), particularly for unrepaired-FFPE samples (Fig. R6a), but not for repaired FFPEs (Fig. R6b). Nonetheless, the assigned weights are grossly mis-estimated for repaired FFPEs (Fig 3b & 3d).

Fig. R6 Reconstruction accuracy of our refitting algorithm for biological, corrected and uncorrected mutation profiles using fits of true-biological signatures.

In a real case, refitting signatures to those uncorrected samples would either lead to a different set of signatures or a different weight-assignment, causing serious mistakes. However, our corrected samples by FFPEsig showed a similarly high reconstruction accuracy compared to the true biological profiles (Fig 3e & 3f).

Reply-to-Comment 2.5

Comment 2.5. *Data presentation is a bit confusing. If I understand it right, Figure 2 deals entirely with "synthetic" FFPE samples, except for part d. In the synthetic samples, FFPE mutations are deliberately spiked in, the data set is computationally decomposed into COSMIC signatures, and not surprisingly, some of the C>T substitutions "stick" to the fake FFPE mutations that were spiked in. So the net result is a slight correction at the margins. 2d deals with real, unaltered cancer samples but the accuracy results are pretty mixed. I wonder if the correction is introducing more confusion and obfuscation for some samples instead of better insight? There are samples where the correction accuracy is especially low. This suggests a one size (or one correction) fits all approach doesn't really make sense. I can*

think of at least two possible reasons for so much variability. Maybe some samples fit the study 1 profile, where FFPE induces many T>C, which aren't corrected, while other samples fit the study 2 pattern more, where the correction does better. Another possibility is that some of the variability is related to different treatment of samples. Were some samples sequenced fresh, without going through FFPE? How much variability was there in FFPE treatments? Supposedly, there were some samples with 10⁶ FFPE artefact mutations. I don't think this can be a major reason for the spotty performance of the correction because there aren't many cancer genomes with 10⁶ mutations, even in total.

Reply 2.5 We would like to clarify that the data used in Fig. 2 is entirely from synthetic FFPE samples, including Fig. 2d. To avoid any future confusion, we now emphasise this in the main text (lines 246~247) as well as in Fig. 2d captions. In addition, we now included a schematic figure to illustrate our simulation process (Fig. S7). Briefly, we added *in silico* FFPE noise (~10⁴) to the mutational profiles derived in fresh, whole genome sequenced tumour samples in the PCAWG project. FFPEsig takes the synthetic FFPE samples and the noise (FFPE pattern) to predict the signal (biological profile). In this section, we evaluated the performance of FFPEsig on these simulated FFPE samples and identified the main factors that influence its performance, given the advantages of knowing the noise level and the ground-truth of biological profiles in such a set-up experiment.

Fig. S7 The process of generating synthetic FFPE samples. To simulate FFPE samples, we mixed the FFPE noise to the biological mutation profiles derived from fresh frozen

tumours in the PCAWG project. In the main simulation set-up, the FFPE noise count is about 10^4 . The unrepaired FFPE made from unrepaired-noise pattern (Fig. 1d) and repaired FFPE samples made from repaired-noise patterns (Fig. 1e).

The reviewer noted that FFPEsig performed poorly in some cancer types. The two main factors affecting its performance are signal-to-noise ratio (SNR) and signal-to-noise similarity (SNS) (Fig. 2e & 2f). In a low SNR case (e.g. CNS-PiloAstro which only has 112 C>T mutations on average), this weak signal poses a huge challenge for FFPEsig, as the noise volume ($\sim 10^4$) is overwhelmingly pronounced in such a case. In order to improve SNR, we therefore recommend using the upstream artefact-filtering approaches to concentrate the signal (see Reply 1.1 & 1.11).

The reviewer also pointed out that some synthetic C>T could ‘mimic’ the contextual distribution of true C>T mutations across the genome. Indeed, this is exactly how the second factor SNS affects FFPEsig’s performance. For example, colorectal cancer generally has those SBS1-like C>T features (similarity: ~ 0.9), and the repaired FFPE noise (also with SBS1-like features) would partially stick to these true signals. In such a case, over-correction could happen and cause the failure as it would be difficult for a decomposition tool to distinguish the true signal from the synthetic ones. We recommend using an unrepaired protocol for such a cancer type *prior* to performing signature analysis.

Therefore, the poor performance in these cancer types is not due to the reasons listed by the reviewer, namely presence of T>C mutations (T>C were not introduced as FFPE noise during simulation), or sample sources (all synthetic samples are generated use the same protocol), or chemical treatment variability (no treatment being applied).

Finally, we would also like to highlight there is unfortunately no “one size fits all” approach for all tumours from all cancer types. We now included a flowchart (Fig. S10) to guide the potential users in using cancer-type specific prior knowledge to decide how suitable FFPEsig is for correcting a given sample (see Reply 1.4).

Reply-to-Comment 2.6

Comment 2.6. *The censoring of T>C substitutions (Figure 4 and thereafter) is very dubious and objectionable. The importance of T>C mutations is uncertain and surely up for debate. Summarily removing T>C mutations is very much contrary to scientific transparency, especially considering that T>C substitutions are the most common type of mutation in the Figure 4 data set! As a reviewer, I am highly skeptical of this ploy, which throws away a lot of*

data. The authors admit that T>C substitutions can be frequent (~40% in their WGS samples) and do not show consistent patterns. In other words, they don't know what to do with all the T>C substitutions. I also noticed that the proportions of the six substitution types are pretty much the same between unrepaired and repaired samples. That seems a surprising result because it implies that the repair cocktail reduced all mutation types on a roughly equal basis, which appears to contradict the two cited studies showing that C>T mutations are rectified. There's more going on here than the authors are wanting to tell in their story and there are parts that don't fit neatly together. All the more reason for them to redouble their investigations with an open mind so they can write a paper that enlightens more than it baffles.

Reply 2.6 We disagree with the statement that our investigation was not done with an open-minded attitude. However, we thank the reviewer for questioning the foundation of our study as this made our newly revised manuscript much stronger. We provide a full overview of mutational profiles of the studies including the discrepancy in T>C channels and justified the reasons why these mutations should not be included when establishing the FFPE-induced artefact signature (see also in Reply 2.1). We also summarised the reasons for observing a similar-repair-effect on all SBS in our WGS samples in Reply 1.9.

As we demonstrate in Table S1, we are not the only team to discover this discrepancy in FFPE samples, suggesting T>C mutations are caused by an alternative process. We believe these results together demonstrate that the inclusion of those large unexplained mutations is equivalent to introducing large unknown noise. Therefore, to derive the FFPE noise patterns, we chose to treat the T>C mutation count as missing data and assigned them with zero values, but we do not suggest that T>C mutations in general are not important biologically. The validity of our FFPE-derived signature, as well as the ambiguous nature of excess T>C is also supported by a third dataset newly added to the manuscript (Fig. S6d). We discussed the impact of these batch-related artefacts on our study aims in Reply 1.10.

In terms of scientific transparency, open data science with reproducible results is always one of the most important principles we insist on throughout all our scientific work. Therefore, we uploaded all analysis codes and source data we used in our study in Jupyter-Notebooks (<https://github.com/QingliGuo/FFPEsig>), which allows anyone to run the entire analysis across platforms. So, we are confident about our investigation results and our method.

References

1. Ding, J. *et al.* Feature-based classifiers for somatic mutation detection in tumour-normal paired sequencing data. *Bioinformatics* **28**, 167–175 (2012).
2. Pich, O. *et al.* The mutational footprints of cancer therapies. *Nat. Genet.* **51**, 1732–1740 (2019).
3. Rosenthal, R., McGranahan, N., Herrero, J., Taylor, B. S. & Swanton, C. deconstructSigs: delineating mutational processes in single tumors distinguishes DNA repair deficiencies and patterns of carcinoma evolution. *Genome Biol.* **17**, 31 (2016).
4. Jabbari, K. & Bernardi, G. Cytosine methylation and CpG, TpG (CpA) and TpA frequencies. *Gene* **333**, 143–149 (2004).
5. Xu, C., Nezami Ranjbar, M. R., Wu, Z., DiCarlo, J. & Wang, Y. Detecting very low allele fraction variants using targeted DNA sequencing and a novel molecular barcode-aware variant caller. *BMC Genomics* **18**, 5 (2017).
6. Xu, C. A review of somatic single nucleotide variant calling algorithms for next-generation sequencing data. *Comput. Struct. Biotechnol. J.* **16**, 15–24 (2018).
7. Do, H. & Dobrovic, A. Sequence artifacts in DNA from formalin-fixed tissues: Causes and strategies for minimization. *Clin. Chem.* **61**, 64–71 (2015).
8. Flores Bueso, Y., Walker, S. P. & Tangney, M. Characterization of FFPE-induced bacterial DNA damage and development of a repair method. *Biol Methods Protoc* **5**, bpaa015 (2020).
9. Chen, G., Mosier, S., Gocke, C. D., Lin, M. T. & Eshleman, J. R. Cytosine Deamination Is a Major Cause of Baseline Noise in Next-Generation Sequencing. *Mol. Diagn. Ther.* **18**, 587–593 (2014).
10. Williams, C. *et al.* A high frequency of sequence alterations is due to formalin fixation of archival specimens. *Am. J. Pathol.* **155**, 1467–1471 (1999).
11. Do, H., Wong, S. Q., Li, J. & Dobrovic, A. Reducing sequence artifacts in amplicon-based massively parallel sequencing of formalin-fixed paraffin-embedded DNA

- by enzymatic depletion of uracil-containing templates. *Clin. Chem.* **59**, 1376–1383 (2013).
12. Do, H. & Dobrovic, A. Dramatic reduction of sequence artefacts from DNA isolated from formalin-fixed cancer biopsies by treatment with uracil- DNA glycosylase. *Oncotarget* **3**, 546–558 (2012).
 13. Yost, S. E. *et al.* Identification of high-confidence somatic mutations in whole genome sequence of formalin-fixed breast cancer specimens. *Nucleic Acids Res.* **40**, e107 (2012).
 14. Spencer, D. H. *et al.* Comparison of clinical targeted next-generation sequence data from formalin-fixed and fresh-frozen tissue specimens. *J. Mol. Diagn.* **15**, 623–633 (2013).
 15. Oh, E. *et al.* Comparison of accuracy of whole-exome sequencing with formalin-fixed paraffin-embedded and fresh frozen tissue samples. *PLoS One* **10**, 1–13 (2015).
 16. Serizawa, M. *et al.* The efficacy of uracil DNA glycosylase pretreatment in amplicon-based massively parallel sequencing with DNA extracted from archived formalin-fixed paraffin-embedded esophageal cancer tissues. *Cancer Genet.* **208**, 415–427 (2015).
 17. Lin, M.-T. *et al.* Clinical validation of KRAS, BRAF, and EGFR mutation detection using next-generation sequencing. *Am. J. Clin. Pathol.* **141**, 856–866 (2014).
 18. Ofner, R. *et al.* Non-reproducible sequence artifacts in FFPE tissue: an experience report. *J. Cancer Res. Clin. Oncol.* **143**, 1199–1207 (2017).
 19. Gallegos Ruiz, M. I. *et al.* EGFR and K-ras mutation analysis in non-small cell lung cancer: Comparison of paraffin embedded versus frozen specimens. *Cell. Oncol.* **29**, 257–264 (2007).
 20. Sah, S. *et al.* Functional DNA quantification guides accurate next-generation sequencing mutation detection in formalin-fixed, paraffin-embedded tumor biopsies. *Genome Med.* **5**, 77 (2013).
 21. Alborelli, I. *et al.* Robust assessment of tumor mutational burden in cytological

- specimens from lung cancer patients. *Lung Cancer* **149**, 84–89 (2020).
22. Parker, J. D. K. *et al.* Fixation Effects on Variant Calling in a Clinical Resequencing Panel. *J. Mol. Diagn.* **21**, 705–717 (2019).
 23. Quach, N., Goodman, M. F. & Shibata, D. In vitro mutation artifacts after formalin fixation and error prone translesion synthesis during PCR. *BMC Clin. Pathol.* **4**, 1 (2004).
 24. Marchetti, A., Felicioni, L. & Buttitta, F. Assessing EGFR mutations. *The New England journal of medicine* vol. 354 526–8; author reply 526–8 (2006).
 25. Wong, C., DiCioccio, R. A., Allen, H. J., Werness, B. A. & Piver, M. S. Mutations in BRCA1 from fixed, paraffin-embedded tissue can be artifacts of preservation. *Cancer Genet. Cytogenet.* **107**, 21–27 (1998).
 26. Do, H. & Dobrovic, A. Limited copy number-high resolution melting (LCN-HRM) enables the detection and identification by sequencing of low level mutations in cancer biopsies. *Mol. Cancer* **8**, 82 (2009).
 27. Lamy, A. *et al.* Metastatic colorectal cancer KRAS genotyping in routine practice: results and pitfalls. *Mod. Pathol.* **24**, 1090–1100 (2011).
 28. Kunkel, T. A. DNA replication fidelity. *J. Biol. Chem.* **279**, 16895–16898 (2004).
 29. McInerney, P., Adams, P. & Hadi, M. Z. Error Rate Comparison during Polymerase Chain Reaction by DNA Polymerase. *Molecular Biology International* vol. 2014 1–8 (2014).
 30. Eckert, K. A. & Kunkel, T. A. DNA polymerase fidelity and the polymerase chain reaction. *PCR Methods Appl.* **1**, 17–24 (1991).
 31. Van Allen, E. M. *et al.* Whole-exome sequencing and clinical interpretation of formalin-fixed, paraffin-embedded tumor samples to guide precision cancer medicine. *Nat. Med.* **20**, 682–688 (2014).
 32. Dingler, F. A. *et al.* Two Aldehyde Clearance Systems Are Essential to Prevent Lethal Formaldehyde Accumulation in Mice and Humans. *Mol. Cell* **80**, 996–1012.e9 (2020).
 33. Kucab, J. E. *et al.* A Compendium of Mutational Signatures of Environmental Agents. *Cell* **177**, 821–836.e16 (2019).

REVIEWER COMMENTS

Reviewer #1 (Remarks to the Author):

I thank the authors for their considerable effort to address my concerns. I think the manuscript has improved substantially. However, I am unfortunately now somehow even slightly less convinced by the robustness of the results and also the proposed workflow for using the algorithm.

1.2. Even though the authors clarified that their algorithm should only be used after very stringent filtering of mutations by the mutation calling pipeline, this is another limitation of usability among several others (that I will also comment on below). When they applied this type of criterion to the data from study 1, this yielded an SNR range between 1 and 1/7 for samples fixed in formalin for less than 48 h. The authors did not specify the distribution across this range, but I assume 1/7 was observed for formalin fixation times of 24 h, which is not an uncommon choice. Hence, it is to be expected that many samples that the algorithm should be applied to fall near the SNR mark of 0.1, at which point, however, the performance is not very good (Fig. 2g,h; the cosine similarity is a quite lenient measure of accuracy, so 0.9 can still be relatively far off).

1.3. Expressing the results as a function of the signal-to-noise ratio instead of absolute mutation counts is a great improvement.

1.4. I appreciate the effort that the authors made in order to try and come up with some criteria to guide the use of FFPEsig.

However, I must admit that the criteria in the flowchart do not appear very convincing to me. Clearly, the flow chart as it stands should be at least amended with the decision regarding which FFPE mode to use ("repaired" vs "unrepaired") based on actual information about the repair status of the FFPE sample. In the absence of this information, the flowchart FFPE mode decision seems quite problematic, since the authors have not tested the performance of the algorithm when used with the "wrong" FFPE signature (i.e. "repaired" on unrepaired samples and vice versa).

Along these lines, how was this flowchart applied to the simulation data: for example, in the n=11 cases where the "repaired" mode was recommended (Fig. S10), was FFPEsig applied to simulations that had added noise from the "repaired" signature or were both "repaired" and "unrepaired" tested?

Regarding the criterion on SNR, the success of estimation of SNR by the proposed A/(O-A) method depends on the variance of the typical mutation load in normal tumors. This can render the assessment quite noisy for any individual sample from cancer types with high variance.

If a given sample then passes the SNR criterion and is assigned to the correct category of FFPE signature, the uncertainty about its prediction of signature weights still remains quite high (Fig. 2, Fig. 5, Fig. S19).

1.7. The analysis of false positives and false negatives is very informative.

1.8. The authors made a commendable effort to elucidate the distribution of concordant mutations across datasets. However, I do not think this can explain the problematic observations.

To be clear, I am not concerned about the higher rate of concordant mutations between repaired samples than between unrepaired samples, this makes sense. I am concerned about the high concordance between mutations from entirely different patients. To see a concordance of 20% between e.g. fresh frozen tumor from patient 1 and repaired FFPE15H from patient 3 should occur with a probability of practically zero. Yet a similar concordance is observed between that FF sample and repaired FFPE2H from patient 2 (and many other pairs show high values as well). The authors did not comment on this issue in their response, even though I had raised it explicitly. Clearly, it has confounding character for the inference of the FFPE-associated signatures, which are at the heart of the paper.

The very informative results from the analysis that the authors carried out actually confirm the suspicion about the dataset (which beyond that has the major discrepancies in the T>C mutation class, which are addressed by omitting this class entirely). While the "unrepaired" signature (Fig. 1d) shows a striking deficit in CpG>TpG mutations compared to other mutation types (already corrected for target size), the signature of mutations that are concordant between unrepaired samples shows higher similarity with the "repaired" signature, which has primarily CpG>TpG mutations. The explanations "So, we think FFPE noise is more prevalent at CpG sites (5mC), but more random at non-CpG sites." and "Given that 5mC is almost exclusively found at CpG sites, this finding implies that the C>T artefacts generated via formalin-induced deaminated-5mC are more likely to be detected in the sequence data." (l. 120) do not make any obvious sense to me. What do the authors mean to imply? Given the relative deficit of CpG mutations in the "unrepaired" signature, the other C>T mutation types should be much more likely to be concordant (yet also still much less concordant than 20%).

The argument about the methylation status of the genome could be playing somewhat of a role, however it could not explain the observation in pairs of unrepaired samples, nor the concordance between FF and FFPE samples from different patients. It could, however, lead to a mis-estimation of the

target size for CpG>TpG mutations by the amount of sites that are not methylated (as the authors estimate, between 20-30%). This could confound the signature inference, possibly giving too little relative weight to this mutation class.

Reviewer #2 (Remarks to the Author):

In this revision, Guo et al. have significantly clarified their arguments, adding more analyses, data, citations, etc. These additions are welcome, the authors' arguments are clearly laid out and some are persuasive.

However, there remain significant questions about the T>C mutations. The authors suggest that some unknown, ill-defined batch effect is responsible for T>C substitutions in some FFPE samples, but not others. The authors themselves observe 40% T>C in their own data (Supp. Fig. 13), which they do recognize as being inconsistent with other results. What I still find troubling is that these data are hand waved away and not being presented in a transparent way. It's as if the authors have convinced themselves that they must know what the right answer is, to the point that they are comfortable with throwing out nearly half of their own dataset because it doesn't fit their narrative. If one of my students ran an experiment and then threw away 40% of the data because it doesn't fit their favorite model, that would certainly set off alarm bells.

I am left wondering the same thing as last time: Why not simply report the data fully and let the reader make up their own mind? To spell this out more: Can the T>C mutations be attributable to mismatch repair deficiency (SBS21, SBS26)? If that's what happened with the CRC WGS, then that entire dataset is of questionable validity because significant fractions of the C>T mutations should also be attributed to other COSMIC signatures. The data should be reported fully so the reviewers can make a fully informed assessment.

We are delighted that both reviewers find our revised manuscript substantially improved and we remain very grateful for their thorough critique. We now clarified the parts that led to confusions to the reviewers. For the remaining issues, we conducted further analysis to validate the reliability of our findings and to address their concerns. We believe the current manuscript is further strengthened with these new revisions.

Table of Content

Comments from Reviewer #1

Reply-to-comment 1.0

Reply-to-comment 1.2

Reply-to-comment 1.3

Reply-to-comment 1.4

Reply 1.4.1

Reply 1.4.2

Reply 1.4.3

Reply 1.4.4

Reply-to-comment 1.7

Reply-to-comment 1.8

Reply 1.8.0

Reply 1.8.1 Concordant mutation pattern shows no impact on genomic FFPE signature inference using real and synthetic FFPE samples

Reply 1.8.2 Possible sources of where concordant mutations come from

Comments from Reviewer #2

Reply-to-comment 2.0

Reply-to-comment 2.1

References

Comments from Reviewer #1

Reply-to-comment 1.0

Comment 1.0 *I thank the authors for their considerable effort to address my concerns. I think the manuscript has improved substantially. However, I am unfortunately now somehow even slightly less convinced by the robustness of the results and also the proposed workflow for using the algorithm.*

Reply 1.0 We are glad that the reviewer finds our revised manuscript has been substantially improved. We carefully studied the reviewer's comments and we believe there were some misunderstandings that possibly caused the reviewer's concerns. We hope that our following replies clarify them, as well as address the remaining criticism.

Reply-to-comment 1.2

Comment 1.2. *Even though the authors clarified that their algorithm should only be used after very stringent filtering of mutations by the mutation calling pipeline, this is another limitation of usability among several others (that I will also comment on below). When they applied this type of criterion to the data from study 1, this yielded an SNR range between 1 and 1/7 for samples fixed in formalin for less than 48 h 1/7 was observed for formalin fixation times of 24 h. The authors did not specify the distribution across this range, but I assume 1/7 was observed for formalin fixation times of 24 h, which is not an uncommon choice. Hence, it is to be expected that many samples that the algorithm should be applied to fall near the SNR mark of 0.1, at which point, however, the performance is not very good (Fig. 2g,h; the cosine similarity is a quite lenient measure of accuracy, so 0.9 can still be relatively far off).*

Reply 1.2 We understand the reviewer's concern about the achievable SNR in real FFPE samples.

However, we would like to emphasise that the SNR depends strongly on how the derived variant calls are filtered. A higher SNR can be achieved by applying a more strict set of filters to eliminate more artefacts (e.g. allele frequency > X, probability of being somatic mutations > Z, etc.). Admittedly, more stringent filtering comes at the cost of potentially removing true somatic mutations. However, this is not a major concern for signature analysis, because the signal derives from the ensemble of mutations across the genome rather than any individual variant, and so the exclusion of a few true variants has minimal impact¹.

On the dataset of study 1, we applied a filter of selecting variants with probability of being somatic mutations >0.90. The averaged SNR is 0.5 (range 1/7~1) for n=21 FFPE samples. The lowest SNR of 1/7 is observed for one 48h-fixed repaired FFPE, one 24h-fixed unrepaired and one 15h-fixed unrepaired FFPE sample. Specimens with 24h fixation time had SNRs of 1/7, 1/5, 1/4, 1/3, 1/3 and 1/2. However, all above three cases (with SNRs of 1/7) would have an SNR>1/3 if a slightly more stringent filtering is applied only keeping

variants with a probability score >0.92 instead of >0.90 . Therefore we believe samples that have been fixed for up to 24 hours (which, as the reviewer points out, is probably not uncommon) are very likely to fall safely in the zone of well-reconstructable samples.

We note that in practice, wet-lab DNA quality control (fragment size analysis and PCR-ability) also plays an essential role in filtering out low DNA-yield FFPE samples due to over fixation or inappropriate storage². Thus very degraded samples are unlikely to feature in practice, or if they do, the researcher would know in advance which samples are of poor quality³.

Taken together, we think obtaining $SNR > 0.1$ and above is very feasible from typical formalin-fixed specimens (e.g. 12-24 hours of fixation period which is ideal for fixation of a 5-mm-thick specimen) with standard DNA quality control and downstream filterings. In the revised manuscript, we now discuss that SNR can be improved via adjusting upstream variant filtering, and advise future users to adjust filtering if they suspect their data might fall closer to the $SNR = 0.1$ limit (e.g. due to low mutation burden of the cancer type).

With regards to the appropriateness of cosine similarity (CS), we emphasise that CS is the widely accepted and used statistic to assess goodness of fit in mutational signature studies⁴⁻⁶. In our study, we applied a stringent/conservative accuracy measure (only on C>T channel), in which FFPEsig achieved an averaged CS of ~ 0.90 for the main simulation set, whereas the overall accuracy is ~ 0.95 when measured via 96-channel (Fig. S8). Furthermore, the accurate and robust decomposition results derived from those well-corrected mutational profiles continually highlight the outstanding performance of FFPEsig (Fig. 3, Fig. S9 & S10).

Reply-to-comment 1.3

Comment 1.3. *Expressing the results as a function of the signal-to-noise ratio instead of absolute mutation counts is a great improvement.*

Reply 1.3 We are grateful for the reviewer's suggestion.

Reply-to-comment 1.4

Comment 1.4.1 *I appreciate the effort that the authors made in order to try and come up with some criteria to guide the use of FFPEsig.*

However, I must admit that the criteria in the flowchart do not appear very convincing to me. Clearly, the flow chart as it stands should be at least amended with the decision regarding which FFPE mode to use ("repaired" vs "unrepaired") based on actual information about the repair status of the FFPE sample. In the absence of this information, the flowchart FFPE mode decision seems quite problematic, since the authors have not tested the performance of the algorithm when used with the "wrong" FFPE signature (i.e. "repaired" on unrepaired samples and vice versa).

Reply 1.4.1

We are glad the reviewer finds our flowchart useful, but with the benefit of hindsight we recognise that the flowchart was insufficiently structured, leading to a misunderstanding here. The aim of the flowchart was to guide the complete experimental workflow, both in the lab and the downstream bioinformatics analysis using our simulated data as a benchmark. Importantly this includes guiding the decision of whether or not chemical repair should be used in the laboratory analysis to maximise signal for downstream analysis of mutation signatures. We think that the revised flowchart makes this clear (Fig. S16).

Fig. S16 Recommended workflow for applying FFPEsig in FFPE samples. Our analysis identified two main factors that can affect noise correction accuracy of FFPEsig: 1) signal-to-noise similarity (SNS), and 2) signal-to-noise ratio (SNR). Here, we demonstrate how to implement this knowledge to improve FFPEsig performance for real FFPE samples. Firstly, we recommend the potential users to leverage prior knowledge from available fresh tumours whether a targeted cancer type X has more similar biological mutation patterns to the unrepaired ('UR') or repaired ('R') FFPE signatures (shown in the light-yellow shaded box). For example, if the general biological signal is more similar to the unrepaired signature, we suggest the potential users to repair the FFPE DNA (from sample Y) using UDG treatment. Thus, we can obtain an ideal treatment status for a sample Y in cancer type X. Secondly, we suggest the potential users to compute an approximate SNR for sample Y. Estimation of SNR can be obtained via ' $A/(O-A)$ ', where 'O' is the total number of observed FFPE mutations in FFPE sample Y and 'A' is the averaged mutation load of the non-hyper mutated tumours in the cancer type X (e.g. mutation load $< 5 \times 10^4$). Because FFPEsig performs well on hyper-mutated tumours

regardless of the noise level (Fig. 2g-h), and they can be identified without FFPE artefacts being removed due to their very high and distinctive mutation patterns, or via orthogonal methods.

Our analysis showed us that repaired and unrepaired samples generate markedly different error profiles (Fig. 1d & 1e). The similarity of these profiles to the true biological signatures determines whether artefacts from repaired or unrepaired FFPE can be best eliminated in the downstream bioinformatics analysis (Fig. 2f). Biological signals of course vary by cancer type (Fig. S17).

We now clarify the above points in the main text (lines 440-452). First, we advise future users to consult Fig. S16 & Fig. S17 when deciding whether to apply repair treatment or not. Tumour types where the true biological signature is distinct from the formalin signature are easiest to manage bioinformatically. When the predominant biological signature is ageing (SBS 1) then chemical repair is not advised as the magnitude of formalin artefacts cannot be distinguished from the burden of true ageing-associated mutations. We then advise them to compute the expected SNR given the observed mutations in the sample to decide if FFPEsig is likely to be successful.

For already existing data without direct/obvious chemical treatment status available, we suggest the users contact the original data generator or consult the manufacturing company using the DNA extraction kit provided in the original paper - this information is typically available. Furthermore, we can also gain some hints from the observed mutations in a given FFPE sample: 1) unrepaired FFPE samples typically have much more primary variants than repaired FFPEs; 2) $NCG > NTG$ sites are normally suppressed in mutation profiles of unrepaired FFPEs, but they are much more dominant in repaired FFPEs (Fig. 4a and Fig. S14). Nonetheless, the users should always apply the error signature corresponding to how the sequencing data was generated, and we advise the users against applying FFPEsig without a known repaired/unrepaired status. We have emphasised this in lines 454-457.

Comment 1.4.2 *Along these lines, how was this flowchart applied to the simulation data: for example, in the n=11 cases where the "repaired" mode was recommended (Fig. S10), was FFPEsig applied to simulations that had added noise from the "repaired" signature or were both "repaired" and "unrepaired" tested?*

Reply 1.4.2

FFPEsig was applied in repaired vs unrepaired mode in accordance with whether the simulated data was repaired or unrepaired. We did not consider mismatches between the application mode and real repair status of the sample for the reasons explained in response to the previous point in 1.4.1. Therefore, we collected FFPEsig correction results for the n=11 cases where the repaired noise was added.

Comment 1.4.3 *Regarding the criterion on SNR, the success of estimation of SNR by the proposed A/(O-A) method depends on the variance of the typical mutation load in normal*

tumors. This can render the assessment quite noisy for any individual sample from cancer types with high variance.

Reply 1.4.3

We argue that a subset of hypermutated tumours is the primary reason for a high variance in mutational load, which we agree is problematic for SNR calculation. We now propose to consider hypermutated tumour subtypes separately as their biological signal is so pronounced (Fig. R1). We are confident that hyper-mutated samples can be identified without FFPE artefacts being removed, due to their very high and distinctive mutation loads (In addition, hyper-mutator status, e.g. mismatch repair deficiency, can be derived via orthogonal methods⁷). We also note that FFPEsig performs well for the hypermutated tumours regardless of the noise levels (Fig. 2g & 2h).

Removing hypermutated tumours should significantly reduce the variation in mutation burden across the remaining non-hypermutant cases, making the SNR calculation reasonable. We now modified the calculation for our recommended application flowchart (Fig. S16).

Fig. R1 The hyper-mutated processes can be identified with a mixture of a large amount of FFPE noise ($\sim 10^5$). We showed two simulated FFPE samples originating from hyper-mutated CRCs: one POLE-CRC (a) and one MSI-CRC (b). The upper panel in (a) and (b) are the original biological profiles, whereas the lower panel are the simulated FFPE samples mixed with 10^5 unrepaired FFPE noise.

Comment 1.4.4 If a given sample then passes the SNR criterion and is assigned to the correct category of FFPE signature, the uncertainty about its prediction of signature weights still remains quite high (Fig. 2, Fig. 5, Fig. S19).

Reply 1.4.4

We politely disagree with this comment as our results clearly demonstrate that signature weights prediction from corrected profiles are accurate, robust and most importantly substantially improved compared to the uncorrected profiles. Those derived weights were

compared against the ground-truth contributions, and the results are shown in Fig. 3, Fig. S9 & S10. We introduced ‘ ϵ ’ and ‘ r ’, which represent the absolute difference between the inferred and true weights, and Spearman’s rho correlation coefficient, respectively. We observed gross misassignment while using uncorrected profiles ($\epsilon \sim 0.19$, $r \sim 0.44$ for unrepaired; $\epsilon \sim 0.14$, $r \sim 0.52$ for repaired-FFPEs). Whereas the corrected FFPE profiles recovered true activities with almost negligible error ($\epsilon \sim 0.04$, $r \sim 0.96$ for unrepaired-FFPEs; $\epsilon \sim 0.03$, $r \sim 0.97$ for repaired-FFPEs). Therefore, the correction by FFPEsig has provided a robust and accurate signature weights prediction.

It is true that we cannot completely recover the original signature weights, as is often the case in general decomposition tasks, especially for highly noisy data, however we argue that the achieved improvement over uncorrected data is substantial. We believe the “uncertain/unstable” assignment mostly happened among co-presented highly similar signatures (SBS5 and SBS40), which both possess a “featureless” nature (Fig. 5, Fig. S18 & S19). However, it is problematic to recover their contributions even when using 96-channels^{6,8}, as more data is required to separate them into more distinguishable patterns.

Reply-to-comment 1.7

Comment 1.7. The analysis of false positives and false negatives is very informative.

Reply 1.7. We thank the reviewer for their suggestion.

Reply-to-comment 1.8

Comment 1.8 *The authors made a commendable effort to elucidate the distribution of concordant mutations across datasets. However, I do not think this can explain the problematic observations.*

To be clear, I am not concerned about the higher rate of concordant mutations between repaired samples than between unrepaired samples, this makes sense. I am concerned about the high concordance between mutations from entirely different patients. To see a concordance of 20% between e.g. fresh frozen tumor from patient 1 and repaired FFPE15H from patient 3 should occur with a probability of practically zero. Yet a similar concordance is observed between that FF sample and repaired FFPE2H from patient 2 (and many other pairs show high values as well). The authors did not comment on this issue in their response, even though I had raised it explicitly. Clearly, it has confounding character for the inference of the FFPE-associated signatures, which are at the heart of the paper.

The very informative results from the analysis that the authors carried out actually confirm the suspicion about the dataset (which beyond that has the major discrepancies in the T>C mutation class, which are addressed by omitting this class entirely). While the “unrepaired” signature (Fig. 1d) shows a striking deficit in CpG>TpG mutations compared to other mutation types (already corrected for target size), the signature of mutations that are concordant between unrepaired samples shows higher similarity with the “repaired” signature, which has primarily CpG>TpG mutations. The explanations “So, we think FFPE

noise is more prevalent at CpG sites (5mC), but more random at non-CpG sites." and "Given that 5mC is almost exclusively found at CpG sites, this finding implies that the C>T artefacts generated via formalin-induced deaminated-5mC are more likely to be detected in the sequence data." (l. 120) do not make any obvious sense to me. What do the authors mean to imply? Given the relative deficit of CpG mutations in the "unrepaired" signature, the other C>T mutation types should be much more likely to be concordant (yet also still much less concordant than 20%).

The argument about the methylation status of the genome could be playing somewhat of a role, however it could not explain the observation in pairs of unrepaired samples, nor the concordance between FF and FFPE samples from different patients. It could, however, lead to a mis-estimation of the target size for CpG>TpG mutations by the amount of sites that are not methylated (as the authors estimate, between 20-30%). This could confound the signature inference, possibly giving too little relative weight to this mutation class.

Reply 1.8.0

We share the reviewer's interest in concordant mutations between samples. The data are not in agreement with our preconceived notion that every mutation is essentially unique (the infinite sites hypothesis). The explanation for concordant mutations can only be that true biological mutations, false somatic-mutation calls including unfiltered germline SNPs, and/or artifactual mutations induced by formalin and lab processing, occur in a biased fashion⁹⁻¹⁴.

The reviewer is concerned that concordant mutations negatively impact our ability to resolve and correct formalin induced mutation signatures. Below we show definitively that this is not the case: our findings are strongly robust to concordant mutations.

Pinpointing the mechanism that generates these concordant mutations is very interesting, but we emphasise that it is tangential to this manuscript. Nevertheless, for scientific interest, for the reviewer only, we have conducted further analysis that demonstrates recurrent mutations could come from unfiltered SNPs, recurrent system errors and plausible 'sampling bias' towards GC-rich templates in FFPE samples.

Reply 1.8.1 Concordant mutation pattern shows no impact on genomic FFPE signature inference using real and synthetic FFPE samples

To address the robustness of our derived signatures, we derived two distinct sets of FFPE signatures using real FFPE samples from studies 1 and 2, respectively (Fig. S6d-e). Firstly, we applied extra filters on the entire dataset to exclude any possible non-FFPE mutations that were also detected in other fresh samples (equivalent to a small "panel-of-normal") or in the known germline SNP databases (Table R1). Adding this step is based on our analysis results shown in reply 1.8.2.

Our re-inference results produced two sets of almost identical FFPE signatures (CS=0.99 for unrepaired signatures; CS=0.98 for repaired signatures). Since the two studies were from different laboratories and analysed with different pipelines, the results highlighted the reliability of the derived signatures we had previously derived using study 1 & 2. We now added the independent inference results in our manuscript (lines 200-202).

Fig S6d-e. Almost identical FFPE signatures generated from independent analysis using real FFPE samples from study 1 and 2. We applied the same methodology described in our “Methods” for deriving FFPE signature from study 1 ($n=102$ FFPE samples). We took the averaged mutation profiles from study 2 to obtain the signatures ($n=8$ FFPE samples). The two sets of signatures are highly similar to each other. (d) Unrepaired FFPE signatures. (e) Repaired FFPE signatures.

We think it is an interesting observation that the pattern of shared mutations from unrepaired pairs is more similar to the repaired signature in study 1, and that the concordant ratio ($\sim 10\%$) is much smaller than that of repaired pairs ($\sim 25\%$). A plausible explanation is that there is a ‘sampling bias’ during library preparation (PCR) and sequencing that causes methylated CpG sites (GC-rich templates; Fig. S3) to become overrepresented in the final sequencing reads. In comparison, the nonCpG sites (uracil-carrying templates; Fig. S3) were under-represented as they may suffer a higher formalin-induced denaturation rate, and/or a lower amplification rate during PCR given the incorporation of U-A pair may be not as efficient as T-A pair. In turn, this will lead to more concordant mutation calls coming from CpG sites (see our detailed discussion about the ‘sampling bias’ in 1.8.2 below).

To understand the effect of concordant mutation on signature inference, we simulated two sets of FFPE samples introduced with or without sampling bias towards CpG sites. We first assume the initial formalin-induced deamination randomly targeted both CpG and nonCpG sites. To simulate samples with bias, we introduced a higher sequencing probability towards CpG sites, whereas we assigned the same probability for both site types for simulations that did not model bias. We annotated our simulation process which can be found via this link https://qingliguo.github.io/FFPEsig/simulation_code_for_reply1.8.html.

We examined the concordant ratio as well as the concordant mutation pattern in unrepaired pairs (Fig. R2a-b). Our observations on the simulated samples with bias is highly similar to study 1. The concordant ratio of CpG sites ($\sim 25\%$) is much higher than that of nonCpG sites ($\sim 8.5\%$) and the mutation pattern is more similar to repaired signature (CS=0.91), instead of unrepaired signature (CS=0.71). In contrast, for samples without bias, concordant ratios are roughly the same for both sites ($\sim 12.6\%$) and the concordant mutation pattern is almost identical to the unrepaired signature.

Fig. R2 Nearly-identical FFPE patterns are derived from synthetic samples with different concontant mutation patterns. Here, we generated two sets of FFPE samples, and each set has $n=100$ samples. We applied the same sampling probability (~ 0.5) for CpG and nonCpG sites to simulate samples without bias, but a higher sampling probability (~ 0.9) is assigned to CpG sites compared to nonCpG sites (~ 0.35) for samples with bias. We then randomly draw artefacts for CpG and nonCpG sites, respectively. Next, we assigned CpG artefacts according to the normalised distribution of CpG channels in FFPE signatures (Fig. 1d& 1e) (the same method applied for nonCpG artefacts). We then added them up as one synthetic sample. The signature is derived by taking the averaged mutation count from $n=100$ samples in each set. (a) Concordant ratios for both sites in unrepaired sample pairs. (b) Comparison of concordant mutation profiles from unrepaired pairs to both FFPE samples. (c-d) Highly similar FFPE signatures are derived from unrepaired (c) and repaired FFPEs (d) samples with or without sampling bias.

Next, we derived two pairs of FFPE signatures from above two sets of simulated samples, by normalising the averaged counts of all samples within each set. The signature pairs were near-perfectly mirrored with each other ($CS > 0.99$ for both) (Fig. R2c-d). We note that the total CpG candidates (including unmethylated ones) only account for $\sim 4\%$ of the C>T candidates in the human genome, whereas $\sim 96\%$ of them are nonCpG sites (Fig. S5c). Such a small fraction ($\sim 4\%$) leads to negligible difference of mutation count in biased and unbiased samples. Therefore, our FFPE signature inference is highly stable regardless of the presence of sampling bias.

In practice, different FFPE DNA extraction methods may introduce varied sampling biases. Therefore, deriving a generalised FFPE signature requires a joint inference from independent datasets, which is exactly what we offered in our study. We now updated the FFPE signatures using the joint datasets (extra-filtered) from both studies. We found that the revised signatures share nearly the same pattern with the previous version ($CS = 0.99$ for unrepaired and $CS = 0.98$ for repaired signatures). We note that the re-inferred unrepaired signature shows slightly higher similarity to SBS30 ($CS = 0.92$) compared to the old version ($CS = 0.90$). The other results using updated signatures maintain the same except some subtle changes of several statics (now highlighted in blue colour in our revised manuscript).

Reply 1.8.2 Possible sources of where concordant mutations come from

Below, we outline some plausible mechanisms for the origin of concordant mutations between FFPE samples, as well as between fresh (FF) and FFPE samples.

It is well established that a significant number of DNA templates are depleted due to additional DNA damage, e.g. denaturation, crosslinks *etc.*^{15,16}. Given the tissue was penetrated by a high-dose formaldehyde for at least a few hours, we argue that the initial formalin-induced deamination is significantly more common than what we observed from real sequence data (from only amplifiable templates). Therefore, the chance of observing concordant mutations is very likely.

We showed that a sampling bias towards GC-rich templates can cause a similar observation in study 1. Indeed, such a GC-bias is also discovered in other FFPE samples in previous studies¹⁷⁻¹⁹. Mutter and Boynton found that insufficient and damaged DNA display amplification bias of GC-rich PCR targets, and they concluded that “the secondary structure of the PCR template is a probable major determination for the observed effect”²⁰. Furthermore, archaeal family-B DNA polymerases (e.g. Vent, *Pfu*) can detect pro-mutagenic templates containing uracil bases, and this function can limit the amplification of uracil-containing templates^{21,22}. Consequently, different denaturation rates, amplification rates, and other possible sources could contribute to a joint sampling bias as we showed in above simulation.

Next, we took all concordant variants in FF & FFPE pairs from different patients, which yielded 78 unique mutations in total. We queried these mutations against five known germline datasets in a sequential order using ANNOVAR²³ (Table R1). 43 out of 78 (~55%) mutations were reported in these known databases, confirming that a majority of these shared mutations of FF-FFPE are unfiltered SNPs or recurrent sequencing errors that would be picked up at germline sequencing¹⁴.

Table R1 Majority of the concordant variants in FF-FFPE pairs are in known germline databases.

Query order	Databases and version number	Number of identified SNPs
1	1000 genome-2015-Apr	7
2	esp6500siv2_all	3
3	exac03nontcga	20
4	gnomad211_genome	2
5	kaviar_20150923	11
Total		43

In study 1, the averaged concordant ratio of FF-FFPE pairs is ~5.5% (15 out of 278), and a very few true somatic mutations (≤ 3) are not able to cause an observant difference between inter- and intra- patient samples. The reviewer also noted that the concordant ratio is higher for some FF & repaired-FFPE pairs (15/155= $\sim 9\%$; not $\sim 20\%$), which is due to their smaller total number of mutations (used as ‘denominator’). In contrast, FF & unrepaired-FFPE pairs hold a relatively smaller concordant ratio (15/401= $\sim 3.8\%$) due to the higher “denominator”.

The reason why we observe those unfiltered SNPs and recurrent errors between all samples is that the concordant ratio in Fig. S4b was derived from completely raw candidate variants (Fig. R3). In this candidate list, a probability score of being somatic mutation is assigned to each candidate by a pre-trained binary classification model²⁴. Instead, other mutation callers (e.g. Mutect2) provide specific annotations, e.g. ‘germline-risk’, ‘panel-of-normal’ etc.¹⁰. Therefore, the candidate variants not only include a few true somatic mutations, but most of them actually are technical errors, unfiltered SNPs and/or FFPE artefacts.

Fig. R3 Schematic figure shows the primary variant candidates used for Fig. S4b. The primary list contains all candidates somatic mutations, except mutations detected in matched normal samples. We added ‘mutect2 filters’ to show the differences of the filtering of two mutation calling pipelines.

In addition, other technical biases (not captured in the above databases) can also give rise to shared variants. The recurrent systematic errors can arise from sample preparation, as well as analysis pipelines. For example, DNA handling, polymerase errors, PCR cycles, as well as sequencing machines are known to introduce biased errors in specific sequence contexts^{11,12}. Furthermore, some low complexity genomic regions are prone to mapping error or systematic errors in all sequencing chemistries¹³.

We would like to thank the reviewer’s comment as we are now reassured about the robustness of the derived signatures, which certainly strengthened our manuscript.

Comments from Reviewer #2

Reply-to-comment 2.0

Comment 2.0 *In this revision, Guo et al. have significantly clarified their arguments, adding more analyses, data, citations, etc. These additions are welcome, the authors' arguments are clearly laid out and some are persuasive.*

Reply 2.0 We are delighted that the reviewer thinks our revised manuscript has been significantly improved by our extra analysis.

Reply-to-comment 2.1

Comment 2.1 *However, there remain significant questions about the T>C mutations. The authors suggest that some unknown, ill-defined batch effect is responsible for T>C substitutions in some FFPE samples, but not others. The authors themselves observe 40% T>C in their own data (Supp. Fig. 13), which they do recognize as being inconsistent with other results. What I still find troubling is that these data are hand waved away and not being presented in a transparent way. It's as if the authors have convinced themselves that they must know what the right answer is, to the point that they are comfortable with throwing out nearly half of their own dataset because it doesn't fit their narrative. If one of my students ran an experiment and then threw away 40% of the data because it doesn't fit their favorite model, that would certainly set off alarm bells.*

I am left wondering the same thing as last time: Why not simply report the data fully and let the reader make up their own mind? To spell this out more: Can the T>C mutations be attributable to mismatch repair deficiency (SBS21, SBS26)? If that's what happened with the CRC WGS, then that entire dataset is of questionable validity because significant fractions of the C>T mutations should also be attributed to other COSMIC signatures. The data should be reported fully so the reviewers can make a fully informed assessment.

Reply 2.1 We totally agree that data should be fully disclosed for reproducibility and further studies. We had previously made the 96-channel counts of WGS-CRCs publicly available from https://github.com/QingliGuo/FFPEsig/blob/main/Data/WGS_FFPEsamples.SBS96.csv (including T>C channels). On top of that, we now made a simple version of all variants (https://github.com/QingliGuo/FFPEsig/blob/main/Data/repaired_CRC_variant_simple.vcf, https://github.com/QingliGuo/FFPEsig/blob/main/Data/unrepaired_CRC_variant_simple.vcf). The full version of VCF files are submitted to the EGA database with accession number EGAS00001005331.

Our investigation on the 20 studies overwhelmingly pointed out that the large T>C mutations are caused by an ill-defined batch effect (Table S1). In the case study of WGS-CRCs, we aimed to show the correction ability of FFPEsig on deaminated artefacts, which are dominantly present in C>T channels. Therefore, removing T>C channels does not impact this purpose at all. We only removed the T>C channels in our figures to provide the potential

readers a better visualisation of this correction effect (Fig. 4a & Fig. S14), which is masked by unknown-noise from T>C channels otherwise. We now explicitly included reference to the whole (96-channel) data in the figure legends to highlight this further.

The reviewer suggested that T>C mutations in our WGS-CRC samples might be caused by MMR deficiency (MMRd), but our analysis shows that is extremely unlikely to be the case:

- MMRd could cause higher mutation load, but not in T>C channels. MMRd-attributed mutations are preferably distributed in C>A and C>T channels based on MSI signatures (e.g. SBS44, SBS6 *etc.*) in CRCs⁶.
- We applied orthogonal analysis using the MSIsensor tool to check the MSI status²⁵. In total, 0.23% of the microsatellite (repeat) loci was detected in our repaired CRC, implying the sample is not MMRd CRC.
- In comparison with PCAWG-CRCs, the C>A and/or C>T mutation pattern in FFPE tumours further confirmed our CRC sample is a MSS CRC (Fig. 4b).

Therefore, we maintain that the large T>C is not of biological origin but batch-related artefacts, which is also observed in other few studies (Table S1).

We think the discovery of batch-related T>C is very valuable information. Given the increasing number of studies focusing on actionable mutations within targeted regions in FFPE specimens, a similar observation of large T>C mutations is an extremely important alert to clinicians/researchers to make any decision using the data.

References

1. Perner, J. *et al.* The mutREAD method detects mutational signatures from low quantities of cancer DNA. *Nat. Commun.* **11**, 3166 (2020).
2. Nakhleh, R. E. & Nowak, J. A. Mining Formalin-Fixed, Paraffin-Embedded Tissues: A Wealth of Knowledge or Fool's Gold? *Archives of Pathology & Laboratory Medicine* vol. 138 1426–1427 (2014).
3. Robbe, P. *et al.* Clinical whole-genome sequencing from routine formalin-fixed, paraffin-embedded specimens: pilot study for the 100,000 Genomes Project. *Genet. Med.* **20**, 1196–1205 (2018).
4. Nik-Zainal, S. *et al.* Mutational processes molding the genomes of 21 breast cancers. *Cell* **149**, 979–993 (2012).
5. Alexandrov, L. B. *et al.* Signatures of mutational processes in human cancer. *Nature* **500**, 415–421 (2013).
6. Alexandrov, L. B. *et al.* The repertoire of mutational signatures in human cancer. *Nature* **578**, 94–101 (2020).

7. Li, K., Luo, H., Huang, L., Luo, H. & Zhu, X. Microsatellite instability: a review of what the oncologist should know. *Cancer Cell Int.* **20**, 16 (2020).
8. Maura, F. *et al.* A practical guide for mutational signature analysis in hematological malignancies. *Nat. Commun.* **10**, 2969 (2019).
9. Demeulemeester, J., D'Entropio, S. C., Gerstung, M. & Van Loo, P. Biallelic mutations in cancer genomes reveal local mutational determinants. *Nat. Genet.* **54**, 128–133 (2022).
10. Van der Auwera, G. A. & O'Connor, B. D. *Genomics in the Cloud: Using Docker, GATK, and WDL in Terra.* ('O'Reilly Media, Inc.', 2020).
11. Ma, X. *et al.* Analysis of error profiles in deep next-generation sequencing data. *Genome Biol.* **20**, 50 (2019).
12. Magi, A., Giusti, B. & Tattini, L. Characterization of MinION nanopore data for resequencing analyses. *Brief. Bioinform.* **18**, 940–953 (2017).
13. Goldfeder, R. L. *et al.* Medical implications of technical accuracy in genome sequencing. *Genome Med.* **8**, 24 (2016).
14. Salk, J. J., Schmitt, M. W. & Loeb, L. A. Enhancing the accuracy of next-generation sequencing for detecting rare and subclonal mutations. *Nat. Rev. Genet.* **19**, 269–285 (2018).
15. Bhagwate, A. V. *et al.* Bioinformatics and DNA-extraction strategies to reliably detect genetic variants from FFPE breast tissue samples. *BMC Genomics* **20**, 1–10 (2019).
16. Wong, S. Q. *et al.* Sequence artefacts in a prospective series of formalin-fixed tumours tested for mutations in hotspot regions by massively parallel sequencing. *BMC Med. Genomics* **7**, (2014).
17. Haile, S. *et al.* Sources of erroneous sequences and artifact chimeric reads in next generation sequencing of genomic DNA from formalin-fixed paraffin-embedded samples. *Nucleic Acids Res.* **47**, 1–10 (2019).
18. Graw, S. *et al.* Robust gene expression and mutation analyses of RNA-sequencing of formalin-fixed diagnostic tumor samples. *Sci. Rep.* **5**, 12335 (2015).
19. Newton, Y. *et al.* Large scale, robust, and accurate whole transcriptome profiling from clinical formalin-fixed paraffin-embedded samples. *Sci. Rep.* **10**, 17597 (2020).
20. Mutter, G. L. & Boynton, K. A. PCR bias in amplification of androgen receptor alleles, a trinucleotide repeat marker used in clonality studies. *Nucleic Acids Res.* **23**, 1411–1418 (1995).
21. Fogg, M. J., Pearl, L. H. & Connolly, B. A. Structural basis for uracil recognition by archaeal family

- B DNA polymerases. *Nat. Struct. Biol.* **9**, 922–927 (2002).
22. Connolly, B. A., Fogg, M. J., Shuttleworth, G. & Wilson, B. T. Uracil recognition by archaeal family B DNA polymerases. *Biochem. Soc. Trans.* **31**, 699–702 (2003).
23. Wang, K., Li, M. & Hakonarson, H. ANNOVAR: functional annotation of genetic variants from high-throughput sequencing data. *Nucleic Acids Res.* **38**, e164 (2010).
24. Ding, J. *et al.* Feature-based classifiers for somatic mutation detection in tumour-normal paired sequencing data. *Bioinformatics* **28**, 167–175 (2012).
25. Niu, B. *et al.* BIOINFORMATICS APPLICATIONS NOTE Sequence analysis MSIsensor : microsatellite instability detection using paired tumor-normal sequence data. **30**, 1015–1016 (2014).

REVIEWER COMMENTS

Reviewer #1 (Remarks to the Author):

My main remaining point of criticism regarded the effects of the many unexplained concordant mutations (Fig. S4) on the FFPE signature estimation. The authors addressed this point by deriving the unrepaired and repaired FFPE signatures separately from each of the studies, which yielded similar signature profiles. The authors further showed that unfiltered germline SNPs explain about half of the mutation overlap between fresh frozen and FFPE samples. Together, I think this is good evidence that the FFPE signatures are reliable despite the putatively high contributions from artifactual variants.

The argument about the SNR is satisfactory and with the caveats now more clearly stated in the recommended application flowchart, I think the method will be used in the right way and hopefully useful to many researchers with FFPE samples ready to be analyzed.

I wish the authors many users and their algorithm a lot of success!

Reviewer #2 (Remarks to the Author):

In this revision, the authors still have not addressed the main issue that I have with this manuscript, namely that they don't have a good understanding of the origins of T>C substitutions, which make up significant fractions of mutations in their own samples. Fraction of T>C for these samples:

53C1_repaired, $10977/36140 = 0.3037$

53C2_repaired (shouldn't one of the 53C samples be unrepaired?), $20311/51270 = 0.3962$

C277_1_unrepaired, $57628/145167 = 0.3970$

C277_2_repaired, $34362/90222 = 0.3809$

In effect, the authors are trying to redact and handwave away 30 to 40% of their own sequencing data because they don't know what to do with it. They went to extraordinary effort to try to convince readers that T>C mutations don't matter for their analyses. It's easy to surmise there's something in the T>C data that isn't straightforward to explain. If it were straightforward, the authors would have simply reported in a transparent way instead of trying to hide it.

Since the authors still refuse to plot these data in the 96 channel format, I had to plot them and see for myself. For the C277 samples, the T>C channels were quite similar, and that's fine.

The baffling results are from the 53C samples, there are distinct peaks at each of the NTG channels in the T>C set. I am wondering if one of these samples is mislabeled, as they are both called repaired when I think the point is to compare repaired vs. unrepaired. Presumably, 53C2 is the unrepaired because it has more mutations. In any case, if we look at the difference between the 53C1 and 53C2 profiles, the main differences are in the T>C channels, which is contrary to the claim that T>C channels are irrelevant for the analyses. If we compare the 53C samples to the C277 samples, again it is the T>C channels that are the most different.

Instead of being forthright and transparent to report that the T>C channels have high variability, the authors somehow decided it's a better idea to throw out data that are inconvenient to explain. To describe this variability, they appeal to this amorphous term "batch effects" which appears synonymous with "we don't really know what's going on". They then came up with computations to correct the COSMIC signatures for the purported FFPE effects.

Assuming for the sake of argument that there should be some correction applied to compensate for FFPE-associated mutations, how could that correction possibly be valid if it ignores entirely the contribution from the most frequent base substitution type? In my opinion, it would not be a valid procedure and instead, would have the counterproductive effect of applying a "correction" which is only partial and does not actually correspond to the complete effects of FFPE exposure. Mutational signature analyses should not be revised wholesale in a dubious, incomplete way based on a dodgy understanding of how to compute a reasonable correction due to mysterious "batch effects". Instead of bringing clarity to answer a perfectly valid scientific question, it would just muddy the waters.

Reviewer #3 (Remarks to the Author): Expert in mutational processes and cancer genomics

After reading the revised manuscript, and the comments by Reviewer 2 (both initial ones, and the second iteration thereof), I would tend to largely agree with concerns of Reviewer 2. The T>C artefacts, while not present in all the FFPE datasets, are present in multiple datasets including the authors' own data, which means they cannot be ignored.

The authors explanation that the T>C is not due to formalin itself (but due to another unknown thing during sample prep/library prep of FFPE samples) is plausible, but does not really justify not considering these mutations in the spectrum adjustment. The suggested approach of ignoring the entire T>C part of the mutation spectrum (16 out of 96 channels, as in Fig 4 and 5) is perhaps servicable, even though this will prevent measuring some signatures e.g. SBS12 or SBS16 or reduce accuracy for other signatures. I think their overall method would (in the lack of better solution!) need to always include this step of omitting T>C mutations when generating the adjusted spectrum, since it is not in advance known if a FFPE sample will contain these T>C artefacts. It should be presented as a method that generates an 80-channel (not 96-channel) mutational spectrum, in the current state.

In the manuscript they should be transparent about amounts of T>C mutations in some FFPE samples (also in the figures - e.g. Fig 1 should probably have an equivalent of 1E and 1D for T>Cs, even if the 'known' signature for the T>Cs cannot be recovered). They should not dismiss the importance of these mutations in the text, since they can be a problem to a similar extent as the C>T mutations. They should be clear that a recommended use of their method/workflow is to fully ignore the T>C mutations, which will reconstruct only partial mutation spectra (however I do recognize that this is probably still better than using the uncorrected spectrum). Their synthetic bechmarks, which currently I think do not spike-in the T>C mutations, should therefore also ignore the 16 channels of the T>C spectra (and the T>C box on e.g. Fig 2 would need to be greyed out) and report results based on that, if this is not currently done.

We thank the reviewers for their continued interest and critique of our work. In response to reviewer #3's helpful suggestions, we now presented our method using the adjusted 80-channel mutation spectrum throughout. We also textually explain the reasons for omitting the unexplained T>C features in the context of signature analysis from a data science perspective. We hope our revisions are sufficient to address the reviewers' concerns.

Table of Content

Comments from Reviewer #1	1
Comments from Reviewer #2	2
Reply-to-comment 2.0	2
Reply-to-comment 2.1	2
Reply-to-comment 2.2	4
Comments from Reviewer #3	6
Reply-to-comment 3.0	6
Reply-to-comment 3.1	6
Reply-to-comment 3.2	8

Comments from Reviewer #1

Comment 1.0 *My main remaining point of criticism regarded the effects of the many unexplained concordant mutations (Fig. S4) on the FFPE signature estimation. The authors addressed this point by deriving the unrepaired and repaired FFPE signatures separately from each of the studies, which yielded similar signature profiles. The authors further showed that unfiltered germline SNPs explain about half of the mutation overlap between fresh frozen and FFPE samples. Together, I think this is good evidence that the FFPE signatures are reliable despite the putatively high contributions from artifactual variants.*

The argument about the SNR is satisfactory and with the caveats now more clearly stated in the recommended application flowchart, I think the method will be used in the right way and hopefully useful to many researchers with FFPE samples ready to be analyzed.

I wish the authors many users and their algorithm a lot of success!

Reply 1.0 We are delighted that the reviewer found our reply satisfactorily addressed their remaining concerns. We thank the reviewer for their insightful comments that have helped us to improve the work.

Comments from Reviewer #2

Reply-to-comment 2.0

Comment 2.0 *In this revision, the authors still have not addressed the main issue that I have with this manuscript, namely that they don't have a good understanding of the origins of T>C substitutions, which make up significant fractions of mutations in their own samples.*

Reply to 2.0 We appreciate the reviewer's continued interest in the T>C mutations. The reviewer is correct that we do not have a good understanding of the origins of these mutations. However, we are sure - and believe that the other reviewers concur - that the T>C mutations are not a consequence of formalin exposure but instead a consequence of an ill-defined batch effect during library preparation.

In response to the reviewer's early comments, we had performed an extensive literature review (N=20 papers; Table S1), that shows that abundant T>C mutations are only rarely reported to be consequence of FFPE (3/20 papers; 15%) whereas C>T mutations are always reported (20/20 papers; 100%). Because these studies span cancer types, methodologies and laboratories, we consider this provides very strong evidence that T>C mutations are not a consequence of formalin exposure. Instead we suspect they are a consequence of the choice of polymerase and/or the reaction conditions used in PCR which certainly differ between studies, and/or other sampling processing differences including the bioinformatics analysis. This evidence was discussed in the main manuscript at lines 144-151, 223-228, and 607-623 (Table S2).

We also emphasise that our discovery of the formalin-exposure associated signatures was performed on two distinct datasets and then verified in a third independent dataset (Fig. S6f; lines 214-221). This "training-test" approach is the gold standard for statistical discovery methods which we think offers rigorous evidence that only C>T mutations are predominantly associated with formalin.

We emphasise that the primary aim of our study was to characterise mutational artefacts that originate from formalin treatment. Our analysis shows that C>T, not T>C mutations, arise from formalin exposure. We understand, and indeed share, the reviewer's interest in T>C variants but reiterate that thoroughly characterising their origin is tangential to the aim of our study.

Reply-to-comment 2.1

Comment 2.1 *Fraction of T>C for these samples:*

53C1_repaired, 10977/36140 = 0.3037

*53C2_repaired (shouldn't one of the 53C samples be unrepaired?),
20311/51270 = 0.3962*

C277_1_unrepaired, 57628/145167 = 0.3970

C277_2_repaired, 34362/90222 = 0.3809

In effect, the authors are trying to redact and handwave away 30 to 40% of their own sequencing data because they don't know what to do with it. They went to extraordinary effort to try to convince readers that T>C mutations don't matter for their analyses. It's easy to surmise there's something in the T>C data that isn't straightforward to explain. If it were straightforward, the authors would have simply reported in a transparent way instead of trying to hide it.

Since the authors still refuse to plot these data in the 96 channel format, I had to plot them and see for myself. For the C277 samples, the T>C channels were quite similar, and that's fine.

The baffling results are from the 53C samples, there are distinct peaks at each of the NTG channels in the T>C set. I am wondering if one of these samples is mislabeled, as they are both called repaired when I think the point is to compare repaired vs. unrepaired. Presumably, 53C2 is the unrepaired because it has more mutations. In any case, if we look at the difference between the 53C1 and 53C2 profiles, the main differences are in the T>C channels, which is contrary to the claim that T>C channels are irrelevant for the analyses. If we compare the 53C samples to the C277 samples, again it is the T>C channels that are the most different.

Instead of being forthright and transparent to report that the T>C channels have high variability, the authors somehow decided it's a better idea to throw out data that are inconvenient to explain. To describe this variability, they appeal to this amorphous term "batch effects" which appears synonymous with "we don't really know what's going on". They then came up with computations to correct the COSMIC signatures for the purported FFPE effects.

Reply to 2.1 We thank the reviewer for the care and detail with which they have assessed our work.

We emphasise that our decision to remove T>C mutations is based on:

- Our rigorous analysis showing that only C>T mutations are predominantly associated with formalin exposure (the first two result sections of the manuscript discuss this);
- Our analysis showing that removal of T>C mutations does not negatively impact mutational signature assessment (a large portion of the second half of our manuscript demonstrates this).

We strongly disagree with the reviewer's suggestion that we are trying to hide inconvenient data: the exact opposite is true. Our decision to remove T>C mutations from the analysis is clearly emphasised throughout the main manuscript where we have also provided a full justification for their removal that includes a quantitative analysis. In detail:

- We reported the T>C proportions in our WGS data (lines 394-405; Fig. S11c), in study 1 & 2 (lines 96-103; Fig. S1 and S2), and in study 3 (223-226). We summarised the prevalence of T>C variants in our literature reviewed results of 20 studies (lines 114-151; Table S1).
- We reported the variability of T>C mutation patterns using n=110 FFPE samples in study 1 & 2 (lines 205-212; Fig. 1c). Considering the variance of T>C sequencing

errors induced by different polymerase shown in Table S2, it is likely to observe high variance among T>C mutations from different laboratories.

- Evidence that T>C mutations are due to an undefined batch effect is presented in numerous places: Table S1, Fig. 1, Fig. 4, Fig. S1, Fig. S2, and Fig. S6f.
- We have added the 96-channel plot of two C277 samples and their concordant mutations in Fig. S11d-f.

With regards to sample 53C: the data from this sample were erroneously listed in the raw supplementary data file, but no analysis from this case has ever been included in our manuscript. The link to the raw data has now been updated to remove this case: https://github.com/QingliGuo/FFPEsig/blob/main/Data/WGS_FFPEsamples.SBS96.csv (also provided as Table S4)

Nevertheless, given the reviewer's concern it is important we explain the provenance of this case. We sequenced two regions of colorectal cancer 53C in 2015 to generate pilot data for a grant application: we emphasise that it was not sequenced with this study in mind. The coverage for this sample was only ~5X, we are unsure if repair was used in the library preparation, and read quality did not pass the standard quality control so we abandoned the sample for this study.

Reply-to-comment 2.2

Comment 2.2 *Assuming for the sake of argument that there should be some correction applied to compensate for FFPE-associated mutations, how could that correction possibly be valid if it ignores entirely the contribution from the most frequent base substitution type? In my opinion, it would not be a valid procedure and instead, would have the counterproductive effect of applying a "correction" which is only partial and does not actually correspond to the complete effects of FFPE exposure. Mutational signature analyses should not be revised wholesale in a dubious, incomplete way based on a dodgy understanding of how to compute a reasonable correction due to mysterious "batch effects". Instead of bringing clarity to answer a perfectly valid scientific question, it would just muddy the waters.*

Reply 2.2 We understand the origin of the reviewer's concern but reiterate that the removal of T>C mutations from our analysis is entirely justified. Our analysis shows that T>C mutations do not predominantly result from formalin exposure, and further that removing them does not typically negatively impact signature assessment. A large portion of our revised manuscript details these analyses in a straightforward and explicit way.

In our revised manuscript, following reviewer #3's suggestions, we now presented our method using the adjusted 80-channel mutation spectrum (see our reply 3.1 and reply 3.2). The updated results again confirmed that our established signature analysis framework is not detrimental to removing the T>C mutations in both real and synthetic FFPE samples.

In general, in the literature SBS signatures have been assessed from 6, 96, 192 or 1536 dimensional data matrix depending whether some features (e.g. sequence context, transcribed status) are considered or not. In our study, we demonstrated 80 features (removing T>C mutations) still contain enough information to successfully deconvolve mutational signature activities in most cases in (Fig. 5 and Fig. S16). Overall, feature

selection is a well established method to deal with variables with unexplained variance and/or redundant information in data science. Therefore, our solution of omitting $T > C$ features is task-oriented and we have proven it works.

Comments from Reviewer #3

Reply-to-comment 3.0

Comment 3.0 *After reading the revised manuscript, and the comments by Reviewer 2 (both initial ones, and the second iteration thereof), I would tend to largely agree with concerns of Reviewer 2. The T>C artefacts, while not present in all the FFPE datasets, are present in multiple datasets including the authors' own data, which means they cannot be ignored.*

Reply 3.0 We completely agree that the overrepresentation of T>C mutations in some FFPE samples have to be dealt with, and indeed a large portion of manuscript has been devoted to appropriately addressing this issue.

We thank the reviewer for their constructive comments, especially for proposing to present our method in an 80-channel format. We followed this suggestion in our revised figures of real and synthetic samples, and included a recommendation to future users regarding T>C mutations. We hope the reviewer finds the new presentation is clear and consistent.

Reply-to-comment 3.1

Comment 3.1 *The authors explanation that the T>C is not due to formalin itself (but due to another unknown thing during sample prep/library prep of FFPE samples) is plausible, but does not really justify not considering these mutations in the spectrum adjustment. The suggested approach of ignoring the entire T>C part of the mutation spectrum (16 out of 96 channels, as in Fig 4 and 5) is perhaps servicable, even though this will prevent measuring some signatures e.g. SBS12 or SBS16 or reduce accuracy for other signatures. I think their overall method would (in the lack of better solution!) need to always include this step of omitting T>C mutations when generating the adjusted spectrum, since it is not in advance known if a FFPE sample will contain these T>C artefacts. It should be presented as a method that generates an 80-channel (not 96-channel) mutational spectrum, in the current state.*

Reply 3.1 We thank the reviewer for acknowledging our proposed plausible origins for the batch-related T>C mutations and for their accurate insights about our proposed solution of 80-channel spectra.

We completely agree that an 80-channel approach would affect prediction accuracy for signatures featured in T>C channels, e.g. SBS12 or SBS16 as the reviewer listed here. Nonetheless, we noted that 1) these two signatures are exclusively identified in one cancer type in PCAWG — Liver-HCC; and 2) their status of being present in a sample is not affected (binary classification instead of computing activity, Fig. S14). We now emphasised these shortcomings to the potential users in lines 482-515 and lines 532-541.

We thank the reviewer for their suggestion to report our results using the 80-channel spectrum to clarify the presentation. We revised the text and figures to highlight this limitation throughout the manuscript. And we are confident that our established framework will soon be

applicable to the complete format of data once the sources of excess T>C artefacts are experimentally identified and a standardised data collection laboratory protocol is set up for FFPE samples.

Comment 3.2 In the manuscript they should be transparent about amounts of T>C mutations in some FFPE samples (also in the figures - e.g. Fig 1 should probably have an equivalent of 1E and 1D for T>Cs, even if the 'known' signature for the T>Cs cannot be recovered). They should not dismiss the importance of these mutations in the text, since they can be a problem to a similar extent as the C>T mutations. They should be clear that a recommended use of their method/workflow is to fully ignore the T>C mutations, which will reconstruct only partial mutation spectra (however I do recognize that this is probably still better than using the uncorrected spectrum).

Reply 3.1 We thank the reviewer for this suggestion. We followed it and revised our data presentation in Fig. 1.

Fig. 1 Mutational signatures of formalin exposure. (a) C>T FFPE-only mutation count increases with formalin fixation time. We observed this increase in both unrepaired and repaired FFPE samples from study 1 (fixation group). FFPE-only mutations refers to mutations that are only discovered in FFPE but not in FF samples or known germline databases. The bar height represents the average

C>T count, and the three individual counts in $n=3$ patients are marked as black dots. (b) Consistent and separate mutation patterns in repaired and unrepaired FFPE samples ($n=110$). We clustered normalised FFPE mutation profiles 80-channel (without T>C) from study 1 and 2 using t-SNE. The normalisation is processed using the targeted region and whole genome mutational opportunities (See Methods). Each sample is classified as unrepaired (without UDG treatment, in salmon colour) or repaired (with UDG treatment, in olive-green colour) FFPE. We marked samples from study 1 as circle shape and samples from study 2 in cross shape. (c) Cluster of T>C mutations from study 1 and 2. (d) Comparison of our derived FFPE signatures to known COSMIC SBS signatures. We inferred two FFPE signatures from representative samples (see Methods; Supplementary Fig. 6a-b). (e-f) Unrepaired FFPE signature is highly similar to SBS30 (e). Repaired FFPE signature is highly similar to SBS1 (f). We treated T>C features as missing data due to the strong batch-effect found in study 1 (also observed in a few other studies in Supplementary Table 1) and therefore they were assigned to zeros. We noted that zero values are approximately close to the true mutation probabilities of T>C in FFPE datasets without this batch-effect (Supplementary Fig. 6f). We added error bars to the other 80 mutation channels, which showed the mean of standard deviation in the representative samples during the signature inference. (g-h) No consistent patterns found for T>C mutations. We derived the patterns using the same methods applied in (e-f). The error bars showed the averaged standard deviations in the representative samples where the T>C patterns are derived from.

In summary:

- We now included an equivalent set of derived patterns for T>C mutations (Fig. 1g-h), using the same signature extraction method used in Fig 1e-f. We added error bars to each T>C channel to demonstrate the large variance that prevents a “systematic” signature to be extracted.
- We also added error bars for the remaining 80-channels in Fig. 1e-f. We annotated the T>C channels in both figures with a grey box and text.
- We specified the absolute amount of T>C in our manuscript for the four datasets we analysed in our study - study 1 & 2 (lines 96-1-3), study 3 (lines 223-226) and our own CRC samples (lines 398-399).
- We moved the cluster result of T>C mutation profiles from the previous supplementary Fig. S4c to Fig. 1c.
- We added grey boxes and text annotation in Fig. S6 to highlight the omission of T>C mutations in our analysis.

Reply-to-comment 3.2

Comment 3.3 *Their synthetic bechmarks, which currently I think do not spike-in the T>C mutations, should therefore also ignore the 16 channels of the T>C spectra (and the T>C box on e.g. Fig 2 would need to be greyed out) and report results based on that, if this is not currently done.*

Reply 3.3 We thank the reviewer for this advice. We updated all results derived from simulated samples using the 80-channel spectrum as the reviewer suggested, and added a grey shade box on the Fig. 2a. Note that we continue to show all 96-channels throughout the manuscript, with T>C channels labelled “removed” (rather than omitting T>C from the plots), as this format is standardised in the literature.

The simulated samples were used to 1) study the correction performance of FFPEsig (Fig. 2, FigS8); and 2) compare the decomposition results between corrected and uncorrected FFPE

profiles (Fig. 3, Fig. S9 & S10). The main results derived from the 80-channel synthetic samples were similar to what we observed using 96-channel:

- We again observed significant improvement of inferred signature weights from corrected, compared to using uncorrected profiles (exactly as the reviewer predicted) (Fig. 3).
- The FFPEsig performance varied between cancer types (Fig. 2d).
- The two factors affecting correction performance of FFPEsig are signal-to-noise ratio and signal-to-noise similarity (Fig. 2e-f).

The changed stats are highlighted in blue colour in our revised manuscript. We thank the reviewer again for the constructive suggestions for a clearer presentation.

REVIEWERS' COMMENTS

Reviewer #3 (Remarks to the Author):

In this iteration of the revision, the authors have adequately addressed (via textual changes and analyses) the issue of the T>C (A>G) FFPE-associated sequencing artefacts of unknown origin. A final minor request I have is to be more clear about that the T>C artefacts are indeed -- as evident in their data -- probably associated with the FFPE library prep or downstream steps. I do not recall reports in the literature that a T>C artefact commonly appeared in genome sequencing and analysis of non-FFPE samples. For example, the sentence in L117-118 "Overall, T>C mutations are not randomly distributed and they are also present in all three FF samples, strongly suggesting they are batch-related non-FFPE artefacts." would be better rewritten to something like "Overall, T>C mutations are not randomly distributed, suggesting they are batch-related artefacts of unknown cause that may be associated with the FFPE sample DNA sequencing in some cases.", or such. (also other minor edits may be warranted) Were the T>C really similarly abundant in the FF samples as the FFPE samples? Fig. S4 seems to show only % concordance rather than mutation counts, so it is not evident from that visualization. Also Fig. S1b does show some increase of FFPE-only T>Cs with formalin fixation time.

Typo: line 200-201 repetition of fragment. Also mentioning the word "salmon" (referring to the color) in the text is distracting, this belongs to figure or the legend.

Comments from Reviewer #3

Comment 3.0 *In this iteration of the revision, the authors have adequately addressed (via textual changes and analyses) the issue of the T>C (A>G) FFPE-associated sequencing artefacts of unknown origin.*

Reply 3.0 We are pleased that our revised manuscript has fully addressed the reviewer's main concerns. We thank the reviewer for their helpful comments.

Comment 3.1 *A final minor request I have is to be more clear about that the T>C artefacts are indeed -- as evident in their data -- probably associated with the FFPE library prep or downstream steps. I do not recall reports in the literature that a T>C artefact commonly appeared in genome sequencing and analysis of non-FFPE samples. For example, the sentence in L117-118 "Overall, T>C mutations are not randomly distributed and they are also present in all three FF samples, strongly suggesting they are batch-related non-FFPE artefacts." would be better rewritten to something like "Overall, T>C mutations are not randomly distributed, suggesting they are batch-related artefacts of unknown cause that may be associated with the FFPE sample DNA sequencing in some cases.", or such.*

Reply 3.1 We thank the reviewer for this suggestion. We followed their advice and included the plausible origins of T>C mutations in several places in our manuscript (L119-120, L141-142, L395-396 and L602-603).

Comment 3.2 *(also other minor edits may be warranted) Were the T>C really similarly abundant in the FF samples as the FFPE samples? Fig. S4 seems to show only % concordance rather than mutation counts, so it is not evident from that visualization.*

Reply 3.2 Yes, we found the T>C mutations are similarly prevalent among FF sample pairs in study 1 (~57.8%, 1,275/2,207), compared FFPE sample pairs in that study (~58.4%, 1,260/2,156) (Fig. S4b). We added these figures in L119-121. The excess amount of T>C mutations in these FF samples further confirmed there is a strong batch effect in study 1 (due to library prep or sequencing). An excess of T>C mutations in FF samples is not typical: for instance, we checked a further $n=10$ FF CRCs where we have previously generated multi-region sequencing data, and only found ~0.02% T>C concordant mutations in this set.

Comment 3.3 *Also Fig. S1b does show some increase of FFPE-only T>Cs with formalin fixation time.*

Reply 3.3 We also noticed that T>C mutation count showed a slight increase from fixation time of 15 hours to 48 hours, but not from 2 hours to 15 hours, in repaired FFPE samples (Fig. S1b). In addition, the T>C artefact count is negatively associated with longer fixation time in unrepaired FFPE samples (slope=-0.05, Fig. S1a). We think the sudden increase of T>C in Fig. S1b is likely caused by the side effect from chemical treatment or downstream pipelines in repaired samples.

Comment 3.4 *Typo: line 200-201 repetition of fragment. Also mentioning the word "salmon" (referring to the color) in the text is distracting, this belongs to figure or the legend.*

Reply 3.4 We thank the reviewer for pointing out this typo and we corrected this in L197. We now removed the word 'salmon' in the main text and described it in the figure legend and caption. We further corrected similar issues in our main text (e.g. 'cyan').